# Microbial carbon use efficiency promotes global soil carbon storage

Feng Tao[1,2,3], Yuanyuan Huang[4], Bruce A. Hungate[5,6], Stefano Manzoni[7], Serita D. Frey[8], Michael W. I. Schmidt[9], Markus Reichstein[2], Nuno Carvalhais[2,10], Philippe Ciais[11], Lifen Jiang[12], Johannes Lehmann[13], Ying-Ping Wang[14], Benjamin Z. Houlton[15], Bernhard Ahrens[2], Umakant Mishra[16,17], Gustaf Hugelius[7], Toby D. Hocking[6], Xingjie Lu[18], Zheng Shi[19], Kostiantyn Viatkin[3,13], Ronald Vargas[3], Yusuf Yigini[3], Christian Omuto[3], Ashish A. Malik[20], Guillermo Peralta[3], Rosa Cuevas-Corona[3], Luciano E. Di Paolo[3], Isabel Luotto[3], Cuijuan Liao[1], Yi-Shuang Liang[1], Vinisa S. Saynes[3], Xiaomeng Huang[1✉] & Yiqi Luo[12✉]

Soils store more carbon than other terrestrial ecosystems[1,2]. How soil organic carbon (SOC) forms and persists remains uncertain[1,3], which makes it challenging to understand how it will respond to climatic change[3,4]. It has been suggested that soil microorganisms play an important role in SOC formation, preservation and loss[5–7]. Although microorganisms affect the accumulation and loss of soil organic matter through many pathways[4,6,8–11], microbial carbon use efficiency (CUE) is an integrative metric that can capture the balance of these processes[12,13]. Although CUE has the potential to act as a predictor of variation in SOC storage, the role of CUE in SOC persistence remains unresolved[7,14,15]. Here we examine the relationship between CUE and the preservation of SOC, and interactions with climate, vegetation and edaphic properties, using a combination of global-scale datasets, a microbial-process explicit model, data assimilation, deep learning and meta-analysis. We find that CUE is at least four times as important as other evaluated factors, such as carbon input, decomposition or vertical transport, in determining SOC storage and its spatial variation across the globe. In addition, CUE shows a positive correlation with SOC content. Our findings point to microbial CUE as a major determinant of global SOC storage. Understanding the microbial processes underlying CUE and their environmental dependence may help the prediction of SOC feedback to a changing climate.

Losses of soil organic carbon (SOC) could accelerate global warming, whereas sequestering carbon dioxide ($CO_2$) into soils as SOC can help mitigate climate change[2,16]. How organic carbon is formed and preserved in the soil has been debated for over a century and remains controversial[3,17,18]. A classical paradigm emphasizes the roles of plant carbon inputs and soil organic matter decomposition in driving SOC storage and persistence. The rates of plant primary production determine the amount of organic carbon delivered to soils through litterfall, root turnover and exudation. In addition, organic matter decomposition is the major component in determining the rate of SOC loss, as soil decomposers (mainly microorganisms) break down organic matter and release carbon back to the atmosphere as $CO_2$. Tremendous

efforts have been made to track the quantity[19] and decomposability[20] of external carbon sources to soils, and their rate of decomposition[21], variations in space and time[22,23], and the nuanced interactions with complex local environments (for example, temperature, moisture and the soil mineral matrix)[3,24,25]. Nevertheless, studies of these controls have not led to sufficiently improved quantification of SOC storage[26]. The mechanisms underlying the magnitude of global SOC storage and its spatial distributions remain largely unknown[27], hindering reliable projections of terrestrial biosphere feedback to a changing climate[28].

Recent studies have highlighted the critical roles that soil microorganisms play not only in organic carbon loss via microbial decomposition[8] but also in SOC formation and persistence as indicated

[1]Department of Earth System Science, Ministry of Education Key Laboratory for Earth System Modelling, Institute for Global Change Studies, Tsinghua University, Beijing, China. [2]Max Planck Institute for Biogeochemistry, Jena, Germany. [3]Food and Agricultural Organization of the United Nations, Rome, Italy. [4]Key Laboratory of Ecosystem Network Observation and Modeling, Institute of Geographic Sciences and Natural Resources Research, Chinese Academy of Sciences, Beijing, China. [5]Center for Ecosystem Science and Society, Department of Biological Sciences, Northern Arizona University, Flagstaff, AZ, USA. [6]School of Informatics, Computing and Cyber Systems, Northern Arizona University, Flagstaff, AZ, USA. [7]Department of Physical Geography and Bolin Centre for Climate Research, Stockholm University, Stockholm, Sweden. [8]Center for Soil Biogeochemistry and Microbial Ecology, Department of Natural Resources and the Environment, University of New Hampshire, Durham, NH, USA. [9]Department of Geography, University of Zurich, Zurich, Switzerland. [10]Departamento de Ciências e Engenharia do Ambiente, DCEA, Faculdade de Ciências e Tecnologia, FCT, Universidade Nova de Lisboa, Caparica, Portugal. [11]Laboratoire des Sciences du Climat et de l'Environnement, LSCE/IPSL, CEA-CNRS-UVSQ, Université Paris-Saclay, Gif-sur-Yvette, France. [12]School of Integrative Plant Science, Cornell University, Ithaca, NY, USA. [13]Soil and Crop Sciences Section, School of Integrative Plant Science, Cornell University, Ithaca, NY, USA. [14]CSIRO Environment, Aspendale, Victoria, Australia. [15]Department of Ecology and Evolutionary Biology and Department of Global Development, Cornell University, Ithaca, NY, USA. [16]Computational Biology and Biophysics, Sandia National Laboratories, Livermore, CA, USA. [17]Joint BioEnergy Institute, Lawrence Berkeley National Laboratory, Emeryville, CA, USA. [18]School of Atmospheric Sciences, Sun Yat-sen University, Guangzhou, China. [19]Institute for Environmental Genomics and Department of Microbiology and Plant Biology, University of Oklahoma, Norman, OK, USA. [20]School of Biological Sciences, University of Aberdeen, Aberdeen, UK. ✉e-mail: hxm@tsinghua.edu.cn; yiqi.luo@cornell.edu

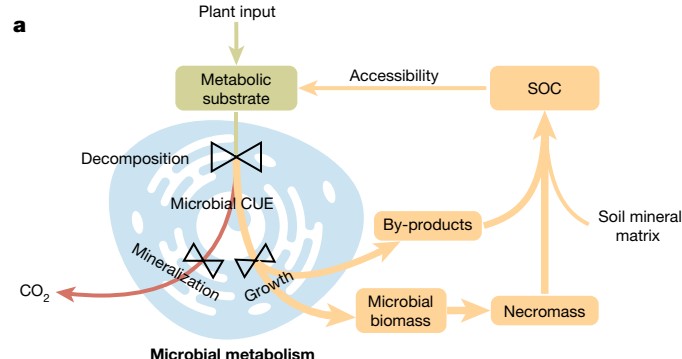

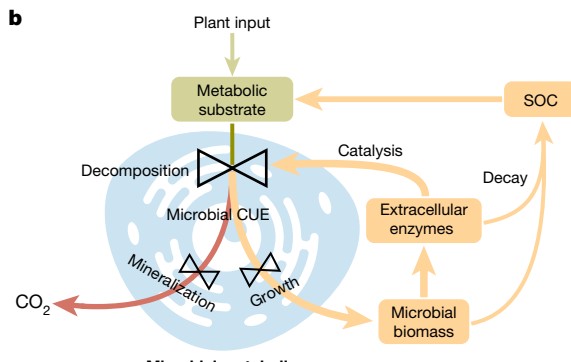

**Fig. 1 | Two contrasting pathways in determining the relationship between microbial CUE and SOC storage. a**, The first pathway indicates that a high CUE favours the accumulation of SOC storage through increased microbial biomass and by-products. **b**, The second pathway emphasizes that a high CUE stimulates SOC losses via increased microbial biomass and subsequent extracellular enzyme production that enhances SOC decomposition.

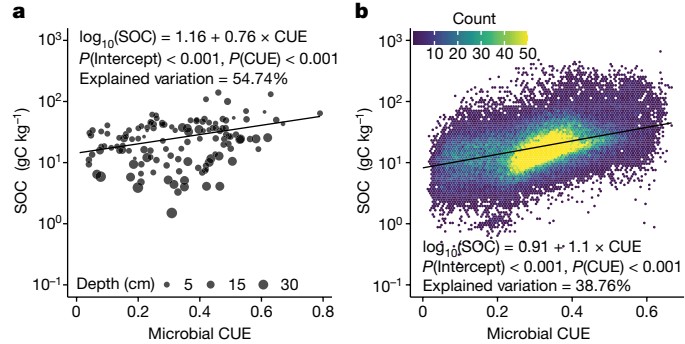

**Fig. 2 | CUE–SOC relationship. a**,**b**, The CUE–SOC relationship that emerged from the meta-analysis of 132 measurements (**a**) and data assimilation using the microbial model with 57,267 globally distributed vertical SOC profiles (**b**). The black lines and statistics shown are the partial coefficients from mixed-effects model regressions (see Extended Data Tables 1 and 2 for details).

by the covariance between microbial biomass, necromass and SOC content[4,6,9–11] (Fig. 1). Although there are many pathways through which microorganisms affect both the accumulation and loss of soil organic matter, microbial carbon use efficiency (CUE) is an integrative metric that captures the balance of these processes. CUE describes the microbial partitioning of carbon used in metabolism that goes towards growth versus respiration and, thereby, expresses a dual microbial control point between SOC accumulation and loss. Although it has the potential to act as a strong predictor of variation in SOC storage around the world, the role of CUE in SOC persistence is ambiguous in at least two ways. First, whether CUE is positively or negatively correlated with SOC storage is under debate[7,14,15]. Second, the relative influence of CUE vis-a-vis other controls on SOC storage remains poorly resolved[3,4]. Here we examined the relationship between CUE and the preservation of carbon as SOC, and interactions with climate, vegetation and edaphic properties, using a combination of global-scale datasets, a microbial-process explicit model, data assimilation, deep learning and meta-analysis.

A high CUE promotes biosynthesis in microbial carbon metabolism[12,13], causes the accumulation of microbial by-products and necromass that favours SOC formation (for example, via the entombing effect)[5,9,29,30], and could generate a positive relationship between CUE and SOC storage (Fig. 1a). Alternatively, a high CUE promotes microbial biomass production, enhances extracellular enzyme production[31] and could eventually trigger SOC loss over time (for example, via the priming effect)[7,30,32] (Fig. 1b). If the second pathway dominates the role of CUE in SOC storage, a negative CUE–SOC relationship would be expected. To distinguish the relative strength of these two pathways, we first collated 132 pairs of measured CUE and SOC content at 46 locations

across continents from 16 experimental studies previously published in the peer-reviewed literature (Extended Data Fig. 1a and Supplementary Table 1). Microbial CUE is positively correlated with SOC content after accounting for the methodological differences across studies (Fig. 2a and Extended Data Table 1). A high CUE not only accompanies high microbial biomass carbon but also has a positive correlation with non-microbial biomass carbon (that is, the remaining amount of organic carbon after excluding microbial biomass; Supplementary Table 2). Thus, our meta-analysis supports the idea that the first pathway plays a dominant role in SOC storage, and that high microbial CUE is mainly associated with high SOC storage.

To explore whether the positive CUE–SOC relationship obtained from local experiments is widespread across the globe, we retrieved CUE from 57,267 globally distributed vertical SOC profiles (Extended Data Fig. 1b; see Methods for data sources) by a process-guided deep learning and data-driven modelling (PRODA) approach (Extended Data Fig. 2; see Methods and refs. 33,34 for details). The PRODA approach first fuses a process-based model with SOC observations by a data-assimilation algorithm to estimate parameters for each of the SOC profiles. A steady-state assumption for the soil carbon cycle (that is, SOC storage does not change with time) at each observational profile is employed to facilitate computation (Methods). Then a deep learning model (that is, a multilayer neural network) generalizes the results from the site-level data assimilation to obtain the globally optimized parameterization and maximally match model simulations to observations. The process-based model we used (referred to as 'microbial model' hereafter) explicitly partitions carbon substrates obtained from enzymatic depolymerization between microbial biomass accrual and microbial respiration[7,35] (Methods and Extended Data Fig. 3). In this way, CUE is calculated as the ratio of microbial growth over carbon uptake (equation (1) in Methods)—the same definition adopted in the empirical studies as in the meta-analysis (Fig. 2a). Although the microbial model could generate negative, null or positive relationships between CUE and SOC storage when model parameters change (for example, turnover time for microbial mortality and enzyme decay, as shown in Extended Data Fig. 4), here we used observed SOC profiles to inform parameter estimation and thus identify the most probable CUE–SOC relationship under a Bayesian inference framework (Methods).

The CUE retrieved from 57,267 globally distributed vertical SOC profiles with the microbial model showed a positive correlation with SOC (Fig. 2b, Extended Data Table 2 and Supplementary Table 3). The results after data assimilation (that is, the first procedure of the PRODA approach) also confirmed the association of high CUE values with both microbial and non-microbial organic carbon storage (Supplementary Table 4). Although many modelling[7,14,32,36] and empirical studies[31,37–39] have suggested that a high CUE stimulates microbial

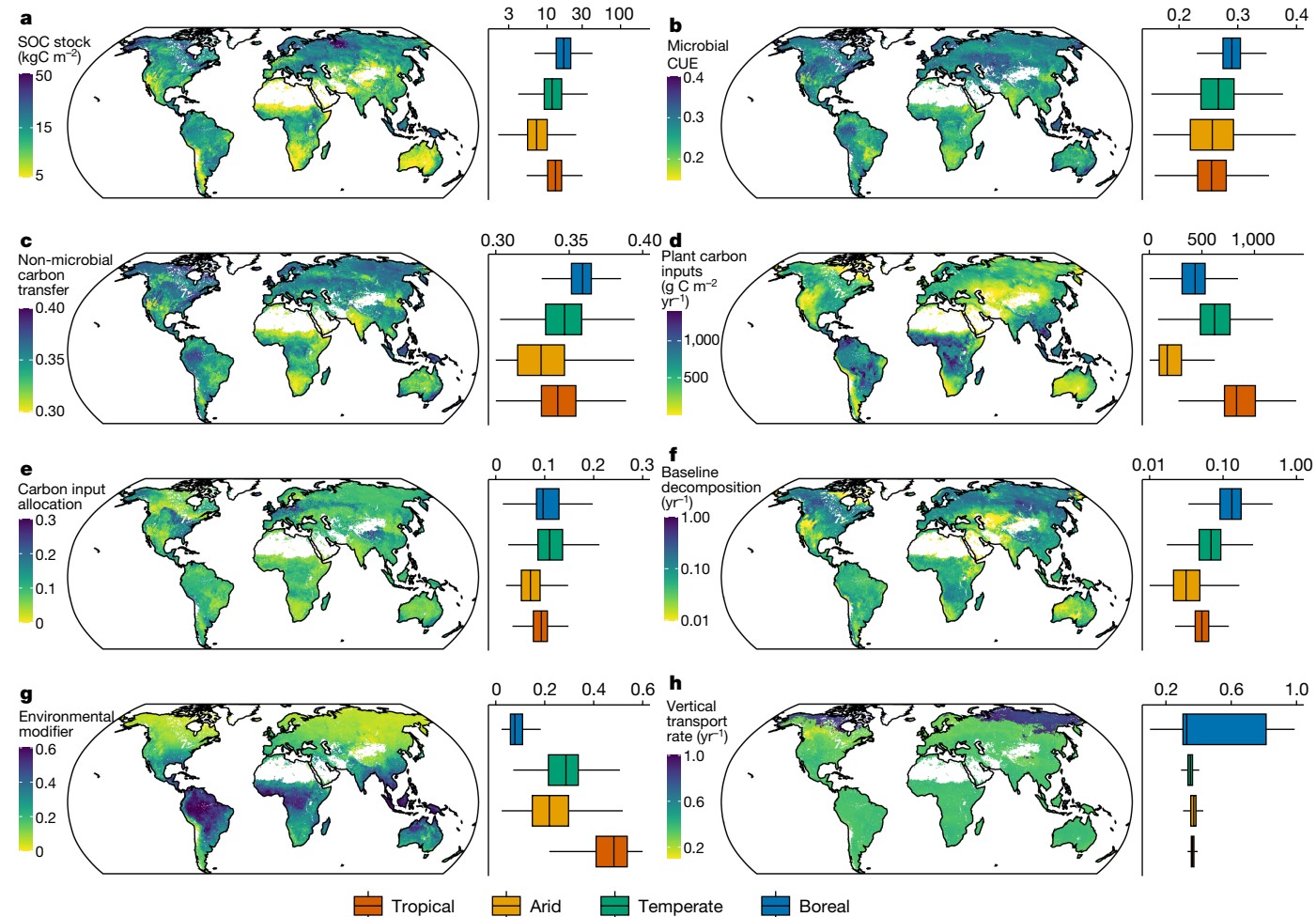

**Fig. 3 | Maps of global SOC stock and related components.** The maps were obtained from 57,267 globally distributed vertical soil profiles using the PRODA approach with the microbial model (see Methods for details). **a**–**h**, Global distributions of SOC stock (**a**), microbial CUE (**b**), non-microbial carbon transfer (**c**), plant carbon inputs (**d**), carbon input allocation (**e**), baseline decomposition (**f**), environmental modifier (**g**) and vertical transport rate (**h**). Values shown are predicted by the best-guess model calibrated using all available data

(Methods; see Extended Data Fig. 7 for their uncertainties). Boxplots represent the SOC stock for the top 1 m and model components in different pre-defined climate zones (Supplementary Fig. 3). The lower, middle and upper hinges show the first, median and third quartiles of the distribution. Whiskers in the boxplot represent the 1.5 times the interquartile range from the hinges. The components and units for the maps and the boxplots are the same.

exoenzyme activities and possibly triggers respiratory carbon loss (Fig. 1b), our data-assimilation results indicate that a high CUE predominantly promotes SOC storage. Our results agree with findings from our meta-analysis (Fig. 2a), a landscape-scale pattern across the United Kingdom[10] and results from a soil microcosm study[9]. Meanwhile, our data-assimilation results indicate that the positive CUE–SOC relationship dampens with soil depth (Extended Data Fig. 5), possibly indicating stronger interactions of organic matter with mineral particles at depth[3].

We assessed the importance of microbial CUE on global SOC storage relative to six other model components: plant carbon inputs, partitioning of input carbon into different soil layers, the fraction of carbon transfer that is not mediated by microbial processes (that is, non-microbial carbon transfer hereafter), substrate decomposability, environmental modifications that account for the effects of environmental conditions on SOC decomposition processes, and vertical transport (see equation (3) in Methods). We used the full dataset of 57,267 SOC profiles to optimize the microbial model at the global scale by the PRODA approach. The optimized microbial model explains 54% (median value of 53%, $2\sigma$ confidence interval 52–55% from 200-time bootstrapping) of the spatial variation in observed SOC. Moreover, the PRODA-retrieved CUE values in those pixels where CUE was measured in the meta-analysis agree well with the measured values in a mixed-effects

model considering methodological differences across sites ($R^2 = 0.54$, the regression slope is not significantly different from 1 at a significance level of 0.05; Supplementary Table 5).

Using the microbial model with the optimized parameter values, we calculated system-level values of the seven components (that is, CUE, plant carbon inputs, carbon input allocation, non-microbial carbon transfer, substrate decomposability, environmental modifications and vertical transport) to assess their relative importance in determining SOC storage over the globe (Fig. 3 and Supplementary Table 6). For example, the system-level CUE ($\text{CUE}_{\text{system}}$) was calculated by weighting CUE values from three litter and one dissolved organic carbon (DOC) assimilation pathways with their respective fluxes (equation (10) in Methods). $\text{CUE}_{\text{system}}$ is well correlated (Pearson correlation coefficient = 0.98, d.f. = 56,270, $P < 0.001$) with the microbial CUE of the mineral soil part of the model (that is, $\eta_{\text{DOC}}$ in Supplementary Table 6; Supplementary Fig. 1).

Over the globe, CUE and SOC share similar patterns, with lower $\text{CUE}_{\text{system}}$ and SOC stocks in tropical than in boreal regions (Fig. 3a,b and Supplementary Fig. 2). $\text{CUE}_{\text{system}}$ is also positively correlated with the fraction of carbon transfer that is not mediated by microbial processes but may be related to organo–mineral interactions (Fig. 3c). The spatial pattern in $\text{CUE}_{\text{system}}$ is consistent with theoretical arguments

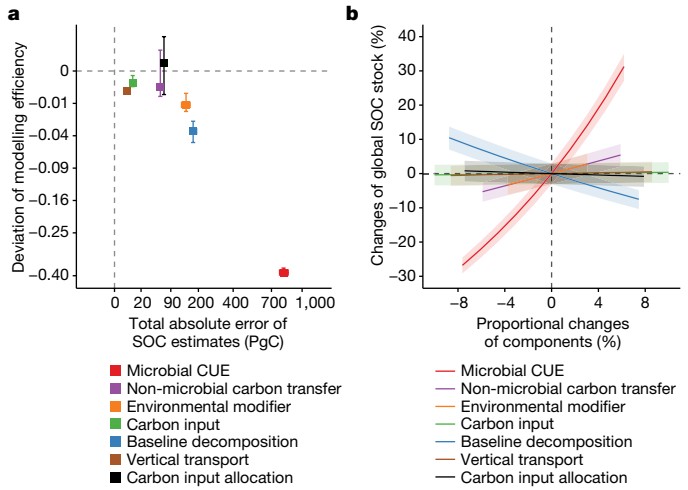

**Fig. 4 | Microbial CUE as the primary regulator of global SOC storage.**
**a**, Spatially explicit model components obtained through the PRODA approach with the microbial model were substituted by their spatially invariant counterparts to assess their influence on SOC stocks (that is, the sum of absolute deviations from PRODA estimates over the globe) and distributions (that is, deviation of explained variation in observations defined by equation (2)).
**b**, We further proportionally changed the retrieved parameter values of different components and found that global SOC stock is most sensitive to changes of CUE. Error bars and shaded areas show the $2\sigma$ confidence interval in the 200-time bootstrapping. It is noted that the axes in **a** are scaled by a signed square-root function.

that microorganisms in warm environments should reduce allocation to biosynthesis because of the increased energy requirements for metabolism[14]. However, different spatial patterns of microbial CUE may emerge when estimation of CUE considers nutrient limitation to microbial metabolism, stoichiometric adjustments[40–42] and food-web interactions[43]. In comparison, primary producers in the tropics generate much more carbon inputs than in boreal regions (Fig. 3d) and deliver a comparable proportion of the carbon input to deeper soils through their rooting systems and other processes (Fig. 3e). These processes do not lead to high SOC stocks in the tropics owing to low carbon retention in microbial biomass as implied by the low CUE$_{system}$ values and high decomposition rates.

Organic carbon loss from soil via decomposition is jointly determined by substrate decomposability (or baseline decomposition rate at reference temperature and moisture levels) and environmental modifications. We found a trade-off between substrate decomposability and limitations imposed by soil temperature and moisture. In boreal regions, SOC is more vulnerable to loss as indicated by high baseline decomposition rates (Fig. 3f), probably resulting from high accessibility of SOC substrates owing to, for example, the scarcity of interactions with soil minerals in organic soils[44]. Yet, the cold climates restrict organic carbon loss by reducing decomposition rates through environmental stresses (for example, low temperature, permafrost immobilization and waterlogging in peatlands) as represented by the low environmental modifier values (Fig. 3g). Moreover, vertical movement that transports organic matter from surface to subsoils is more significant in the permafrost region than elsewhere, possibly owing to high rates of cryoturbation (Fig. 3h).

We conducted two sensitivity analyses to assess the relative importance of the seven components to global SOC storage. The first analysis quantified the relative importance of one individual component using the change in simulated global SOC with or without spatial variation of that component while allowing the other parameters to vary (Supplementary Table 6 and Methods). This analysis is particularly relevant to Earth system models, which mostly do not account for spatial variations

of model parameters within a pre-defined classification (for example, plant functional types)[45]. Results from the microbial model point to CUE as the most important predictor of the global SOC stock and its spatial variation compared with the other components (Fig. 4a). Globally invariant CUE causes a total absolute deviation of SOC estimates by 812 PgC in the top 1 m of soil ($2\sigma$ confidence interval 774–857 PgC) from the originally PRODA-estimated 1,358 PgC ($2\sigma$ confidence interval 1,317–1,399 PgC). Correspondingly, the explained spatial variation in SOC decreased from the original 53% ($2\sigma$ confidence interval 52–55%) to 15% ($2\sigma$ confidence interval 14–16%) (Fig. 4a). In contrast, setting spatially uniform parameter values for the other model components resulted in fewer biases in simulated global SOC storage and its spatial variation than that caused by CUE (Fig. 4a). When all parameters related to all model components except carbon inputs (that is, CUE, carbon input allocation, non-microbial carbon transfer, substrate decomposability, environmental modifications and vertical transport) are fixed to the global means of their retrieved values, the microbial model could explain only 11% of the spatial variation in observations.

The second sensitivity analysis assesses how the global SOC storage changes in response to proportional changes of each of the seven components over the globe (Methods). The result shows that a 10% increase in global SOC storage (equivalent to an additional 136 Pg of SOC accumulation worldwide) requires a 2.0% increase in CUE (that is, an increase of global median CUE$_{system}$ from 0.28 to 0.29), an 8.4% decrease in baseline SOC decomposition rate (that is, a decrease of the global median from 0.066 yr$^{-1}$ to 0.060 yr$^{-1}$), an 11% increase in environmental modifiers (that is, equivalent to an increase of global median temperature from 10.3 °C to 12.2 °C, or an increase of global median precipitation from 503 mm to 611 mm), or an 11% increase in non-microbial carbon transfer fraction (that is, an increase of the global median from 0.35 to 0.39) (Fig. 4b). Changes of carbon inputs, allocation and vertical transport only marginally influence the global SOC storage. Although the two sensitivity analyses evaluated the relative importance of the seven model components, projecting future changes in SOC given this knowledge will need to examine the full space of temporal variations in parameters and changing environments, and consider microbial physiological acclimation and genetic adaptations. Moreover, future studies need to carefully examine how sensitive the evaluation of the relative importance of CUE to global SOC storage is to different model structures.

Microbial CUE is influenced by and interacts with different environmental variables. A permutational analysis (Methods) indicates that soil structural properties explain more of the CUE spatial variability than geographic, climatic, soil chemical and vegetation variables (Extended Data Fig. 6a). Previous studies have discussed the importance of soil structural variables such as bulk density, texture and porosity in affecting microbial activities. A well structured soil with medium physical heterogeneity may help foster niche complementarity for diverse soil microbial communities and eventually benefit high CUE[46–48]. In turn, accumulation of SOC owing to high CUE, could also benefit the development of fertile soils. More quantitative understanding is needed on the mechanistic relationships between CUE and soil structural variables to facilitate the effective management of soil carbon storage in the future.

This study provides evidence from global-scale observations that microbial CUE plays a pivotal role in determining SOC storage. On the basis of information retrieved from global vertical soil profiles using the PRODA approach, we found that microbial CUE is at least four times as important in determining SOC storage at the global scale as any of the other six components evaluated: plant carbon inputs, carbon input allocation, non-microbial carbon transfer, substrate decomposability, environmental modifications and vertical transport. Moreover, our results from the microbial model and meta-analysis of field observations support the argument that a high microbial CUE promotes SOC storage more than loss. The positive relationship between microbial CUE and SOC reflects organic carbon partitioning by microbes: a high

CUE means more allocation to biomass and by-products, which leads to SOC accumulation, whereas a low CUE value indicates the partitioning of more carbon towards cellular respiration, which drives SOC loss. Our findings help prioritize future research on microbial processes in addition to SOC decomposition and organic carbon input for improving prediction of SOC dynamics. Further understanding of microbial processes underlying CUE and their environmental dependence will be critical to both predicting SOC feedbacks to changing climate and enhancing SOC sequestration.

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

## Methods

### Sources of SOC data

We used two data sources for this study. One is from experimental studies published in the literature for a meta-analysis. The other is from globally distributed soil profiles to retrieve processes underlying SOC storage.

We obtained published data from experimental studies that simultaneously reported SOC content (unit gC kg$^{-1}$ soil) and CUE, by searching the literature via Web of Science (https://apps.webofknowledge.com). The key words used were: 'soil' AND 'carbon use efficiency' AND ('organic carbon' OR '*biomass carbon') AND 'incubation'. This study collected data only from studies where CUE was measured as ratio of microbial biomass production over substrate uptake as described below:

$$CUE = \frac{biomass\ production}{substrate\ uptake} \quad (1)$$

These studies used isotope (that is, $^{13}C$, $^{14}C$ or $^{18}O$)-labelled substrates. In cases of manipulation experiments (for example, fertilization experiments), we included only the data from control plots. In total, we found 132 pairs of data with both CUE and SOC content measured at 46 locations from 16 studies (Extended Data Fig. 1a and Supplementary Table 1). Our meta-analysis covered the main vegetation types of the world yet there is still a lack of data in tropical and arid regions and clay soils (Supplementary Fig. 4). The mean (variance) values of the measured CUE are 0.33 (0.049) using the carbon isotopes $^{13}C$ or $^{14}C$ and 0.33 (0.022) using the oxygen isotope $^{18}O$.

The relationship between CUE and SOC content was tested by linear mixed-effects models[49,50] (Extended Data Table 1). In the linear mixed-effects model, we took CUE, mean annual air temperature (MAT) and depth of the measurement as the fixed effects to predict SOC content. We expected these factors to have an effect. We then added the study sources ('Source' in Supplementary Table 1, $n = 16$) as the random effects to acknowledge that different studies have used different methodologies to estimate CUE. We explored different structures of mixed-effects models (that is, random intercepts with common slopes or random intercepts with random slopes) to test the CUE–SOC relationship. Among all the fixed-effect variables (that is, CUE, MAT and soil depth), only the correlation between CUE and MAT is significant at the significance level of 0.05 (Pearson correlation coefficient = −0.24, $P = 0.005$, d.f. = 130). Therefore, the interaction between CUE and MAT was also considered in the mixed-effects model. The results of model structures that converged in regressions are presented in Extended Data Table 1. We found that the variance inflation factors for the main-effects predictors were all low; thus, including the interaction term did not improve the prediction power of the mixed-effects model. Accordingly, we did not include in the main text the interaction term between CUE and MAT in the mixed-effects model, but considered random intercepts with common slopes in regression. In addition, we explicitly separated results measured by different isotopes (that is, $^{13}C/^{14}C$ or $^{18}O$) and conducted the same mixed-effects model regressions. The positive CUE–SOC relationship is not affected by the kinds of isotopes used in the measurements (Supplementary Table 7).

We obtained organic carbon data in globally distributed soil profiles from the World Soil Information Service (WoSIS). WoSIS compiled soil data, after quality assessment, from soil profiles distributed across 173 countries[51]. The 2019 snapshot of the WoSIS dataset consists of 111,380 soil profiles with SOC content information (unit gC kg$^{-1}$ soil). We estimated the SOC stock (gC m$^{-3}$) by SOC stock = SOC content × BD (ref. 52), where BD is the bulk density of soil (g m$^{-3}$). It is noted that when calculating the total global SOC stock, the volumetric coarse fragment fraction ($G$, unit %) at each grid of the global map (data source: SoilGrids, https://soilgrids.org) was considered as a multiplier (that is, $1 - \frac{G}{100}$) to estimated SOC stock. When the measured bulk density was absent from the dataset, we used a pedo-transfer function to estimate the bulk density[52,53]: BD = $\alpha + \beta \times \exp(-\gamma \times OM)$, where OM is organic matter, calculated as SOC × 1.724, with SOC content in per cent (%), and $\alpha$, $\beta$ and $\gamma$ are fitting parameters. After fitting the WoSIS data (that is, 78,913 layers from 16,248 profiles that simultaneously recorded bulk density and SOC content) to this equation, we obtained $\alpha = 0.32$, $\beta = 1.30$ and $\gamma = 0.0089$. The pedo-transfer function explained 55% of the variation in the bulk density. In addition, we obtained data from ref. 54 and the Northern Circumpolar Soil Carbon Database[55]. This dataset contained 2,546 soil profiles with SOC stock (gC m$^{-3}$) information for permafrost regions in North America, northern Eurasia and the Qinghai–Tibet Plateau. In total, we obtained data from 113,926 soil profiles as the raw data from these two data sources. The geographical distributions of all soil profiles are shown in Extended Data Fig. 1b.

We pre-processed the 113,926 SOC profiles to ensure the quality of the data before we conducted our analysis in this study. We first excluded SOC profiles with no more than two observation layers or the maximum observation depths of no deeper than 50 cm from this study as such data do not provide enough information on key processes underlying SOC storage. After that, we obtained 72,377 profiles.

To further examine the quality of data along the vertical profiles, we conducted data assimilation for each of the 72,377 SOC vertical profiles with a microbial model using the Markov chain Monte Carlo (MCMC) method. The structure of the microbial model is described in 'The microbial model for representing processes underlying SOC storage'. The method of data assimilation is briefly described in 'Process-guided deep learning and data-driven modelling' and in detail by ref. 33. We used two statistics (that is, the Gelman–Rubin (G–R) statistic and the coefficient of efficiency) to ensure the quality of the SOC data along the vertical profiles. We calculated the G–R statistic[56] for each of the SOC profiles to test the convergence of the site-level data-assimilation results after running three independent series of MCMC simulations. A G–R value approaching 1.0 suggests well converged data-assimilation results. A large G–R value, in contrast, indicates inconsistent data-assimilation results from independent MCMC simulations, and such results may not offer enough information to constrain estimation of parameters[57]. Therefore, we set a threshold of G–R = 1.05 to exclude SOC profiles from our analysis in this study. The remaining 59,476 profiles went through the next analysis below.

We used the coefficient of efficiency or Nash–Sutcliffe modelling efficiency coefficient[58] ($E$) to evaluate the effectiveness of retrieving information from observations by the microbial model. The $E$ is expressed as:

$$E = 1 - \frac{\sum (obs_i - mod_i)^2}{\sum (obs_i - \overline{obs_i})^2} \quad (2)$$

where obs is the SOC observation, and mod is the simulated SOC by the microbial model. A value of $E$ close to 1 indicates that SOC distributions with depth can be well captured by the microbial model so that information contained in the observations can be retrieved to evaluate processes underlying SOC storage. A small value of $E$ indicates that the model cannot capture the variability in the data, suggesting that such SOC vertical profiles may not offer enough information on the processes underlying SOC storage investigated in this study. We set a threshold $E = 0.0$ to exclude SOC profiles from the analysis. We randomly selected a subset of these excluded SOC profiles to visually cross-check their shapes. We found that the threshold $E = 0.0$ is effective for controlling the quality of data.

After all the data pre-processing procedures, we eventually obtained data-assimilation results from 57,267 soil profiles, with which we estimated global SOC storage and its components. Our data pre-processing criteria did not cause significant discrimination against profiles

belonging to specific soil orders or ecosystems (Supplementary Fig. 4). The SOC profiles eventually used in this study present inclusiveness to different vertical shapes (Supplementary Fig. 5). Although the majority of the 57,267 profiles (66.2%) show monotonically decreasing SOC storage with soil depths, 4.4% of them record the highest SOC storage at the middle of the soil depths and 29.4% of them show zigzagged SOC storage with increasing soil depths. Thus, the main conclusions drawn from this study are unlikely to be influenced by our data pre-processing criteria.

## The microbial model for representing processes underlying SOC storage

We constructed a microbial model (Extended Data Fig. 3) to examine the role of microbial CUE in global SOC storage. The microbial model follows the same structure proposed by ref. 7 for soil carbon dynamics, which is embedded within the structure for 20-layered vertical soil profiles. The latter was adopted from the Community Land Model version 5 (CLM5)[59,60]. Organic carbon dynamics represented by the microbial model can be expressed in a matrix equation[61]:

$$\frac{d\mathbf{X}(t)}{dt} = \mathbf{B}I(t) + \mathbf{A}\xi(t)\mathbf{K}\mathbf{X}(t) + \mathbf{V}(t)\mathbf{X}(t) \tag{3}$$

where $\mathbf{X}$ is the states of different carbon pools in the soil system, and $t$ indicates time. This matrix equation has six terms ($I(t)$, $\mathbf{B}$, $\mathbf{A}$, $\xi(t)$, $\mathbf{K}$ and $\mathbf{V}(t)$) and represents the seven model components that were investigated in this study: plant carbon input ($I(t)$), carbon input allocation ($\mathbf{B}$), substrate decomposability (or baseline decomposition rates) ($\mathbf{K}$), microbial CUE (the elements of matrix $\mathbf{A}$ that transfer carbon to the microbial biomass carbon pool), carbon transfer without the involvement of microbial processes (all the other elements of matrix $\mathbf{A}$), environmental modifier ($\xi(t)$) and vertical transport ($\mathbf{V}(t)$).

The microbial model as described by equation (3) is a block matrix equation with 160 dimensions to represent 8 pools in each of the 20 soil layers. Vector $\mathbf{X}(t)$ has 8 block elements, respectively, to represent four litter carbon pools (that is, coarse woody debris (CWD), metabolic litter (ML), cellulose litter (CL) and lignin litter (LL) carbon pools) and four mineral SOC pools (that is, DOC, mineral-associated soil organic carbon (mSOC), microbial biomass carbon (MIC) and extracellular enzymes (ENZ)):

$$\mathbf{X}(t) = \begin{bmatrix} \mathbf{x}_{CWD}(t) \\ \mathbf{x}_{ML}(t) \\ \mathbf{x}_{CL}(t) \\ \mathbf{x}_{LL}(t) \\ \mathbf{x}_{DOC}(t) \\ \mathbf{x}_{MIC}(t) \\ \mathbf{x}_{ENZ}(t) \\ \mathbf{x}_{mSOC}(t) \end{bmatrix} \tag{4}$$

Each of the 8 block elements (that is, $\mathbf{x}_i(t)$) of $\mathbf{X}(t)$ has 20 elements to represent the 20 soil layers. Similarly, there are 160 dimensions for vector $\mathbf{B}$ of carbon input allocation, matrix $\mathbf{K}$ of substrate decomposability, matrix $\mathbf{A}$ of both microbial carbon partitioning to microbial respiration versus accrual to soil organic matters and carbon transfer that is not mediated by microbial processes, matrix $\xi(t)$ of environmental modifier and matrix $\mathbf{V}(t)$ of vertical transport. Plant carbon input ($I(t)$) is a scalar.

Plant carbon input ($I(t)$) is allocated to different litter pools in different layers along the soil profile via the allocation vector $\mathbf{B}$. Organic carbon in pool vector $\mathbf{X}(t)$ can be either decomposed by the soil microorganisms or broken down without going through microbial metabolism according to the baseline decomposition matrix $\mathbf{K}$:

$$\mathbf{K} = \mathrm{diag}\begin{pmatrix} \mathbf{k}_{CWD} \\ \mathbf{k}_{ML} \\ \mathbf{k}_{CL} \\ \mathbf{k}_{LL} \\ \mathbf{k}_{DOC}(\mathbf{x}_{DOC}, \mathbf{x}_{MIC}) \\ \mathbf{k}_{MIC} \\ \mathbf{k}_{ENZ} \\ \mathbf{k}_{mSOC}(\mathbf{x}_{mSOC}, \mathbf{x}_{ENZ}) \end{pmatrix} \tag{5}$$

Whereas all the litter organic carbon pools and two mineral organic carbon pools (that is, MIC and ENZ) are decomposed following first-order kinetics where their baseline decomposition rates are constants, the baseline decomposition rates of DOC and mSOC are functions of carbon pool states. Specifically, the baseline decomposition rate of DOC (also known as the baseline rate of microbial assimilation of DOC) is: $\mathbf{k}_{DOC}(\mathbf{x}_{DOC}, \mathbf{x}_{MIC}) = \frac{v_{max,assim}\mathbf{x}_{MIC}}{K_{m,assim}\xi + \mathbf{x}_{DOC}}$; the baseline decomposition rate of mSOC is: $\mathbf{k}_{mSOC}(\mathbf{x}_{mSOC}, \mathbf{x}_{ENZ}) = \frac{v_{max,decom}\mathbf{x}_{ENZ}}{K_{m,decom}\xi + \mathbf{x}_{mSOC}}$. Parameters $v_{max,assim}$ and $v_{max,decom}$ represent the maximum DOC assimilation and mSOC decomposition rates, respectively. $K_{m,assim}$ and $K_{m,decom}$ are the Michaelis constants for DOC assimilation and mSOC decomposition, respectively. Moreover, we used the environmental modifier (that is, $\xi(t)$) to account for the effects of environmental conditions on the decomposition processes. $\xi(t)$ is calculated from functions of soil temperature ($\xi_T$), soil water potential ($\xi_w$), nitrogen and oxygen availability ($\xi_{N-O}$) and soil depth ($\xi_D$).

The decomposed organic carbon is either partitioned by microorganisms to accrued microbial biomass via microbial growth versus $CO_2$ released to the atmosphere by microbial respiration (that is, microbial CUE), or, alternatively, transferred to other carbon pools with a fraction that is not mediated by microbial processes (that is, non-microbial carbon transfer). All these processes can be summarized in the $\mathbf{A}$ matrix:

$$\mathbf{A} = \begin{bmatrix} -1 & 0 & 0 & 0 & 0 & 0 & 0 & 0 \\ 0 & -1 & 0 & 0 & 0 & 0 & 0 & 0 \\ \mathbf{a}_{CL,CWD} & 0 & -1 & 0 & 0 & 0 & 0 & 0 \\ \mathbf{a}_{LL,CWD} & 0 & 0 & -1 & 0 & 0 & 0 & 0 \\ 0 & \mathbf{a}_{DOC,ML} & \mathbf{a}_{DOC,CL} & 0 & -1 & \mathbf{a}_{DOC,MIC} & 1 & \mathbf{a}_{DOC,mSOC} \\ 0 & \mathbf{a}_{MIC,ML} & \mathbf{a}_{MIC,CL} & \mathbf{a}_{MIC,LL} & \mathbf{a}_{MIC,DOC} & -1 & 0 & 0 \\ 0 & 0 & 0 & 0 & 0 & \mathbf{a}_{ENZ,MIC} & -1 & 0 \\ 0 & 0 & 0 & \mathbf{a}_{mSOC,LL} & 0 & \mathbf{a}_{mSOC,MIC} & 0 & -1 \end{bmatrix} \tag{6}$$

where all the block elements in the $\mathbf{A}$ matrix ($\mathbf{a}_{i,j}$) are a diagonal matrix with a dimension of 20. $\mathbf{a}_{i,j}$ represents the carbon transfer fraction from the donor ($j$) pool to the recipient ($i$) pool (see carbon transfer flows in Extended Data Fig. 3). Because DOC is always assimilated by the microbes to microbial biomass with release of $CO_2$ (Extended Data Fig. 3), the microbial CUE for DOC ($\eta_{DOC}$) equals $\mathbf{a}_{MIC,DOC}$. In contrast, organic carbon in the metabolic, cellulose and lignin litter pools is decomposed by microbes to generate $CO_2$ and grow biomass while a fraction of litter organic carbon is broken down without going through microbial metabolism and, thus, directly transferred to DOC or mSOC. In this case, the microbial CUE for the three litter carbon pools can be expressed as: $\eta_{ML} = \frac{\mathbf{a}_{MIC,ML}}{1 - \mathbf{a}_{DOC,ML}}$, $\eta_{CL} = \frac{\mathbf{a}_{MIC,CL}}{1 - \mathbf{a}_{DOC,CL}}$ and $\eta_{LL} = \frac{\mathbf{a}_{MIC,LL}}{1 - \mathbf{a}_{mSOC,LL}}$, respectively. In 'Global maps of SOC and underlying model components', we separately assessed the relative importance of the CUE and the other non-microbial carbon transfer fractions in the matrix $\mathbf{A}$ to global SOC storage. It should be noted that in the field experiment, the measured microbial CUE is closer to $\eta_{DOC}$ because the litter is often removed from the soil samples before incubation. Thus, we compared the microbial

CUE for the mineral soil (that is, $\eta_{DOC}$) with field measurements in the meta-analysis.

The transport matrix $\mathbf{V}$ of the microbial model is a tridiagonal matrix and describes vertical carbon movement between adjacent soil layers within the same litter carbon pool via bioturbation, cryoturbation and leaching. The matrix representation for process-based soil carbon cycle models has been described in detail by refs. 61–63.

Equation (3) can be separated into two equations: one for litter carbon cycle and the other for mineral SOC cycle, because there is no carbon transfer from mineral soil carbon pools to litter carbon pools (that is, $\mathbf{a}_{litter\ pool,soil\ pool} = 0$ in the $\mathbf{A}$ matrix). As $\mathbf{A}$, $\mathbf{K}$, $\boldsymbol{\xi}(t)$ and $\mathbf{V}$ are all independent from litter carbon pool states (that is, $\mathbf{X}$), the analytical solution of litter carbon stock at the steady state (SS) can be calculated as $\mathbf{X}_{litter,SS} = [\mathbf{A}_{litter}\ \overline{\xi(t)_{litter}}\mathbf{K}_{litter} + \overline{\mathbf{V}(t)_{litter}}]^{-1}[-\mathbf{B}_{litter}\ \overline{\mathbf{I}(t)_{litter}}]$. For the mineral soil organic carbon pools, the related $\mathbf{K}$ matrix is carbon pool state dependent (equation (5)). We calculated their steady-state solutions according to a method reported by ref. 64 (see Supplementary Information for a detailed explanation):

$$\mathbf{X}_{soil,SS} = \begin{bmatrix} \mathbf{x}_{DOC,SS} \\ \mathbf{x}_{MIC,SS} \\ \mathbf{x}_{ENZ,SS} \\ \mathbf{x}_{mSOC,SS} \end{bmatrix}$$

$$= \begin{bmatrix} \dfrac{\mathbf{k}_{MIC}\xi K_{m,assim}\xi \mathbf{x}_{MIC,SS} - \mathbf{u}_{MIC}K_{m,assim}\xi}{(\eta_{DOC}\upsilon_{max,assim} - \mathbf{k}_{MIC})\xi \mathbf{x}_{MIC,SS} + \mathbf{u}_{MIC}} \\[3mm] \dfrac{\mathbf{u}_{MIC} + \eta_{DOC}(\mathbf{u}_{mSOC} + \mathbf{u}_{DOC})}{(1 - \eta_{DOC})\mathbf{k}_{MIC}\xi} \\[3mm] \dfrac{\mathbf{a}_{ENZ,MIC}\mathbf{k}_{MIC}\mathbf{x}_{MIC,SS}}{\mathbf{k}_{ENZ}} \\[3mm] \dfrac{(\mathbf{u}_{mSOC} + \mathbf{a}_{mSOC,MIC}\mathbf{k}_{MIC}\xi \mathbf{x}_{MIC,SS})K_{m,decom}\xi}{(\upsilon_{max,decom}\xi \mathbf{x}_{ENZ,SS} - \mathbf{a}_{mSOC,MIC}\mathbf{k}_{MIC}\xi \mathbf{x}_{MIC,SS} - \mathbf{u}_{mSOC})} \end{bmatrix} \quad (7)$$

where $\mathbf{u}_{S_i}$ is the carbon input from litter pools ($L_j$) to a mineral soil carbon pool ($S_i$; see Extended Data Fig. 3 for corresponding carbon flows for each mineral soil carbon pool) and can be expressed as $\sum_{L_j}(\mathbf{a}_{S_i,L_j}\mathbf{k}_{L_j}\xi \mathbf{x}_{L_j})$. It is noted that all the elements with bold font indicate vectors of the corresponding variables or parameters for the 20 soil layers. All the multiplications shown in equation (7) are element-wise operations.

The carbon input for the litter carbon pools (that is, net primary productivity) and environmental forcings (for example, soil temperature and moisture) are from 20 years of monthly model outputs (Supplementary Table 8) by CLM5 at the steady state using a preindustrial forcing (that is, I1850Clm50Bgc) at 0.5° resolution. We used the 20-year annual mean values of different components in equation (3) to calculate the total SOC stock at steady state.

The relationships between microbial CUE and SOC storage generated by the microbial model depend on the choices of model parameter values. When we applied the original parameter values used by ref. 7 in the model simulation, the microbial model generated a negative relationship between CUE and SOC. However, when we change the parameter values for turnover time of enzyme decay and microbial mortality, the same microbial model can also generate null or positive CUE–SOC relationships (Extended Data Fig. 4).

## Process-guided deep learning and data-driven modelling

The PRODA approach integrates big data with Bayesian data assimilation and deep learning to optimize soil carbon cycle simulation with process-based models[34] (Extended Data Fig. 2). We used the PRODA approach to optimize the microbial model at the global scale. Data assimilation was first applied at each SOC profile to estimate parameter values that best fit the observations. The estimated parameter values after the site-level data assimilation were further generalized to the global scale by a neural network model. The global parameter maps predicted by the neural network model were then used in the process-based model to simulate SOC storage and retrieve the spatial patterns of related model components over the globe.

We conducted Bayesian data assimilation for each of the 57,267 SOC profiles with the MCMC method to estimate the parameter values of the microbial model that best fit the model simulations with SOC observations. One empirical constraint applied to data assimilation is that simulated MIC at steady state should be no more than 10% of the SOC storage over the entire soil profile[65]. Because the soil profile data collected from field measurement of SOC includes all components of the organic matter (for example, microbial biomass carbon), we used the sum of modelled carbon in the four mineral soil pools to be compared with soil profile data from the WoSIS database.

We applied an adaptive Metropolis algorithm[66] to generate the posterior distributions of a total of 23 parameters related to the seven model components with two phases of simulations (that is, a test run and a formal run). We first conducted a test run assuming uniform distributions for each of the 23 preselected parameters as the proposal distributions (that is, prior distributions). The prior ranges of the uniform distributions for each parameter are shown in Supplementary Table 6. The proposal distributions continuously generated a set of parameter values for the microbial model to simulate SOC storage. We then judged whether the proposed parameter values should be accepted or not by comparing their model simulation results with SOC observations. In the formal run, we used the accepted sets of parameter values obtained in the test run as the proposal distributions and assumed that these 23 parameters are multivariate Gaussian distributed. We proposed new sets of parameter values and judged them to be accepted or not following the same rule in the test run. Unlike the test run, the proposal distributions in the formal run were continuously adjusted according to the newly accepted sets of parameters.

We set 20,000 iterations for the test run and 50,000 iterations for the formal run. Eventually, we controlled the acceptance ratio (that is, the ratio of accepted sets of parameters out of the total number of iterations) of the formal run between 10% and 50%. We set the burn-in coefficient as 50%, where the first half of the accepted parameter values in the formal run was discarded, and the second half was used to generate the posterior distributions of the parameters. We calculated the mean values of the posterior distributions of the parameters as the final point estimates. We ran three independent series of MCMC for each SOC profile and calculated the G–R statistic to test the convergence of data-assimilation results. The mean G–R values of the 23 parameters were further calculated as the holistic performance of MCMC for each SOC profile. The mathematical foundations of Bayesian data assimilation and technical details of the MCMC method are documented in ref. 33.

It should be noted that the data assimilation was conducted under the assumption that SOC profiles are at steady state (that is, $\frac{d\mathbf{X}(t)}{dt} = 0$). This assumption makes data assimilation computationally more feasible than that under non-steady states (see the non-steady-state data assimilation[67,68]). Although soil carbon stocks in some ecosystems (for example, agricultural soils) may not be at the steady state because of the concurrent climate change and human activities, previous research has shown that such a disequilibrium component of the transient carbon cycle dynamics, especially in SOC pools, is minor compared with the amount of SOC storage that was developed over thousands of years[69].

We trained a fully connected multilayer neural network to predict the site-level parameter values estimated from data assimilation with a suite of 60 environmental variables (Supplementary Table 9). To achieve better training effectiveness, we first normalized all the environmental variables and parameters to the interval of [0, 1] according to their maximum and minimum values. We then conducted a set

of pre-experiments to determine the best configuration setting of the neural network. The neural network used in the final training consisted of four hidden layers. The node numbers for each hidden layer were 256, 512, 512 and 256, respectively. We used a rectified linear unit as the activation function and a gradient descent optimization algorithm (adadelta) as the optimizer. The loss function was designed as the multiplication of L1 (that is, ratio loss: $\mathrm{RL} = \frac{\sum_{i=1}^{N} \left| \frac{\mathrm{para}_{i,\mathrm{true}} - \mathrm{para}_{i,\mathrm{pred}}}{\mathrm{para}_{i,\mathrm{true}}} \right|}{N}$) and L2 (that is, mean squared error: $\mathrm{MSE} = \frac{\sum_{i=1}^{N}(\mathrm{para}_{i,\mathrm{true}} - \mathrm{para}_{i,\mathrm{pred}})^2}{N}$) errors, where $\mathrm{para}_{i,true}$ is the $i$th parameter value optimized in the site-level data assimilation, $\mathrm{para}_{i,pred}$ is the $i$th parameter predicted by the neural network, and $N$ is the total number of parameters of the microbial model to be predicted by the neural network ($N$ = training size × 23). We decided to use this composite L1 × L2 loss function because training with either L1 or L2 loss alone did not yield sufficient prediction accuracy. The batch size for each iteration of optimization was 32. We set a maximum of 6,000 epochs to train the neural network and selected the model with the lowest validation loss as the final training result. To avoid overfitting in training the neural network, we set a dropping out ratio of 20% for each of the hidden layers.

Moreover, we designed a 200-time bootstrapping to estimate the prediction uncertainty of the neural network for the microbial model. The whole original database (that is, 57,267 sets of optimized parameter values from the data assimilation) was sampled with replacement for 200 times and was used to train and validate the neural network. For each bootstrapping, 90% of the data were used as the training data and the remaining 10% were used for validation. The predicted parameter values after neural network training were then applied to the microbial model to simulate the SOC stock and its underlying processes for each soil profile. We used the remaining SOC profiles that were not sampled for training and validation to test the performance of the optimized microbial model. We used the coefficient of efficiency (equation (2)) to calculate the explained variation in observed SOC by the microbial model.

### Global maps of SOC and underlying model components

We used the best-guess model, which was calibrated with all available optimized parameter values (that is, 57,267 sets of site-level data-assimilation results) to make the final prediction of global SOC storage and its related model components. Global maps of parameters that were predicted by the trained neural network using the gridded environmental variables were applied to the microbial model to generate global maps of SOC storage and its related components. Meanwhile, we calculated the $2\sigma$ confidence interval in the 200-time bootstrapping as the uncertainty for the microbial model (Extended Data Fig. 7). It is worthwhile to point out that because we took the point estimates (that is, the mean values of the posterior distributions after site-level data assimilation) to train the neural network, the uncertainties of parameters and SOC simulation did not propagate from the site-level data assimilation but only reflected uncertainties generated by the neural network.

We retrieved the system-level CUE, plant carbon inputs, allocation of input carbon to different soil layers, non-microbial carbon transfer, substrate decomposability, environmental modifications and vertical transport from the optimized parameters of the microbial model (Supplementary Table 6) via the PRODA approach. It is noted that all the seven model components referred to in this study are ensembles of processes that were represented by different parameters in the process-based model. For example, the environmental modifier is an ensemble of all parameters that reflect how the physical environmental conditions (for example, temperature and soil moisture) will influence the apparent decomposition rate of SOC. These ensembles of processes from the same category are a useful way to describe the holistic system dynamics of SOC over the globe.

We calculated the system-level CUE of the microbial model according to its definition using equation (1). Specifically, assimilating both litter organic carbon and DOC of the mineral soils contributes to the production of microbial biomass (Extended Data Fig. 3). In these processes, the total substrate carbon utilized in microbial metabolism is:

$$\mathrm{Substrate\ uptake} = \sum_i \sum_z [x_{L_i,z} k_{L_i,z}(1 - a_{\mathrm{nonMIC},L_i})\xi_z \Delta z] \\ + \sum_z \left( v_{\mathrm{max,assim}}\xi_z x_{\mathrm{MIC},z} \frac{x_{\mathrm{DOC},z}}{K_{\mathrm{m,assim}}\xi_z + x_{\mathrm{DOC},z}}\Delta z \right) \quad (8)$$

where $\Delta z$ is the thickness of the $z$th soil layer and $i$ is from 1 to 3, representing metabolic, cellulose and lignin litter pools. The $a_{\mathrm{nonMIC},L_i}$ from the **A** matrix (equation (6)) is $a_{\mathrm{DOC,ML}}$ for metabolic litter, $a_{\mathrm{DOC,CL}}$ for cellulose litter and $a_{\mathrm{mSOC,LL}}$ for lignin litter. Correspondingly, the total microbial biomass production is:

$$\mathrm{Biomass\ production} \\ = \sum_i \sum_z [\eta_{L_i} x_{L_i,z} k_{L_i,z}(1 - a_{\mathrm{nonMIC},L_i})\xi_z \Delta z] \\ + \sum_z \left( \eta_{\mathrm{DOC}} v_{\mathrm{max,assim}}\xi_z x_{\mathrm{DOC},z} \frac{x_{\mathrm{DOC},z}}{K_{\mathrm{m,assim}}\xi_z + x_{\mathrm{DOC},z}}\Delta z \right) \quad (9)$$

Substituting equations (8) and (9) into equation (1) gives the system-level CUE:

$$\mathrm{CUE}_{\mathrm{system}} = \frac{\begin{aligned}&\sum_i \sum_z [\eta_{L_i} x_{L_i,z} k_{L_i,z}(1 - a_{\mathrm{nonMIC},L_i})\xi_z \Delta z] \\ &+ \sum_z \left( \eta_{\mathrm{DOC}} v_{\mathrm{max,assim}}\xi_z x_{\mathrm{DOC},z} \frac{x_{\mathrm{DOC},z}}{K_{\mathrm{m,assim}}\xi_z + x_{\mathrm{DOC},z}}\Delta z \right)\end{aligned}}{\begin{aligned}&\sum_i \sum_z [x_{L_i,z} k_{L_i,z}(1 - a_{\mathrm{nonMIC},L_i})\xi_z \Delta z] \\ &+ \sum_z \left( v_{\mathrm{max,assim}}\xi_z x_{\mathrm{MIC},z} \frac{x_{\mathrm{DOC},z}}{K_{\mathrm{m,assim}}\xi_z + x_{\mathrm{DOC},z}}\Delta z \right)\end{aligned}} \quad (10)$$

The $\mathrm{CUE}_{\mathrm{system}}$ combines all the microbial CUEs for both litter organic carbon and DOC into one single metric. The variation of $\mathrm{CUE}_{\mathrm{system}}$ is mainly controlled by the CUE of the mineral soils (that is, $\eta_{\mathrm{DOC}}$ in Supplementary Table 6). We found a strong correlation between the system-level CUE and $\eta_{\mathrm{DOC}}$ retrieved from the 57,267 SOC profiles via data assimilation (Pearson correlation coefficient = 0.98, d.f. = 56,270, $P < 0.001$; Supplementary Fig. 1).

We also synthesized carbon transfer fractions that indicate the organic carbon transfer from one pool to another without the involvement of microbial processes but also present spatial variability (Supplementary Table 6) into a system-level value. The spatial patterns of non-microbial carbon transfer fractions may indicate the mineral–organic interactions in the soil. We defined the system-level, non-microbial carbon transfer fraction as the sum of transfer coefficients other than CUE in the **A** matrix (equation (6)) weighted by the carbon fluxes over all the related transfer pathways in the soil system:

$$T_{\mathrm{system}} = \sum_{ij} a_{ij} \frac{\sum_z x_{j,z} k_{j,z}\xi_z \Delta z}{\sum_j \sum_z x_{j,z} k_{j,z}\xi_z \Delta z} \quad (11)$$

The baseline decomposition rate (unit $\mathrm{yr}^{-1}$) expresses the rate of organic carbon decomposition other than microbial uptake at optimal soil temperature and water conditions. We calculated the system-level baseline decomposition rate ($K_{\mathrm{system}}$, unit $\mathrm{yr}^{-1}$) by weighting the baseline decomposition rate of different litter (that is, CWD, metabolic litter, cellulose litter and lignin litter, denoted by $L_i$) and mineral soil (that is, MIC, ENZ and mSOC, denoted by $S_j$) organic carbon pools by their carbon pool sizes:

$$K_{\mathrm{system}} = \sum_{L_i} k_{L_i} \frac{x_{L_i}}{\sum x_{L_i}} + \sum_{S_j} k_{S_j} \frac{x_{S_j}}{\sum x_{S_j}} \quad (12)$$

Similarly, we weighted the vertical movement rate (yr$^{-1}$) and environmental modifiers (unitless) at different soil depths by their carbon pool sizes:

$$V_{\text{system}} = \sum_z \left( v_z \frac{x_z}{\sum x_z} \right) \tag{13}$$

$$\xi_{\text{system}} = \sum_z \left( \xi_{\text{T},z} \xi_{\text{W},z} \xi_{\text{D},z} \frac{x_z}{\sum x_z} \right) \tag{14}$$

To quantify how effectively the input allocation process distributes litterfall and root exudation to different soil depths, according to an asymptotic equation that describes the distribution of root biomass at different soil depths[23], we calculated the total fraction of carbon input allocated to soils that are deeper than 5 cm as the system-level index for plant carbon input allocation:

$$B_{\text{system}} = \left[ \frac{\sum_z \exp\left[ \frac{\ln(1-Y_z)}{D_z} \right]}{n} \right]^5 \tag{15}$$

where $Y_z$ is the cumulative fraction of input carbon at soil depth of $D_z$ and $n$ is the number of soil layers. A larger system-level input allocation index indicates that more carbon from litterfall and root exudation will be allocated to deeper soils. Finally, we took the simulated total litterfall (equivalent to the net primary productivity) in CLM5 as the plant carbon input.

## Sensitivity analyses

We carried out two sensitivity analyses to evaluate the relative importance of different model components in determining global SOC storage and its spatial distribution (Fig. 4).

The first sensitivity analysis quantifies how the spatial variation of different components affected global SOC storage and its spatial distribution. For each component, we harmonized its corresponding parameters (see Supplementary Table 6 for classification) as global constants while allowing the parameters of other components to vary. The global constant values were derived from the mean values of optimized parameter values in the site-level data assimilation. We then evaluated how the spatially constant components influence the global simulation of SOC stock (that is, total absolute deviation from PRODA estimates) and its spatial variation (that is, deviation of explained variation in observations defined by equation (2)) against their best estimates (that is, when all spatial variation was accounted for). Because the plant carbon input was an input variable rather than a model parameter in this study, we perturbed its spatial variation by randomly adding or subtracting its two standard deviations to or from its mean value in the microbial model simulations.

The second sensitivity analysis evaluates how the global SOC storage responds to changes in each of the model components. Specifically, we proportionally changed the parameters belonging to one component (Supplementary Table 6) from their optimal values (that is, the value predicted by PRODA approach) and then calculated the consequent proportional changes of system-level metrics (as shown in equations (10)–(15)) and global SOC storage. When investigating the effect of one component, we set the parameters affecting the other components at their optimal value. Therefore, the response curve of global SOC storage to the changes of system-level values of different components results from only the changes of the underlying parameters of the investigated components. Moreover, we built statistical models describing the relations between the system-level environmental modifier ($\xi_{\text{system}}$) and MAT (that is, $\log(\xi_{\text{system}}) \approx \alpha \text{MAT} + \beta$, regression $\alpha = 0.056$, $\beta = -2.35$, $R^2 = 0.88$) and mean annual precipitation (MAP)

(that is, $\xi_{\text{system}} \approx \alpha \text{MAP} + \beta$, regression $\alpha = 0.00017$, $\beta = 0.084$, $R^2 = 0.52$) across the globe. These regression equations allow associating the changes of the system-level environmental modifier with changes in global temperature and precipitation to build up specific context in discussion.

It is noted that because the parameters used to describe microbial assimilation processes (that is, $v_{\text{max,assim}}$ and $K_{\text{m,assim}}$ in Supplementary Table 6) are directly linked to changes of microbial processes and thus for the calculation of system-level CUE in this study, we also merged the spatial variation of the above-mentioned parameters in the first sensitivity analysis and proportionally changed their values in the second sensitivity analysis to investigate the relative importance of system-level CUE to global SOC storage.

## Influence of environmental variables on ecological processes

We conducted a permutational analysis[70] to evaluate the influence of environmental variables on the six model components other than the plant carbon input. By permuting the original environmental variable information with random values and thus breaking the relationships between the environmental variable and the model component predicted by the trained neural network, we interpreted the consequently decreased prediction performance of the neural network as an indication to the influence of that environmental variable to the component. Specifically, we first classified all the 60 environmental variables (Supplementary Table 9) used in the neural network model into five categories: soil structural properties, soil chemical properties, climatic variables, vegetation variables and geographical variables. We then used the best-guess neural network that was calibrated with all available data to do the permutation test. For each category of environmental variables, we permutated their original information with uniformly distributed random values for 1,000 times. After each permutation, we used the best-guess neural network to predict parameter values from both original environmental information (that is, without permutation) and permutated values. We used the mean squared error (that is, $\text{MSE} = \frac{\sum_{i=1}^N [\sum_{j \in M} (\text{para}_{i,\text{NN}}^j - \text{para}_{i,\text{DA}}^j)^2]}{N}$, where $j$ refers to model components, $i$ to soil profiles, $\text{para}_{\text{NN}}$ to neural network-predicted parameter values and $\text{para}_{\text{DA}}$ to parameter values estimated from data assimilation) to account for the prediction deviation from site-level ($i$) optimized parameter values. Parameters belonging to the same components ($j \in M$) were grouped together in the calculation. The permutation importance (PI) of environmental variables in category $k$ ($\text{cate}_k$) to model component $M$ was then expressed as:

$$\text{PI}_{\text{cate}_k \text{ to } M} = \frac{\sum_{i=1}^N \left[ \sum_{j \in M} (\text{para}_{i,\text{permuNN}}^j - \text{para}_{i,\text{DA}}^j)^2 \right]}{\sum_{i=1}^N \left[ \sum_{j \in M} (\text{para}_{i,\text{nopermuNN}}^j - \text{para}_{i,\text{DA}}^j)^2 \right]} \tag{16}$$

The PI values represent the increase of inaccuracy in the neural network prediction caused by the absent information of environmental variables in category $k$. Thus, a larger value of permutation importance indicates a greater influence of an environmental variable to the prediction of parameters.

## Data availability

All the data that support the findings of this study are available at https://doi.org/10.5281/zenodo.7676632. The raw data for the SOC profiles from the WoSIS database can be accessed at https://www.isric.org/explore/wosis.

## Code availability

All the code used in the analyses presented in this paper can be accessed at https://github.com/phxtao/PRODA_MIC.

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

**Acknowledgements** We thank H. Yang, M. Schrumpf, T. Wutzler, R. Zheng and H. Ma for their comments and suggestions on this study. This work was supported by the National Natural Science Foundation of China (42125503) and the National Key Research and Development Program of China (2020YFA0608000, 2020YFA0607900 and 2021YFC3101600). F.T. was financially supported by China Scholarship Council during his visit at Food and Agricultural Organization of the United Nations (201906210489) and the Max-Planck Institute for Biogeochemistry (202006210289). The contributions of Y.L. were supported through US National Science Foundation DEB 1655499 and 2242034, subcontract CW39470 from Oak Ridge National Laboratory (ORNL) to Cornell University, DOE De-SC0023514, and the USDA National Institute of Food and Agriculture. S.M. has received funding from the ERC under the European Union's H2020 Research and Innovation Programme (101001608). The contributions of U.M. were supported through a US Department of Energy grant to the Sandia National Laboratories, which is a multi-mission laboratory managed and operated by National Technology and Engineering Solutions of Sandia, LLC, a wholly owned subsidiary of Honeywell International, Inc., for the US Department of Energy's National Nuclear Security Administration under contract DE-NA-0003525. We thank the WoSIS database (https://www.isric.org/explore/wosis) for providing the publicly available global-scale SOC database used in this study.

**Author contributions** Y.L., F.T. and X.H. conceived the study. F.T., Y.L., Y.H. and B.A.H. designed the paper and S.M., S.D.F. and M.W.I.S. contributed to the development of the paper structure. F.T. collated CUE and SOC data from peer-reviewed literature and conducted the meta-analysis. F.T. collected the SOC profiles from WoSIS database and U.M. and G.H. contributed additional SOC profiles in permafrost regions. F.T., X.L., C.L. and Y.-S.L. prepared input data for driving the microbial model simulation. F.T., with input from Y.L., J.L., Y.H., Y.-P.W., S.M., B.Z.H., N.C. and M.R., constructed the microbial model. F.T. conducted the data assimilation and deep learning in the PRODA approach. X.H., T.D.H. and N.C. provided suggestions on deep learning. F.T., Y.L., Y.H. and B.A.H. wrote the first drafts. All authors contributed to major revisions of the paper and approved the final version.

**Competing interests** The authors declare no competing interests.

**Additional information**
**Correspondence and requests for materials** should be addressed to Xiaomeng Huang or Yiqi Luo.

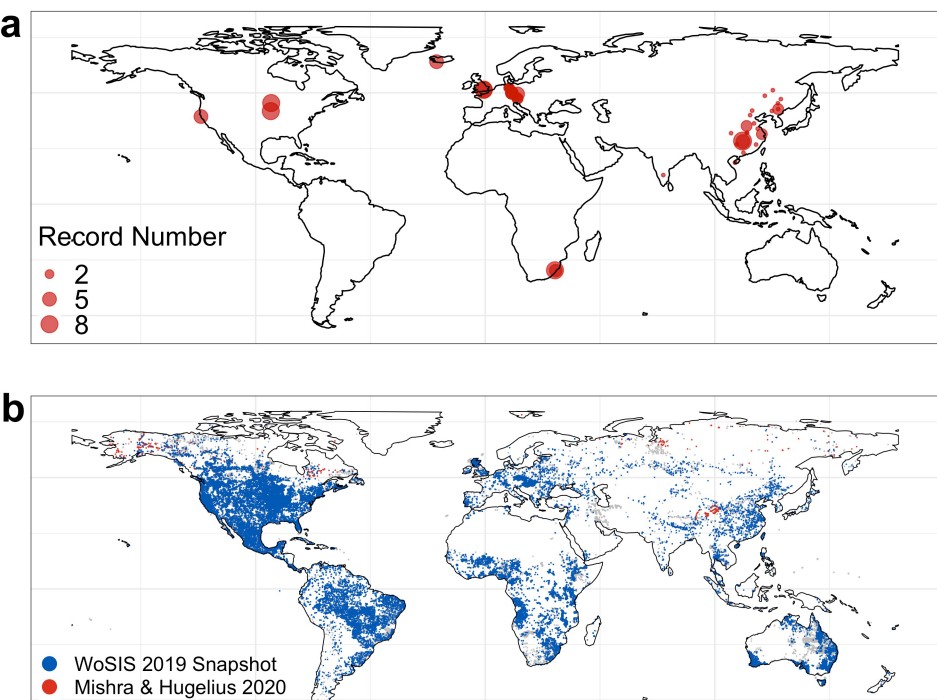

**Extended Data Fig. 1 | Spatial distributions of datasets.** Spatial distributions of datasets used in meta-analysis (a) and vertical SOC profiles used in the PRODA approach (b).

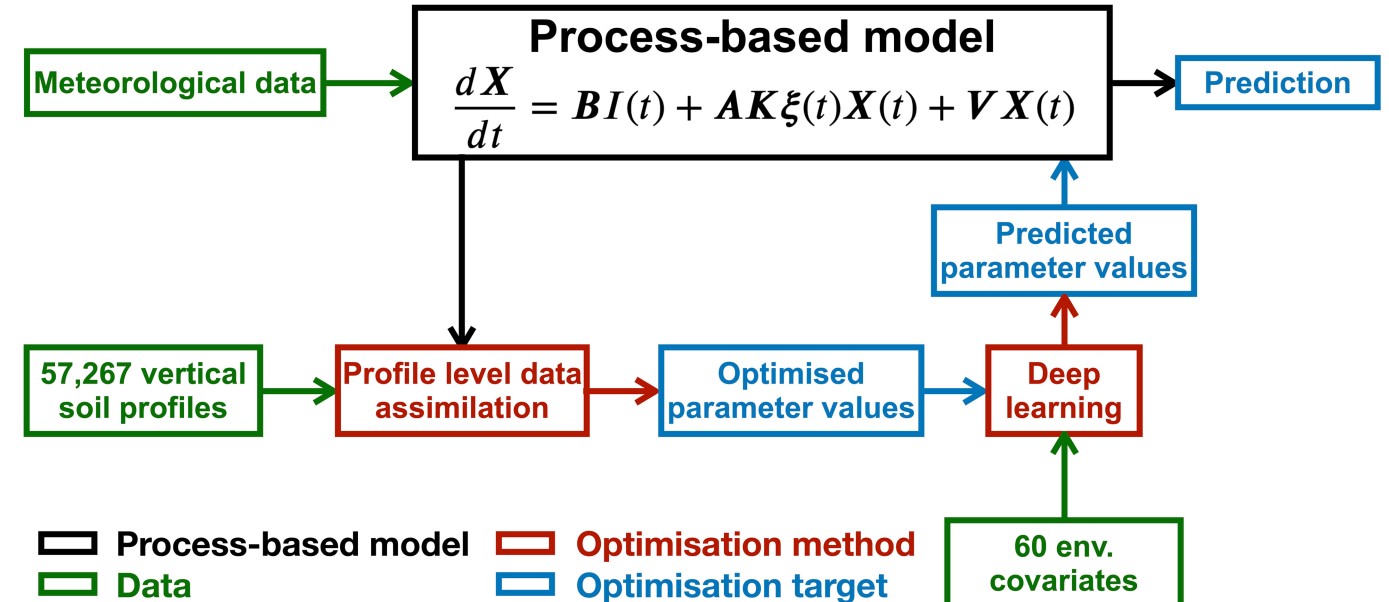

**Extended Data Fig. 2 | Workflow of the PROcess-guided deep learning and DAta-driven modelling (PRODA) approach.** We first applied the Bayesian data assimilation to fuse observational data at each soil profile with the microbial model. Parameters that represent different components in modelling soil carbon cycle were estimated through the Markov Chain Monte Carlo (MCMC) method. A deep learning model then predicted the optimised parameter values (i.e., the mean value of the posterior distribution after MCMC) by a set of 60 environmental variables. The predicted parameter values by the deep learning model were further applied in the microbial model to calculate SOC storage and related components.

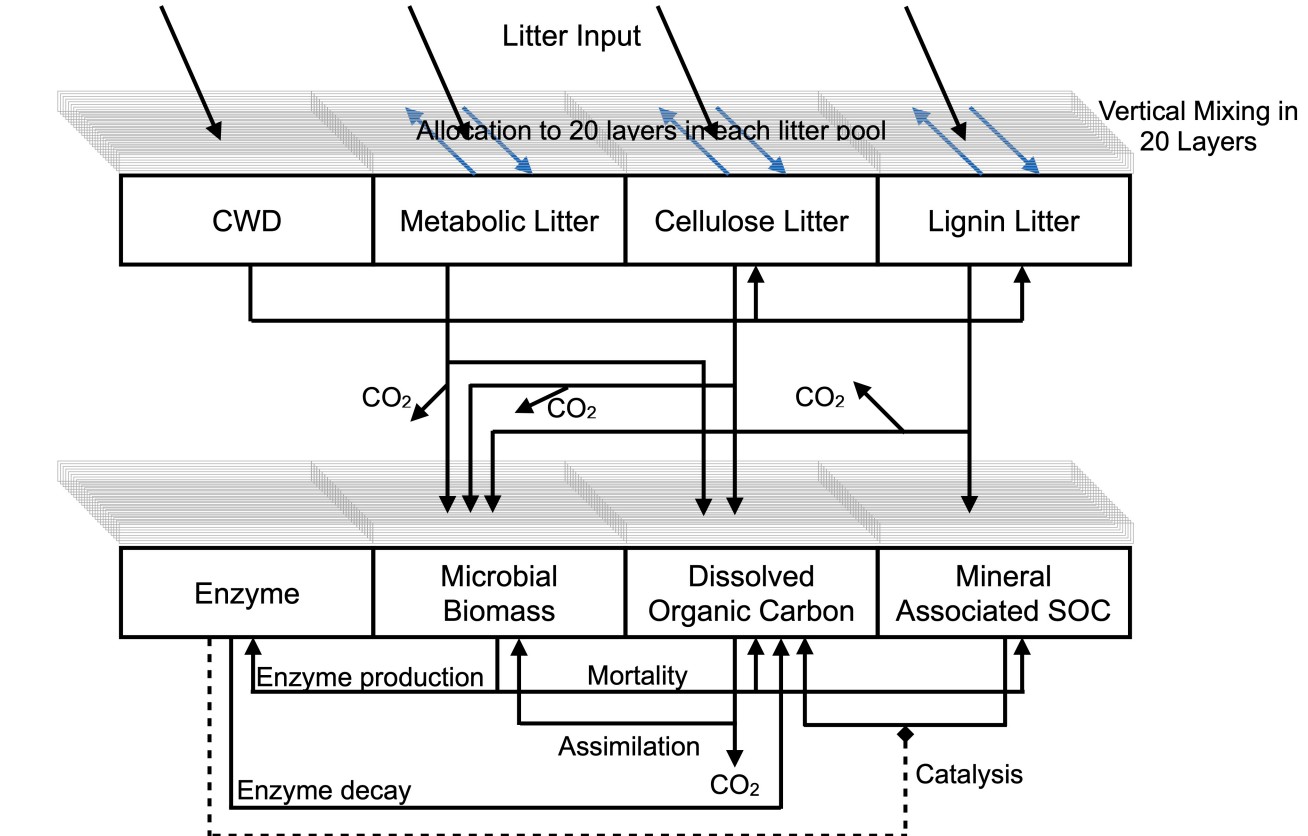

**Extended Data Fig. 3 | Structure of the microbial-process explicit model used in this study.** See Methods for detailed descriptions of the model.

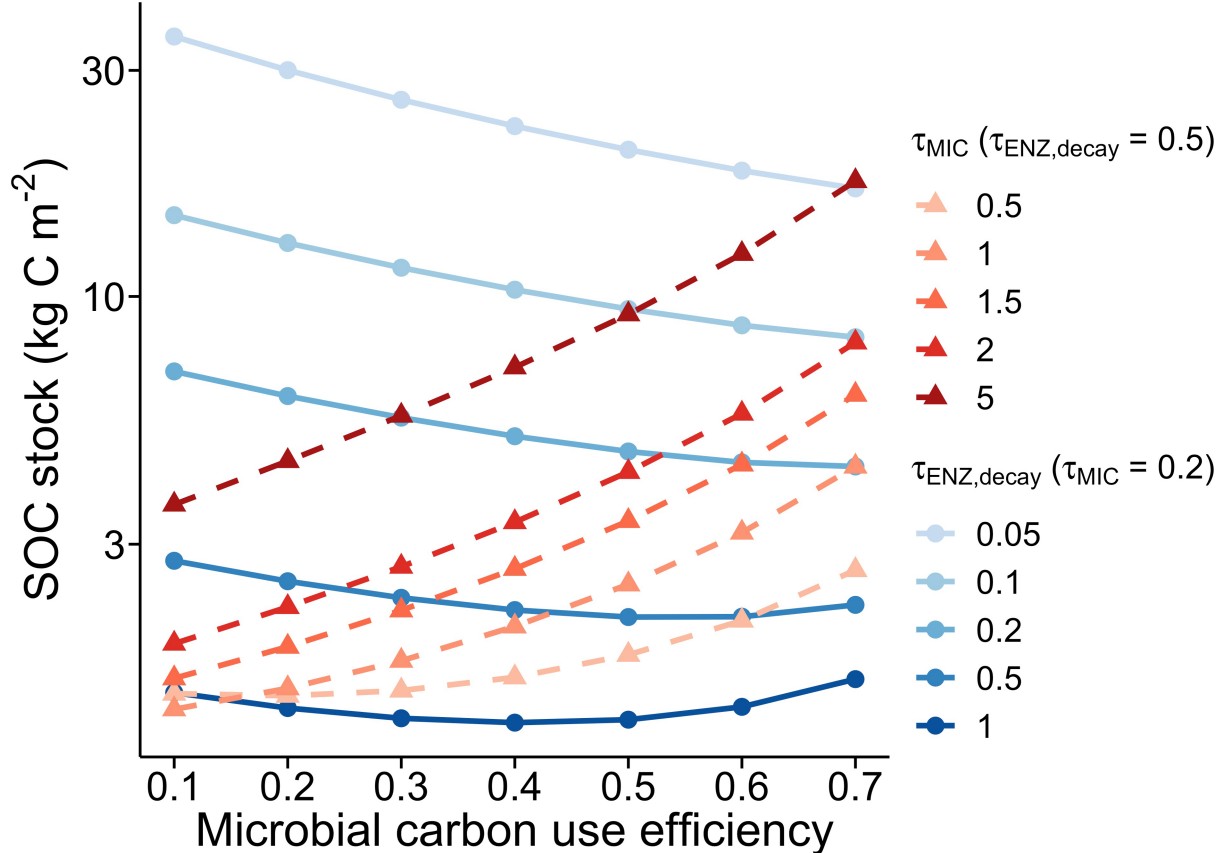

**Extended Data Fig. 4 | Varying CUE-SOC relationships at the steady state under different parameter values in the microbial model.** $\tau_{ENZ,decay}$ is the turnover time for enzyme decay (unit: yr). $\tau_{MIC}$ is the turnover time for microbial mortality (unit: yr).

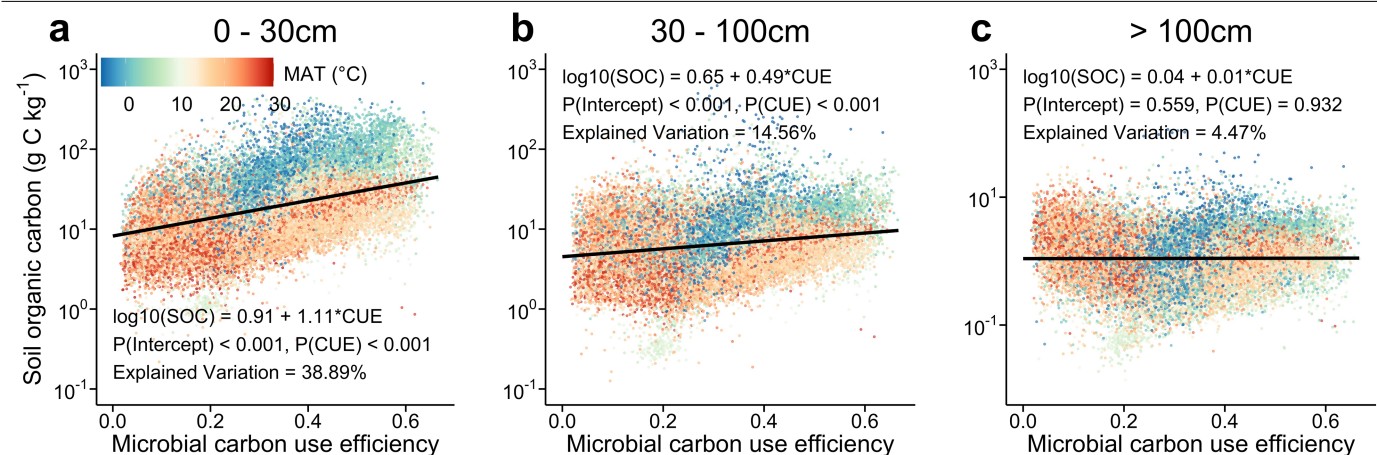

**Extended Data Fig. 5 | CUE-SOC relationships at different soil depths.**
Results are from assimilating all the available soil profiles (n = 57,267) to the microbial model. Declining explanatory power of CUE to the variation in SOC with soil depths possibly indicates more interactions of organic matter with mineral particles at depth.

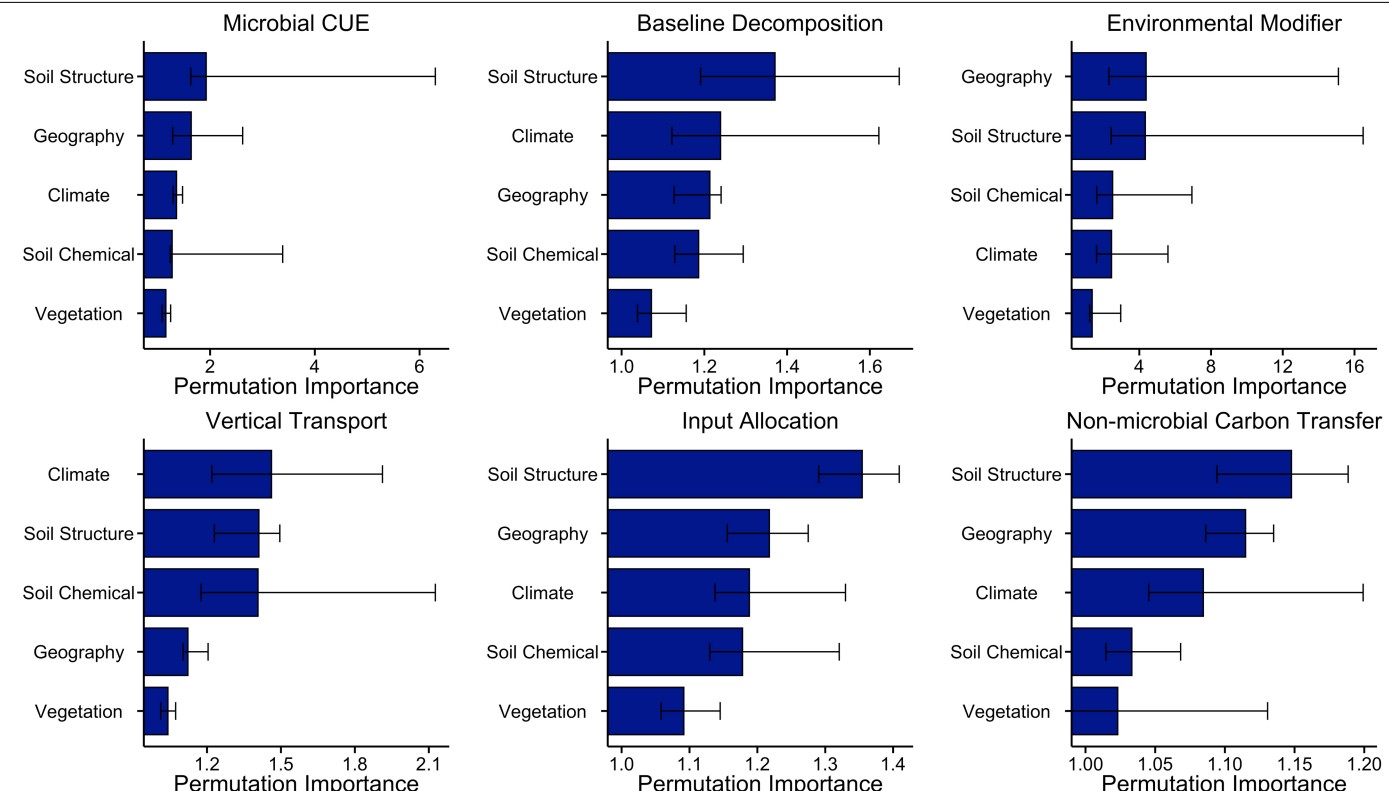

**Extended Data Fig. 6 | Influences of environmental variables on different components investigated in this study.** Results are the median values from 1000-time permutations to the best-guess model. Error bar indicates the two-sigma confidence interval. Results for the plant carbon input is not available because we directly used the simulation results by CLM5 as the carbon input instead of predicting it by the neural network.

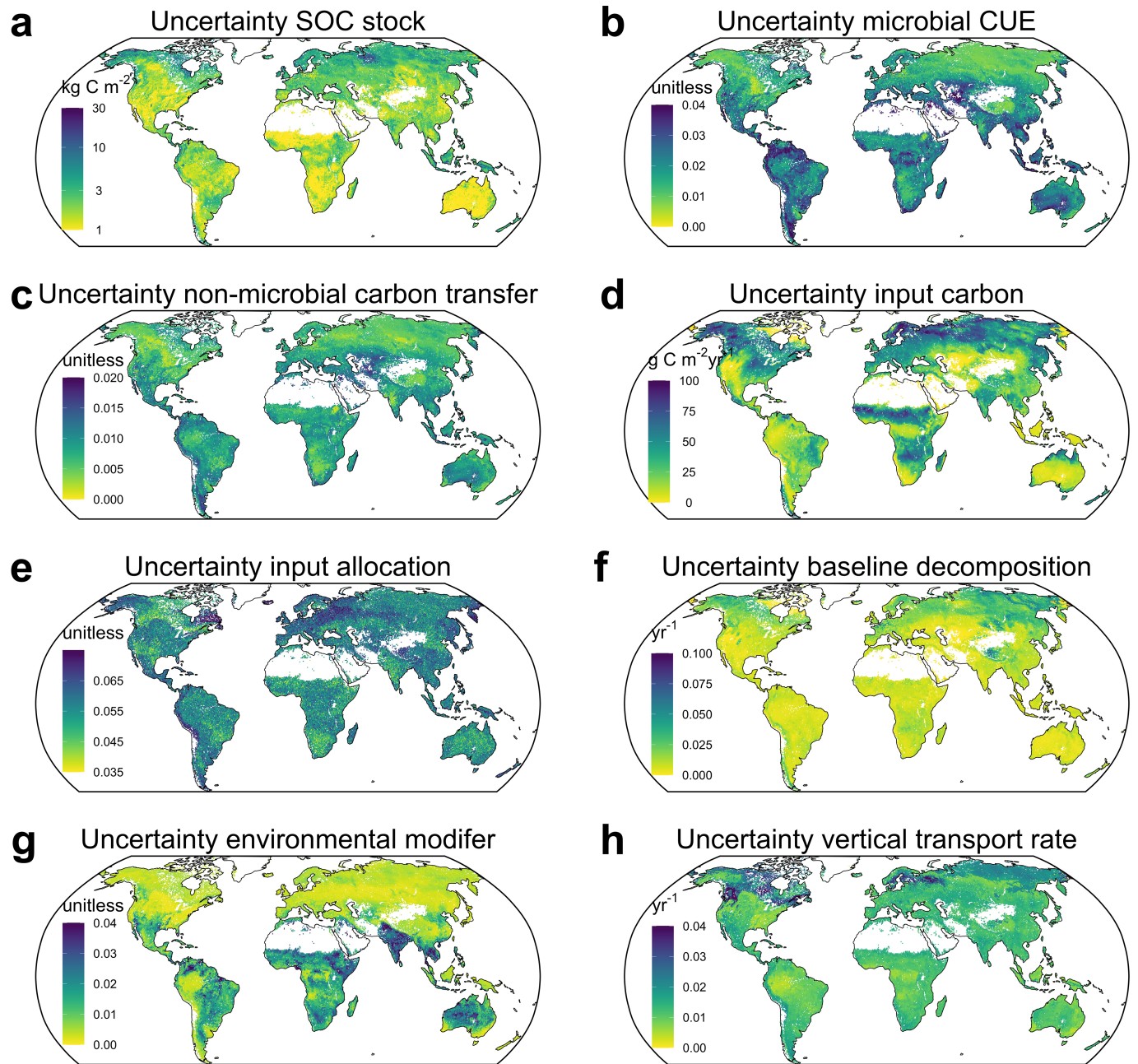

**Extended Data Fig. 7 | Uncertainties of retrieved global SOC storage and related model components.** The uncertainty maps (except carbon input) showed the standard deviations in the 200-time bootstrapping. Uncertainty of carbon input derived from the CLM5 interannual simulations.

**Extended Data Table 1 | Unstandardized coefficients of CUE-SOC relationship in the mixed-effects model with meta-analysis data**

| | | Intercept | CUE | Depth | MAT | MAT*CUE |
|---|---|---|---|---|---|---|
| \multicolumn{7}{c}{$log10(SOC)\sim CUE + Depth + MAT + (1\|Study\ Source)$ <br> variance explained by mixed model: 55%, by fixed effect: 37%} | | | | | | |
| **Fixed Effects** | Estimates | 1.16 | 0.76 | -0.020 | 0.013 | NA |
| | Std. Error | 0.11 | 0.18 | 0.0034 | 0.0053 | NA |
| | t value | 10.58 | 4.14 | -5.93 | 2.49 | NA |
| | P | <0.0001 | <0.0001 | <0.0001 | 0.014 | NA |
| **Random Effects** | Standard Deviation | 0.16 | NA | NA | NA | NA |
| \multicolumn{7}{c}{$log10(SOC)\sim CUE + Depth + MAT + MAT*CUE + (1\|Study\ Source)$ <br> variance explained by mixed model: 55%, by fixed effect: 38%} | | | | | | |
| **Fixed Effects** | Estimates | 1.17 | 0.72 | -0.020 | 0.012 | 0.0042 |
| | Std. Error | 0.17 | 0.43 | 0.0034 | 0.014 | 0.039 |
| | t value | 6.78 | 1.65 | -5.90 | 0.90 | 0.11 |
| | P | <0.0001 | 0.10 | <0.0001 | 0.37 | 0.91 |
| **Random Effects** | Standard Deviation | 0.17 | NA | NA | NA | NA |
| \multicolumn{7}{c}{$log10(SOC)\sim CUE + Depth + MAT + MAT*CUE + (CUE\|Study\ Source)$ <br> variance explained by mixed model: 49%, by fixed effect: 20%} | | | | | | |
| **Fixed Effects** | Estimates | 1.20 | 0.35 | -0.020 | 0.0086 | 0.040 |
| | Std. Error | 0.15 | 0.49 | 0.0031 | 0.011 | 0.037 |
| | t value | 7.79 | 0.72 | -6.46 | 0.75 | 1.10 |
| | P | <0.0001 | 0.48 | <0.0001 | 0.46 | 0.28 |
| **Random Effects** | Standard Deviation | 0.032 | 0.77 | NA | NA | NA |
| \multicolumn{7}{c}{$SOC\sim CUE + Depth + MAT + (1\|Study\ Source)$ <br> variance explained by mixed model: 49%, by fixed effect: 20%} | | | | | | |
| **Fixed Effects** | Estimates | 17.30 | 55.49 | -0.65 | 0.13 | NA |
| | Std. Error | 7.62 | 12.60 | 0.23 | 0.37 | NA |
| | t value | 2.27 | 4.41 | -2.84 | 0.36 | NA |
| | P | 0.026 | <0.0001 | 0.0052 | 0.72 | NA |
| **Random Effects** | Standard Deviation | 12.74 | NA | NA | NA | NA |
| \multicolumn{7}{c}{$SOC\sim CUE + Depth + MAT + (CUE\|Study\ Source)$ <br> variance explained by mixed model: 49%, by fixed effect: 20%} | | | | | | |
| **Fixed Effects** | Estimates | 9.40 | 57.50 | -0.65 | 0.77 | NA |
| | Std. Error | 6.70 | 19.80 | 0.20 | 0.35 | NA |
| | t value | 1.40 | 2.91 | -3.16 | 2.19 | NA |
| | P | 0.16 | 0.012 | 0.0020 | 0.030 | NA |
| **Random Effects** | Standard Deviation | 5.00 | 55.92 | NA | NA | NA |
| \multicolumn{7}{c}{$SOC\sim CUE + Depth + MAT + MAT*CUE + (1\|Study\ Source)$ <br> variance explained by mixed model: 49%, by fixed effect: 20%} | | | | | | |
| **Fixed Effects** | Estimates | 12.67 | 69.39 | -0.65 | 0.57 | -1.38 |
| | Std. Error | 11.80 | 29.56 | 0.23 | 0.92 | 2.65 |
| | t value | 1.07 | 2.35 | -2.85 | 0.62 | -0.52 |
| | P | 0.29 | 0.020 | 0.0051 | 0.54 | 0.60 |
| **Random Effects** | Standard Deviation | 12.55 | NA | NA | NA | NA |

CUE, depth and mean annual temperature (MAT) were set as the fixed effects to both logarithmic and original SOC content. The study source was set as the random effect. We explored different structures of the mixed-effects models (i.e., random intercepts with common slopes or random intercepts with random slopes) to test the CUE-SOC relationship. The interaction between CUE and MAT was also considered. Only results of model structures that converged in regressions are presented. The total observation size $n_{sample}$=132; the random effects size $n_{study}$=16. The variance inflation factors for the main effects model were all low.

**Extended Data Table 2 | Unstandardized coefficients of CUE-SOC relationship in the mixed-effects model with data assimilation results**

| | | Intercept | CUE |
|---|---|---|---|
| $log10(SOC){\sim}CUE + (1|Climate\ Types)$ variance explained by mixed model: 37%, by fixed effect: 23% | | | |
| Fixed Effects | Estimates | 0.89 | 1.18 |
| | Std. Error | 0.055 | 0.0096 |
| | t value | 16.07 | 122.44 |
| | P | <0.0001 | <0.0001 |
| Random Effects | Standard Deviation | 0.19 | NA |
| $log10(SOC){\sim}CUE + (CUE|Climate\ Types)$ Variance explained by mixed model: 39%, by fixed effect: 23% | | | |
| Fixed Effects | Estimates | 0.91 | 1.10 |
| | Std. Error | 0.056 | 0.13 |
| | t value | 16.22 | 8.70 |
| | P | <0.0001 | <0.0001 |
| Random Effects | Standard Deviation | 0.19 | 0.43 |

CUE was set as the fixed effects to logarithmic SOC content. Climate types that soil profiles belong to were set as the random effect. We applied mixed-effects models that considered random intercepts and random slopes to test the CUE-SOC relationship. The random effects size $n_{climate}$=12. The total observation size $n_{obs}$=56,270. The observation size is different from the total soil profile size (i.e., 57,267) because the climate type information is not available for some profiles.