## [Peer Review File · Nature]

Manuscript Title: Microbial carbon use efficiency promotes global soil carbon storage

Redactions – unpublished data

Reviewer Comments & Author Rebuttals

Reviewer Reports on the Initial Version:

Referee #1 (Remarks to the Author):

The main findings in Tao et al. are exciting and the approaches used are technically advanced and appear to open new doors to enquiry. That said, as currently presented, it is hard for the reader to evaluate the veracity of how the main findings were reached. I would like to see the authors wrestle directly with issues of correlation vs. causation, that pattern does not beget process, space-for-time substitutions, and strong inference. I have the following specific comments related to these common challenges to identifying causation that, if satisfactorily dealt with, I hope will help to provide the necessary rigor to support the exciting findings that are presented. Please note that, if I were the Editor, #1 would sink the current paper if not addressed fully; and I would regard #2 as important for ensuring statistical rigor in the work published in my journal.

1. Strong inference and pattern does not beget process. I like the competing hypotheses shown in Fig. 1a, where the possibilities are that higher CUE favors accumulation of SOC vs. loss. These mutually exclusive hypotheses (at least as presented) lend themselves well to a strong inference test. However, the CLM5 belowground carbon cycle module is structured around the first hypothesis being true. That is, CUE determines the proportion of plant C inputs into the soil that form SOC vs. are respired as CO₂. As such the structure of the model is pre-determined to support the first hypothesis and falsify the second. This appears to add strong circularity to the results, meaning the work falls far short of a strong inference test of competing hypotheses and appears to fall into the trap of assigning process to pattern. It would seem that to reliably test between these two hypotheses, you would need a second-order process in CLM5 where the amount of microbial biomass regulates the rate at which SOC is formed and decomposed. If I have misinterpreted what was done, I apologize to the authors but would still ask that these issues be directly addressed in the main paper, so that the same concerns do not surface once the paper is accepted (because these concerns about circularity, if not resolved, suggest a fundamental flaw in the study design that questions the validity of the hypothesis testing).

2. Correlation vs. causation. The authors do spend most of their time - in the Results - on the correlation side of the line, which is appreciated and appropriate given the observational nature of the work. However, the statistical analysis of the meta-analysis data are woefully shallow and inadequate given your recognition that CUE and SOC are outcomes of multiple causative predictors. Put simply, you recognize correlation and mention it, but do not perform the necessary data analytical steps to build confidence in your analysis of the relationships presented. The univariate correlations in Figs. 2a and 2b should be replaced with plots of the partial (or conditional) coefficients from the mixed effects models, so that the relationships shown are corrected for the influence of other predictors (e.g., both depth and MAT for a). Further, you need to present results from analyses showing that your predictors are not correlated, and at least evaluate two-way

interactions among the predictors. Currently, you have a single model shown in Extended Data Table 2 of unstandardized coefficients (without a SE), a cumulative R² (fixed + random), and no recognition that the predictors could be correlated and hence the possibility that the coefficients you present are spurious. Further, nor do you justify the use of both random slopes and random intercepts. Please revise this analysis to build credibility in your findings. When I analyzed your data given in Extended Data Table 1, I will note that your general findings are supported and your predictors appear uncorrelated in the various models I ran. For example, for SOC, using random intercepts and common slopes, the beta coefficients reveal a strong negative MAT effect on SOC, a weaker but still positive CUE effect on SOC, and then a strong negative MAT by CUE interaction. The variance inflation factors for the main effects model were all low and for the model with the pairwise interactions, fixed effect variance explained was 17.6% with a further ~60% of the variance explained by the “study id” random effect. As such, use a regular linear model (without random effects), the results were very different, revealing strong within-study associations in these data that should be mentioned. My take home is that the results from your meta-analysis will be bolstered by doing your due diligence in terms of the mixed-effects modeling. However, as currently presented, this part of your analysis falls far below publication standard.

3. The remainder of my concerns are less consequential but I think important to address nonetheless. They are as follows, (a) Space-for-time. Your work uses spatial, observational data to build knowledge but in numerous instances you translate the significance for understanding change in time. Acknowledgement that you are making inferential jumps here by using space-for-time substitutions in terms of inferring process would be appreciated. (b) The main paper comes across as selectively choosing results that “agree” with the findings you present. For example, lines 202-205 talk about data that are consistent with a positive CUE-SOC relationship and you also cite evidence where CUE and MAT are negatively correlated. Please acknowledge and include some discussion to resolve other observational syntheses combined with modeling work that find the opposite (e.g., Ye et al. 2020 in *Global Biogeochemical Cycles*, <https://agupubs.onlinelibrary.wiley.com/doi/10.1029/2019GB006507>). It would be important for the reader to understand how your work therefore fits into the wider picture painted by the literature on regional to global observational data-model syntheses related to this topic. (c) You refer repeatedly (e.g., lines 176-177) to the five mechanisms in CLM5 that influence SOC turnover and the fact that CUE emerges as the one to which spatial patterns are most sensitive and best predicted by. However, looking into those mechanisms, they are less directly coupled to SOC stocks than CUE because they control how much is going into litter pools or how fast SOC decomposes (which itself is dependent on the size of the SOC pool), but they are not the process/ parameter that actually dictates allocation of inputs to SOC. As such, please address in the main paper the extent to which this close coupling of CUE to SOC formation rates essentially predetermines your finding that it is the mechanism most closely matched to storage in CLM5. (d) Coming back to right where I started, with regards to broader questions around how we do science and what we can learn (e.g., correlation vs. causation), I think it would be very helpful to explain in the main paper (i.e., in the paragraph starting line 110) the philosophy that underpins the PRODA tool. Although many new data science tools are exciting, they can also easily be misused, leading to spurious findings that are only revealed once the philosophy underpinning the approach is dug into. Given that most of the readership will be unfamiliar with the tools applied, and hence what they can do in terms of identifying mechanism (as opposed to, say, being optimized for outcome prediction; think of such

things as IC-based model selection), I think it would be important to spend a handful of sentences explaining the philosophy of the approach.

Referee #2 (Remarks to the Author):

This study proposes to use global and publicly available datasets complemented by a meta-analysis to draw conclusions on the potential drivers of global soil organic carbon (SOC) storage. The statistical modelling of SOC storage was made by a “PROcess-guided deep learning and DATA-driven modelling”, which, to summarize, aim to use a deep learning model to estimate the parameters of a process-based biochemical model.

Overall I found this study of potential interest, but I also raise substantial concerns on specific aspects of the mapping, validation, and interpretation process. I am however positive that the authors could address these concerns by a revision and additional analyses, and that the revised manuscript could be a useful contribution.

Cross-validation

The cross-validation should be used to estimate the validation statistics (as is done), but for prediction and uncertainty quantification a model fitted with all the data should be used. In this paper, the authors used 10 models, resulting from fitting 10 models on the 10 cross-validation folds, and use the mean value from the 10 model predictions as their final prediction. They also use the standard deviation of the 10 predictions as an estimate of prediction uncertainty.

- **Prediction** The final model for mapping should be calibrated with all the available observations. This is the “best guess model”, calibrated with all data available. Map accuracy is estimated previously, using the 10-fold random cross-validation.
- **Prediction uncertainty** Prediction uncertainty taken as the standard deviation of 10 model predictions is not correct. There are several options to estimate the prediction uncertainty. The authors could use bootstrapping and obtain confidence intervals. Note that bootstrapping requires fitting at least 100 models to get a realistic estimate. Another option is to use quantile regression. Currently, the prediction uncertainty maps are a standard deviation value obtained from 10 predicted values. This is too little to obtain a reliable estimate of the standard deviation and, again, not a correct way to obtain a valid estimate of prediction uncertainty. Why do the authors not propagate the uncertainty of the parameters as obtained by the posterior parameter distribution in the Bayesian analysis? There is a great opportunity here with the Bayesian analysis of the parameters and the distribution obtained by the posterior distribution. For the uncertainty, ideally, prediction intervals are reported. When obtaining prediction intervals is not possible, the authors could always obtain a confidence interval by bootstrapping.

Permutation importance

The permutation analysis suffers from the same problem of using 10 models. The authors report grouped permutation variable importance values from 10 models, with error bar representing the

standard deviation obtained by cross-validation. This is not common, and I would say not correct. Authors should compute the permutation importance of a single model, the one calibrated using all data. This way they are sure that the permutation values indeed correspond to the model they are using for mapping. Since the permutation values are not additive, I am afraid that Extended Data Fig. 7 is not realistic. Also, error bar usually represents the permutation error. Since permutation involves randomness, it is important to repeating the permutation many times and also show the 5% and 95% quantile of importance values from the repetitions. The permutation error obtained this way is likely to appear much larger than that currently reported. The MSE values obtained by permutation importance are not additive. This means that the permutation importance values for each individual covariate cannot be simply added to another one to calculate the permutation importance of the group. To compute the group permutation importance, the authors should make a permutation directly on the group of covariates. The results are likely to be different, because permutation importance is a method that is sensitive to dependence among covariates. Authors should make sure that they permuted the group of covariates simultaneously, not the individual covariates and then summed the individual permutation importance values. This is very unclear from the text at L. 663-668.

Statistical validation of maps

L. 579-590: There is a current hype about the idea that spatial autocorrelation should be accounted for when estimating validation statistics in a spatial context. This is a misconception and I urge the authors not to propagate this wrong idea in a scientific paper. It is good that they only tested this but did not include the results of spatial cross-validation in their analysis. In short, spatial autocorrelation does influence the estimation of map accuracy statistics, but spatial cross-validation is clearly not an answer to this problem. The only solution to obtain an unbiased estimate of map accuracy is to collect an additional probability sample from the population (that is, the world) and to use a design-based estimation of the statistics. I acknowledge this is no feasible for this kind of global study, or very difficult to implement. But spatial cross-validation is not the solution. It provides overpessimistic validation statistics and has no underlying theory. Why did the authors not use instead a model-based estimation of map accuracy? Why didn't they use a heuristic method based on spatial weighting? There are many methods available for dealing with estimation of map accuracy in case of clustered data, but clearly spatial cross-validation is the worst. It is important that this paper does not propagate these recent misconceptions of statistical validation of maps. I recommend to simply remove this paragraph.

Additional minor comments:

There are a few sentences that are difficult to read, such that at L. 76-77, for example. The manuscript would benefit from shorter sentences.

L. 87-90: This sentence suggests that deep learning IS a hypothesis. Perhaps adding "with" or "using" would clarify this point.

L. 91-93: Consider rephrasing, this sentence is quite unclear: "We collated 132 pairs of data sets from 16 experimental studies *...+ where SOC content were measured at 46 locations": what is collected? The 132 pairs of data sets or the 46 measurement at locations? AHA this is now clear at L. 405-406.

L. 415: How are the coarse fragments obtained? I think WoSIS has only few samples containing coarse fragment data.

L. 418-419: How many observations were used to fit the pedotransfer function and to obtain the regression parameters for bulk density?

L. 447-460: This description is quite unclear. The authors used what is commonly referred to as a Nash-Sutcliffe modelling efficiency, that some would call the coefficient of determination (of the 1:1 line). I agree with their use of the term “coefficient of efficiency”. Several authors (e.g. Janssen & Heuberger, 1995) call it a modelling efficiency. But the description that the authors make of this statistic is a bit surprising. A modelling efficiency of 0.75 is very high. It means that 75% of the SOC depth distribution variance is explained by CLM5. The authors should also consider rephrasing this paragraph and make shorter sentences.

L. 557-560: Calling deep learning a neural network with four hidden layers and 256, 512, 512, and 256 neurons, respectively, is a bit misleading. I agree that it is not clear when a neural network becomes “deep”, but we usually agree that this is a complex model with many layers/neurons.

L. 563-564: The average in the MSE equation is missing.

L. 649: How many permutations were made. The number of permutations should be sufficiently high to obtain reliable results.

Reference:

Janssen, P. H. M., & Heuberger, P. S. C. (1995). Calibration of process-oriented models. *Ecological Modelling*, 83(1-2), 55-66.

Referee #3 (Remarks to the Author):

This is an interesting study that aims to show the importance of carbon use efficiency for soil carbon storage. The authors leverage a literature synthesis of CUE measurements, 50K+ WoSIS soil profiles with SOC measurements, the CLM5 biogeochemical model, and their PRODA approach to estimate CUE for the soil profiles and globally. What stands out most is the computational framework, which will be an important tool for ongoing and future data-model integration. I also agree that better constraining CUE in models – especially given the interpretability of CUE across different types of measurements and model formulations – is crucial. However, there are several conceptual and practical considerations that compromise the present study and the novelty of its purported findings. I outline these main concerns below.

1. I am not convinced that the main conclusions are particularly novel.

– The finding that CUE and SOC have a positive relationship has been previously demonstrated (Kallenbach et al., 2015; Malik et al., 2018; Buckeridge et al., 2020) and is largely expected. CUE by definition dictates how much carbon stays in the system versus how much leaves the system, especially here where the model-derived CUE is an emergent property of the whole system (rather

than the fraction of C acquired that is specifically used for microbial growth and biosynthesis in the empirical measurements).

– While the positive relationship between CUE and SOC may be expected, the slope and how it is affected by climate and vegetation type, etc., would be interesting to explore further. CUE is a dynamic property that is affected by many factors, including some of those treated as separate mechanisms herein. Understanding the processes shaping CUE is an important step in improving its representation in process-based models, but, other than Ext. Data Fig. 7, this study does not yet provide clear insights on the processes leading to differences (and spread) in observed CUE.

– Furthermore, the CUE that is estimated from CLM5 is an emergent system value (including litter pools, it seems) which is quite different to those measured in soils (that also vary between different methods; Geyer et al. 2018). The slope of the CUE and SOC relationship looks significantly lower in the literature synthesis compared to the global estimates... why? What does this mean and what drives this slope? If the authors used their same framework with SOC values from the CUE literature synthesis to estimate the corresponding CLM5-derived bulk CUEs, how would these compare to the measured ones? Fig. 2b/d with MAT is interesting (though also well-documented; Hagerty et al. 2014, Allison et al. 2010) and the slopes seem to somewhat agree between the literature synthesis and global estimates. How do these slopes compare to the temperature sensitivities reported in other studies?

2. The findings of CUE importance rely on the selected model (i.e., CLM5) and may not be robust. The study is also framed around two alternative hypotheses that are not mutually exclusive and, ultimately, the current study does not appear equipped to truly address them.

— It would be helpful to include several global soil models, including process-based (i.e., microbial-explicit) models. The authors discuss microbial feedbacks extensively, but the approach they take is not appropriate to support their claims (more on this below).

– The authors support their choice of CLM5 as a “process-oriented biogeochemical model... because it is depth resolved and expressed in matrix form” (L113). First, this is a little deceiving because CLM5 is not process-based, or at least does not include the processes outlined in the hypotheses. It does not include microbes, and hence the CUE is simply a fraction that stays in the system versus leaving the system when transferred between first-order pools. Second, the two reasons given for the selection of CLM5 alone as the backbone of their study are not convincing nor appropriate. There are several models that are depth-resolved and virtually any existing model can be written in matrix notation. Indeed, all first-order global models can be easily written in matrix notation and even process-based models can be linearized, as needed, near the steady-state (given that they invoke a steady-state assumption anyway).

– Regarding the framing of negative vs. positive CUE-SOC relationships, the authors propose two alternative hypotheses, but these two hypotheses are by no means mutually exclusive. Both schemes can, and do, occur in real life. The positive feedback of microbial biomass and enzyme production (as in Fig. 1b) could, in addition to catalyzing decomposition, promote mineral-organic associations through the increase in sorption of microbial necromass and byproducts. This should be

discussed, and makes me wary of the current framing of these two simple pathways as conflicting hypotheses.

– Furthermore, I am not sure that the alternative hypothesis of high CUE leading to loss of SOC truly makes sense. High CUE could drive more microbial biomass, and more subsequent decomposition that passes through microbial biomass, but that could just result in a faster cycling system. It does not imply a decrease in storage. (It also doesn't imply young radiocarbon ages, as microbial recycling contributes to older carbon.) Yet, the authors state that microbial models that “simulate direct dependence of SOC loss on microbial biomass via enzyme activities... always generate a negative relationship between CUE and SOC” (L193). I do not believe that this is always the case and I would challenge the authors to demonstrate this. It would be interesting to try a microbial-explicit model in the same framework. The cited studies (e.g., Allison et al. 2010 and others) with simple microbial-explicit models show that microbial feedbacks can indeed lead to losses, but not that higher CUE (and all else equal) increases losses. I feel that they may be conflating trends in microbial biomass with a dependence on the value of CUE.

3. Proportional changes in the selected ‘mechanisms’ are difficult to interpret in a meaningful way.

– When considering proportional changes in the selected mechanisms, what if the mechanisms or relative changes were normalized and based on the distribution of potential values? That is, an 8% change in one mechanism might constitute a small change within the potential range of values for that mechanism, whereas it might span the range of potential values for another mechanism. Instead, maybe values could be varied from their respective means to plus/minus one (or two) standard deviations. A supplemental figure depicting the distributions of all mechanisms (as with CUE in Ext. Data Fig. 5) would be helpful in that regard; from the box plots in Fig. 3, it seems that some of the distributions are indeed wider than others.

– The results also state that a 10% increase in SOC needs a 3% increase in CUE and a 7% decrease in the environmental modifier. The latter is a model construct and not very meaningful. How much does the overall ‘environmental modifier’ vary spatially in the best fit model? How much can it change under future conditions? It is not apparent what a 7% decrease in the environmental modifier really means with regards to temperature, moisture, etc. sensitivity, so it is unintuitive to interpret this.

Other comments:

- The “6 mechanisms” in this study are not really mechanisms per se. For example, “environmental modifier” is not a mechanism... I'd argue that neither is baseline decomposition, but rather it is a model construct of a suite of mechanisms.

- The following sentence in the abstract “Our findings support a hypothesis that high microbial CUE favors preservation of carbon as SOC instead of stimulation of soil respiration” seems trivial, by definition of CUE.

- The authors should consider mentioning more explicitly in the abstract that the ‘synthetic analysis’

of all of these profiles relies on a data-model integration (and specifically CLM5) to quantify the various “mechanisms”, as the majority of these mechanisms (e.g., carbon inputs, vertical transport, CUE) are not directly measured within the globally-distributed soil profiles. Simply saying “the retrieved CUE from the global SOC database” could be deceiving. The finding that CUE is “twice as important” may largely depend on the choice of model (i.e., CLM5), so this context is important.

- The depth findings are interesting, but I would like to see them supported empirically, not just using model-derived values. The mixed-effects model results suggest that depth is important for SOC, of course, but what about exploring depth effects on CUE. You give results from a mixed-effects model of SOC, but could also explore variation in CUE.

- Furthermore, with regards to depth in Ext. Data Fig. 4, how can we interpret these findings in the context of studies that have shown that similar amounts (or at times more) of SOC is microbially-derived at depth compared to the surface? Microbial recycling plays an important role at depth. This could be discussed further.

- Can carbon input be included in Ext. Data Fig. 7? I understand it is not estimated in the same way, because it is CLM5 derived. But, the environmental variables listed do also affect carbon inputs.

- Why is there no/little uncertainty in input carbon (Ext. Data Fig. 10)? I suppose this again has to do with the fact that input carbon is derived from CLM5. How much of a difference does it make if MODIS NPP or similar is used? Of course NPP is only a proxy for what actually enters the soil, and inputs are very difficult to measure accurately (e.g., roots, exudation), especially at scale.

- It is not apparent why CUE is weighted in the way it is. I understand that the goal is to weight CUE by the flux going through each pathway. In contrast, all of the other “mechanisms” are weighted simply by the pool carbon densities. Why isn’t the decomposition rate weighted by the fluxes as with the CUE? In the present way, the weighting of CUE incorporates the decomposition rates, environmental modifiers, and soil thickness... so it is an amalgamation of various “mechanisms” presented herein. It would be helpful if the authors elaborated on their choice of weighting and the ranges of each value. Even the weighting for carbon input allocation, inspired by the ‘beta’ exponential functions of Jackson et al. 1996, could be explained more clearly. What about the proportion of input carbon allocated as surface litter? (Furthermore, I assume that soil depth D_z and thickness Δz are in units of meters, but this should be specified as with the other variables. Though eventually B should come out as unit-less.)

- How large are the uncertainties from the profile-level data assimilation? (L602: “uncertainties of parameters and SOC simulation did not propagate from the profile-level data assimilation but only reflected uncertainties generated by the deep learning model”)

- How would the results change if CLM5 were assimilated to global SOC (e.g., SoilGrids) instead of the profile values? This would introduce some uncertainty from the scale-up of SOC and it’s underlying covariates, but you are also introducing uncertainty with the parameter estimation from similar covariates.

- As the authors say themselves, the findings are based on (and interpreted with) CLM5, which does not explicitly represent microbial processes. However, they say that “CLM5 explicitly represents CUE via partitioning... to respiration versus accrual to SOC” (L188). This is by definition of CUE. This sentence tries to support their choice of model, but really CUE is a necessary construct in any soil model that dictates the allocation of carbon staying in the system versus leaving the system.

- What fraction of the WoSIS and NCSCD profiles were from natural/undisturbed ecosystems? Is the steady-state assumption (L501-505) reasonable?

- L165: ‘Deep learning’ here is jargon-y and doesn’t give much information. Specify neural network in the text.

- L152: The authors state that boreal regions have high quality (low C:N ratio) SOC substrates. They cite Reich et al. 2004, which focuses on N:P ratios, and states that “N is the major limiting nutrient in younger temperate and high-latitude soils,” implying that boreal regions are N-limited. Indeed, boreal regions can have high C:N ratio litter and soil stocks, which is seemingly the opposite of what the authors state.

I won’t go into minor details and typos (e.g., L153 lability not liability, L486 are not is, L649 specifically not specially), given the above major comments that need re-thinking.

This study proposes to use global and publicly available datasets complemented by a meta-analysis to draw conclusions on the potential drivers of global soil organic carbon (SOC) storage. The statistical modelling of SOC storage was made by a “PROcess-guided deep learning and DATA-driven modelling”, which, to summarize, aim to use a deep learning model to estimate the parameters of a process-based biochemical model.

Overall I found this study of potential interest, but I also raise substantial concerns on specific aspects of the mapping, validation, and interpretation process. I am however positive that the authors could address these concerns by a revision and additional analyses, and that the revised manuscript could be a useful contribution.

Cross-validation

The cross-validation should be used to estimate the validation statistics (as is done), but for prediction and uncertainty quantification a model fitted with all the data should be used. In this paper, the authors used 10 models, resulting from fitting 10 models on the 10 cross-validation folds, and use the mean value from the 10 model predictions as their final prediction. They also use the standard deviation of the 10 predictions as an estimate of prediction uncertainty.

- **Prediction** The final model for mapping should be calibrated with all the available observations. This is the “best guess model”, calibrated with all data available. Map accuracy is estimated previously, using the 10-fold random cross-validation.
- **Prediction uncertainty** Prediction uncertainty taken as the standard deviation of 10 model predictions is not correct. There are several options to estimate the prediction uncertainty. The authors could use bootstrapping and obtain confidence intervals. Note that bootstrapping requires fitting at least 100 models to get a realistic estimate. Another option is to use quantile regression. Currently, the prediction uncertainty maps are a standard deviation value obtained from 10 predicted values. This is too little to obtain a reliable estimate of the standard deviation and, again, not a correct way to obtain a valid estimate of prediction uncertainty. Why do the authors not propagate the uncertainty of the parameters as obtained by the posterior parameter distribution in the Bayesian analysis? There is a great opportunity here with the Bayesian analysis of the parameters and the distribution obtained by the posterior distribution. For the uncertainty, ideally, prediction intervals are reported. When obtaining prediction intervals is not possible, the authors could always obtain a confidence interval by bootstrapping.

Permutation importance

The permutation analysis suffers from the same problem of using 10 models. The authors report grouped permutation variable importance values from 10 models, with error bar representing the standard deviation obtained by cross-validation. This is not common, and I would say not correct. Authors should compute the permutation importance of a single model, the one calibrated using all data. This way they are sure that the permutation values indeed correspond to the model they are using for mapping. Since the permutation values are not additive, I am afraid that Extended Data Fig. 7 is not realistic. Also, error bar usually represents the permutation error. Since permutation involves randomness, it is important to repeating the permutation many times and also show the 5% and 95% quantile of importance values from the repetitions. The permutation error obtained this way is likely to appear much larger than that currently reported.

The MSE values obtained by permutation importance are not additive. This means that the permutation importance values for each individual covariate cannot be simply added to another one

to calculate the permutation importance of the group. To compute the group permutation importance, the authors should make a permutation directly on the group of covariates. The results are likely to be different, because permutation importance is a method that is sensitive to dependence among covariates. Authors should make sure that they permuted the group of covariates simultaneously, not the individual covariates and then summed the individual permutation importance values. This is very unclear from the text at L. 663-668.

Statistical validation of maps

L. 579-590: There is a current hype about the idea that spatial autocorrelation should be accounted for when estimating validation statistics in a spatial context. This is a misconception and I urge the authors not to propagate this wrong idea in a scientific paper. It is good that they only tested this but did not include the results of spatial cross-validation in their analysis. In short, spatial autocorrelation does influence the estimation of map accuracy statistics, but spatial cross-validation is clearly not an answer to this problem. The only solution to obtain an unbiased estimate of map accuracy is to collect an additional probability sample from the population (that is, the world) and to use a design-based estimation of the statistics. I acknowledge this is not feasible for this kind of global study, or very difficult to implement. But spatial cross-validation is not the solution. It provides overpessimistic validation statistics and has no underlying theory. Why did the authors not use instead a model-based estimation of map accuracy? Why didn't they use a heuristic method based on spatial weighting? There are many methods available for dealing with estimation of map accuracy in case of clustered data, but clearly spatial cross-validation is the worst. It is important that this paper does not propagate these recent misconceptions of statistical validation of maps. I recommend to simply remove this paragraph.

Additional minor comments:

There are a few sentences that are difficult to read, such that at L. 76-77, for example. The manuscript would benefit from shorter sentences.

L. 87-90: This sentence suggests that deep learning IS a hypothesis. Perhaps adding "with" or "using" would clarify this point.

L. 91-93: Consider rephrasing, this sentence is quite unclear: "We collated 132 pairs of data sets from 16 experimental studies [...] where SOC content were measured at 46 locations": what is collected? The 132 pairs of data sets or the 46 measurement at locations? AHA this is now clear at L. 405-406.

L. 415: How are the coarse fragments obtained? I think WoSIS has only few samples containing coarse fragment data.

L. 418-419: How many observations were used to fit the pedotransfer function and to obtain the regression parameters for bulk density?

L. 447-460: This description is quite unclear. The authors used what is commonly referred to as a Nash-Sutcliffe modelling efficiency, that some would call the coefficient of determination (of the 1:1 line). I agree with their use of the term "coefficient of efficiency". Several authors (e.g. Janssen & Heuberger, 1995) call it a modelling efficiency. But the description that the authors make of this statistic is a bit surprising. A modelling efficiency of 0.75 is very high. It means that 75% of the SOC depth distribution variance is explained by CLM5. The authors should also consider rephrasing this paragraph and make shorter sentences.

L. 557-560: Calling deep learning a neural network with four hidden layers and 256, 512, 512, and 256 neurons, respectively, is a bit misleading. I agree that it is not clear when a neural network becomes “deep”, but we usually agree that this is a complex model with many layers/neurons.

L. 563-564: The average in the MSE equation is missing.

L. 649: How many permutations were made. The number of permutations should be sufficiently high to obtain reliable results.

Reference:

Janssen, P. H. M., & Heuberger, P. S. C. (1995). Calibration of process-oriented models. Ecological Modelling, 83(1-2), 55-66.

Summary of responses to comments by three referees

We greatly appreciate three referees for offering us constructive and insightful comments, which were extremely helpful for us to improve our manuscript. Following their suggestions, we have conducted additional analyses and found that our original findings are robust regarding the role of microbial carbon use efficiency (CUE) in determining the global soil organic carbon (SOC) storage. Now, we have thoroughly revised the manuscript to address all these comments and to reflect the new insights we obtained from the new analyses. Here are a summary of the revision of our manuscript.

1. Microbial model: Both referees #1 and #3 suggested that we use a microbial model, in addition to the Community Land Model (CLM5), to examine the CUE-SOC relationship. In the revision, we followed this suggestion and included a microbial model that explicitly represents the direct dependence of SOC loss on microbial biomass via enzyme activities (i.e., the second possible CUE-SOC relationship as described by Fig. 1b in the original manuscript). We conducted a parameter sensitivity analysis and found that the microbial model generated either a positive or a negative CUE-SOC relationship, depending on parameter values. Constrained parameter values by the globally distributed SOC vertical profiles yielded a positive correlation between CUE and SOC. Thus, the positive CUE-SOC relationship is supported by both CLM5 and the microbial model regardless of their structural differences. Moreover, we found that, although described by the Michaelis-Menten equation, the nonlinear kinetics of SOC decomposition where the enzyme was involved can be approximated by a first-order kinetics with respect to SOC after the microbial model was constrained by globally distributed SOC profiles. Thus, model structures that use first-order kinetics, such as in CLM5, were effective in simulating SOC at the regional and global scales. Detailed results are presented and discussed in the Response Letter section 1.1 (R1.1).

2. Novelty of this study: Referee #3 raised concerns about the novelty of our study. Our response in R3.1 highlights the novel contributions of our study in at least two aspects. First, we identified CUE from the global SOC database as the most important mechanism for determining global SOC storage among the six mechanisms. While microbial physiology in general and microbial CUE in particular have been extensively studied, their relatively importance to SOC storage has not been carefully evaluated. SOC research has been traditionally focused on carbon balance between carbon input and decomposition. Our study,

for the first time, retrieved the global patterns of CUE and other mechanisms (e.g., organic carbon input, decomposition, and vertical transport) from 52,819 soil profiles and quantified their relative importance to global SOC storage and its spatial distributions by the PROcess-guided deep learning and DATA-driven modelling (PRODA) approach. Second, we resolved a controversy on the CUE-SOC relationship at the global scale. Both our meta-analysis and analyses of globally distributed SOC data using two structurally different models support that a high microbial CUE favours the accumulation of SOC rather than SOC loss. Our findings have the potential to shift the SOC research from a classic paradigm that focuses on the roles of plant carbon inputs and decomposition to a new paradigm that emphasizes the role of microbial physiology and ecology in soil carbon sequestration.

3. Correlation vs. causation: Following the suggestions of referee #1, we conducted mixed-effects modeling to ensure the statistical rigor of the findings reported in the manuscript. Our new results showed that the positive relationship between CUE and SOC was robust, even after using different model structures and considering potential interactions among the predictors. This empirical relationship was also consistent with results obtained from data assimilation. As the referee suggested, results of the mixed-effects modeling highlighted the importance of within-study association of the data in explaining the CUE-SOC relationship. The explained variance in SOC was only 37% for the fixed-effects and increased to 55% after considering the random effects in the meta-analysis. Moreover, we investigated relationships among CUE, microbial biomass, and non-microbial biomass storage. We found that a high CUE accompanied not only high microbial biomass carbon, but also high non-microbial biomass carbon. The revised manuscript now shows both statistical (from the meta-analysis) and process-based (from the microbial model results) evidence that microbial partitioning of carbon toward microbial growth over respiration will enhance SOC accumulation via microbial by-products and necromass. The detailed information and statistics can be found in R1.2 and R3.1b.

4. Uncertainty of modeling results: Following the suggestions of referee #2, we presented the results of predictions from the best-guess neural network. Meanwhile, we applied a 200-time bootstrapping method to quantify the prediction uncertainties. The main conclusions, as well as the spatial patterns of CUE and the other five mechanisms, did not change. Yet the uncertainty ranges (e.g., the permutation importance), as suggested by referee #2, did increase in comparison to those from the ten-fold cross-validation method. Moreover, as

suggested by referee #2, validation methods that considered spatial correlation of data may not be well-grounded. We removed results and discussions about spatial autocorrelation from the manuscript to avoid potential confusion.

We hope that our responses and revision of the manuscript are satisfactory to the referees. And we look forward to further feedbacks and comments.

Below are our point-by-point responses (in blue) to referees' comments (in black).

Point-by-point responses to comments by three referees

Referee #1 (Remarks to the Author):

The main findings in Tao et al. are exciting and the approaches used are technically advanced and appear to open new doors to enquiry. That said, as currently presented, it is hard for the reader to evaluate the veracity of how the main findings were reached. I would like to see the authors wrestle directly with issues of correlation vs. causation, that pattern does not beget process, space-for-time substitutions, and strong inference. I have the following specific comments related to these common challenges to identifying causation that, if satisfactorily dealt with, I hope will help to provide the necessary rigor to support the exciting findings that are presented. Please note that, if I were the Editor, #1 would sink the current paper if not addressed fully; and I would regard #2 as important for ensuring statistical rigor in the work published in my journal.

Response: We appreciate the referee's generally positive assessments of our findings and approaches. We have done additional analyses to clarify the causation and revised the text to avoid confusion regarding space-for-time substitution.

1. Strong inference and pattern does not beget process. I like the competing hypotheses shown in Fig. 1a, where the possibilities are that higher CUE favors accumulation of SOC vs. loss. These mutually exclusive hypotheses (at least as presented) lend themselves well to a strong inference test. However, the CLM5 belowground carbon cycle module is structured around the first hypothesis being true. That is, CUE determines the proportion of plant C inputs into the soil that form SOC vs. are respired as CO₂. As such the structure of the model is pre-determined to support the first hypothesis and falsify the second. This appears to add strong circularity to the results, meaning the work falls far short of a strong inference test of competing hypotheses and appears to fall into the trap of assigning process to pattern. It would seem that to reliably test between these two hypotheses, you would need a second-order process in CLM5 where the amount of microbial biomass regulates the rate at which SOC is formed and decomposed. If I have misinterpreted what was done, I apologize to the authors but would still ask that these issues be directly addressed in the main paper, so that the same concerns do not surface once the paper is accepted (because these concerns about

circularity, if not resolved, suggest a fundamental flaw in the study design that questions the validity of the hypothesis testing).

R1.1 We appreciate the referee's point that the CUE-SOC relationship may always be positive because of the structure of CLM5. As suggested by the referee, we used a microbial model to examine the CUE-SOC relationship. Our new results showed that CUE and SOC are positively correlated regardless of the model structures once constrained by data. Below we describe what we have done in response to these comments.

First, we developed a vertically-resolved microbial model, in addition to CLM5, to examine the relationship between microbial CUE and SOC storage. The model structure is shown in Response Letter Fig. 1b. Briefly, the microbial model shares the same structure with CLM5 (Response Letter Fig. 1a) in describing vertical transport and litter dynamics. In each soil layer, the model follows the one proposed by Allison et al.¹ (which is widely used as the basis for microbial model development) and has four pools, which are enzyme carbon pool (ENZ), microbial biomass carbon pool (MIC), dissolved organic carbon pool (DOC) and soil organic carbon pool (SOC), respectively. Similar to CLM5, the microbial model was also expressed in the matrix equation (Equation 2 of the manuscript) but the decomposability matrix $\mathbf{K}(X)$ is dependent on the carbon pool state (i.e., a nonlinear microbial model). Five major processes included in $\mathbf{K}(X)$ are (1) assimilation of dissolved organic carbon to microbial biomass (*ASSIM*): $ASSIM = v_{max,assim}MIC \frac{DOC}{k_{m,assim} + DOC}$; (2) decomposition of soil organic carbon (*DECOM*): $DECOM = v_{max,decom}ENZ \frac{SOC}{k_{m,decom} + SOC}$; (3) enzyme production (*PROD_{ENZ}*): $PROD_{ENZ} = k_{enz,prod}MIC^\beta$; (4) microbial mortality (*DEATH*): $DEATH = k_{death}MIC$; and (5) enzyme decay (*DECAY_{ENZ}*): $DECAY_{ENZ} = k_{enz,decay}ENZ$. Parameters $v_{max,assim}$ and $v_{max,decom}$ represent the maximum assimilation and decomposition rates, respectively. $k_{m,assim}$ and $k_{m,decom}$ are the Michaelis constants for assimilation and decomposition, respectively. $k_{enz,prod}$ describes the enzyme production rate. k_{death} is the mortality rate of the microbes. Note that for enzyme production, Sinsabaugh et al.² found allometric relationships between microbial biomass and extracellular enzyme production in a meta-analysis. We therefore applied an allometric equation (varying β value) instead of the first-order equation ($\beta = 1$) used by Allison et al.¹ to describe enzyme production in the microbial model.

Response Letter Fig. 1 | Structures of vertically-resolved CLM5 (a) and the microbial model (b).

Microbial carbon use efficiency in the microbial model is a predefined parameter as the proportion of carbon assimilation that is allocated to microbial growth (Response Letter Fig. 1b). Different from the first-order kinetics used in CLM5, the microbial model specifies the decomposition of soil organic carbon and dissolved organic carbon (i.e., assimilation process) as Michaelis-Menten kinetics, where the decomposition of substrate is determined by both the catalyst (i.e., microbes for assimilation and enzyme for decomposition) and the substrate (i.e., DOC for assimilation and SOC for decomposition) *per se*. Notably, when the Michaelis constants (i.e., $k_{m,decom}$ and $k_{m,assim}$) are much larger (e.g., 100 times larger) than their corresponding substrate concentration, the corresponding Michaelis-Menten kinetics could be approximated by a first-order functional form with respect to the substrate.

Second, we conducted a parameter sensitivity analysis for the microbial model. By this analysis, we explored how parameters representing enzyme dynamics (i.e., allocation slope β and enzyme turnover term τ in enzyme production) could influence the CUE-SOC relationship. We chose values of parameters in the microbial model that have been reported in the literature³ (Response Letter Table 1) and assessed the CUE-SOC relationship at steady state with different value combinations of the above-mentioned two parameters.

Because of the strong nonlinearity of the microbial model, it is no longer feasible to analytically solve the soil carbon pool sizes at steady state as what we did using CLM5. Instead, we numerically integrated the model forward until all soil carbon pools reached steady state (i.e., the total SOC storage change $< 1 \text{ g C m}^{-2} \text{ year}^{-1}$) for given parameters values. A maximum of 10,000-year simulation by recycling the 20-year forcing from CLM5 was used in the forward simulation. We discarded those parameter sets for which the forward integration did not reach steady state after 10,000-year simulation. Response Letter Fig. 2 shows that the CUE-SOC relationship can be either negative or positive, depending on the

choice of parameters values for enzyme production (i.e., the allometric slope and turnover rate of enzyme production). Notably, lower values of the exponent β tend to cause positive relations between SOC and CUE. This means that less-than-linear scaling of enzyme production and biomass does not allow decomposition to speed up at high CUE (argument illustrated in Fig. 1b), while necromass contributions to SOC become more important and drive the positive SOC-CUE relation (argument illustrated in Fig. 1a).

Response Letter Fig. 2 | Varying steady-state CUE-SOC relationships under different parameter values for enzymes production in the microbial model. β is the allocation slope and τ is the enzyme turnover term in enzyme production.

Third, we applied data assimilation at each soil profile to estimate parameters in the microbial model. By constraining model parameters from the data, we can distinguish one CUE-SOC relationship at the global scale. The numerical forward integration with the microbial model requires much longer time than the analytical solution of the matrix equation with CLM5 to get the SOC storage at steady state. Considering the fact that Bayesian MCMC requires a sufficiently long Markov chain (e.g., 50,000 iterations in our study and therefore 50,000 times 10,000-year forward integrations using the microbial model) to gain reliable results, it is no longer computationally feasible for us to use MCMC to assimilate soil data into the microbial model. As an alternative to Bayesian MCMC, we applied the Shuffled Complex Evolution (SCE) method⁴ to assimilate SOC data into the microbial model at each soil profile. The SCE method is an efficient global optimization algorithm that calibrates model parameters to their global optima against observations. Parameters selected for data

assimilation are listed in Response Letter Table 1. We kept the prior ranges of parameters in the microbial model to be the same as in CLM5 except those newly introduced microbe-related parameters (Response Letter Table 1). Because soil profile data from field measurements contained all components of the organic carbon, we used the total carbon amount of the four soil carbon pools in the microbial model to be compared with soil profile data of WoSIS database.

We applied empirical constraints to model simulations in data assimilation. Simulated SOC at steady state had to exceed microbial biomass carbon (MIC) by at least 50-fold over the entire soil profile and by at least 15-fold for the top 15 cm of soil, consistent with general patterns measured in the field⁵. Similarly, simulated SOC at steady state had to be at least 10-fold higher than DOC for the entire soil profile and at least 100-fold higher for the top 15 cm of soil^{6,7}. Additionally, simulated microbial biomass carbon (MIC) was constrained to be larger than the total pool of soil enzymes (ENZ).

Fourth, we tested the robustness of the estimated CUE-SOC relationship. Even after applying the Shuffled Complex Evolution method, the computational cost for data assimilation with the microbial model was approximately 2,600 times higher than with CLM5 model. It is prohibitively high. To make the analysis feasible, we randomly sampled a subset of 1000 profiles for data assimilation with the microbial model. To avoid over-representation of any specific regions in random sampling, we assigned weights to each soil profile according to its shared area of the corresponding climate types (i.e., total climate zonal area over the number of soil profiles that belongs to this climate zone). Our analysis indicates that this subset of 1,000 sampled profiles offers robust estimates of slope and intercept of the CUE-SOC relationship (Response Letter Fig. 3).

Response Letter Fig. 3 | Robustness of the regression slopes and intercepts of the CUE-SOC relationship in the mixed-effects model. Results from the 1,000 soil profiles in data assimilation with the microbial model were randomly sampled with different subset sizes (each for 1,000 times) and regressed in the mixed-effects model. The regression slope and intercept of the CUE-SOC relationship remain stable under different sample sizes. Thus, the results from the data assimilation of 1,000 profiles with the microbial model are robust. The lower, middle, and upper hinges in the boxplot show the first, median, and third quartiles of the distribution. Whiskers represent the 1.5 interquartile range from the hinges. Red dashed lines are the regression results from the 1,000 soil profiles.

Fifth, we compared the CUE-SOC relationships retrieved from the globally distributed SOC data using the microbial model and CLM5. After applying the Shuffled Complex Evolution method to assimilate SOC data into the microbial model at each of those sampled soil profiles, we retrieved the optimized parameter values and corresponding simulated SOC storage. Results are presented in Response Letter Fig. 4a-c. First, there is a positive relationship between CUE and SOC across different climate zones, at different soil depths, regardless model structures. The positive CUE-SOC relationship emerged not only from CLM5, but also from the microbial model after data assimilation (and is consistent with the empirical results from the meta-analysis).

Response Letter Fig. 4 | CUE-SOC relationships at different soil depths after assimilating the 1,000 representative SOC profiles into the microbial model (a - c) and all the 52,819 profiles into CLM5 (d - f). Black lines and statistics shown in the figure are results from linear mixed-effects model regressions (details of the mixed-effects model structures and sample sizes were reported in Response Letter Table 3). Declining explanatory power of CUE to the variation in SOC with soil depths indicates a shift from biotic-dominated to abiotic-dominated SOC accumulation.

Sixth, we examined parameter values in the Michaelis-Menten equation of the microbial model (Response Letter Fig. 5). For most of the soil profiles, Michaelis constant for decomposition (i.e., $k_{m,decom}$) is 100 times larger than its substrate concentration (i.e., SOC concentration). Thus, the nonlinear kinetics for SOC decomposition where assumptions of enzyme kinetics are involved can be approximated by a first-order kinetics with respect to SOC after the microbial model is constrained by globally distributed SOC vertical profiles. Such approximation has been supported by theoretical upscaling⁸ and observed macroscopic patterns from widely reported litter and SOC decomposition experiments⁹⁻¹².

Redacted

In summary, the first finding from data assimilation with the microbial model (as represented by Response Letter Fig. 4) highlighted the robustness of our conclusion that CUE promotes SOC storage at the global scale. The second finding (as represented by Response Letter Fig. 5) supported the approximation by a first-order kinetics with respect to SOC (i.e., CLM5 in this study) to effectively simulate SOC storage at regional and global scales and the use of the PRODA approach to evaluate the relative importance of different mechanisms for SOC storage.

We revised the main text and extended data to include the new analyses and results. Specifically, we introduced the microbial model, its parameter sensitivity analysis, and its data assimilation in L557 - L633. Results of the microbial model were discussed in L111 - L122. Fig. 2b of the revised manuscript showed the results from the microbial model. Response Letter Figs. 1 - 5 were included in the Extended Data Figs. 3 - 6 and 8. Response Letter Table 1 was included as Extended Data Table 8. Finally, the spatial distributions of the subset of 1,000 representative profiles was shown in Extended Data Fig. 2e.

Response Letter Table 1 | Parameters in the vertically-resolved microbial model that were optimized in the profile-level data assimilation. The column "Origin" indicates whether the parameters are from the microbial model or the original soil carbon module of CLM5.

No.	Name	Matrix term	Corresponding mechanism	Description	Origin	Conventional values	Unit	Prior range
1	mic_cue	A		Microbial carbon use efficiency for soil organic carbon	MIC	0.6	unitless	[0.01 0.7]
2	pdeath2soc	A		Fraction of mic_cue that leads to death	MIC	0.5	unitless	[0.1 0.9]
3	fs2l3	A	Microbial carbon use efficiency (CUE)	Transfer fraction, lignin litter to slow SOC	CLM5	0.5	unitless	[0.2 0.8]
4	fs1l2	A		Transfer fraction, cellulose litter to fast SOC	CLM5	0.5	unitless	[0.2 0.8]
5	fs1l1	A		Transfer fraction, metabolic litter to fast SOC	CLM5	0.45	unitless	[0.1 0.8]
6	fs1s2	A		Transfer fraction, slow SOC to fast SOC	CLM5	0.42	unitless	[0.1 0.74]
7	tau4s1	K		Inverse of $v_{max,assim}$ in assimilation process	MIC	0.011, 0.22	year	[10^{-3} 1]
8	tau4s2_death	K		Turnover time for microbial biomass	MIC	0.07, 0.27	year	[0.01 1]
9	tau4s2_enz	K		Turnover time for enzyme production	MIC	9.8, 20	year	[1 30]
10	tau4s3	K		Turnover time for enzyme decay	MIC	0.14	year	[0 1.5]
11	tau4s4	K		Inverse of $v_{max,decom}$ in assimilation process	MIC	1.1×10^{-4} , 4.6×10^{-5} , 2×10^{-7}	year	[10^{-7} 10^{-2}]
12	allo_slope	K	Substrate decomposability	Allometric slope in enzyme production	MIC	1	unitless	[0 1.5]
13	km_assim	K		Concentration of DOC for half max assimilation reaction from DOC to MIC	MIC	4×10^2	gCm^{-3}	[1 10^4]
14	km_decom	K		Concentration of SOC for half max assimilation reaction from SOC to DOC	MIC	5×10^4 , 6×10^5	gCm^{-3}	[10^4 10^9]
15	tau4l1	K		Turnover time of metabolic litter	CLM5	0.0541	year	[0 0.11]
16	tau4cwd	K		Turnover time of coarse woody debris	CLM5	3.33	year	[1 6]
17	tau4l2	K		Turnover time of cellulose and lignin litter	CLM5	0.2041	year	[0.1 0.3]
18	w-scaling	ξ	Environmental modifiers	Scaling factor to soil water scalar	CLM5	1	unitless	[0 5]
19	q10	ξ		Temperature sensitivity	CLM5	1.5	unitless	[1.2 3]
20	efolding	ξ		E-folding parameter to calculate depth scalar	CLM5	10	metre	[0 1]
21	cryo	V	Vertical transport	Cryoturbation rate	CLM5	0.0005	m^2yr^{-1}	[3×10^{-5} 16×10^{-4}]
22	diffus	V		Bioturbation rate	CLM5	0.0001	m^2yr^{-1}	[3×10^{-5} 5×10^{-4}]
23	beta	I	Carbon input allocatoin	Beta parameter controlling vertical distribution of carbon input to litter pools	CLM5	PFT dependent	unitless	[0.5 1] (CLM5)

2. Correlation vs. causation. The authors do spend most of their time - in the Results - on the correlation side of the line, which is appreciated and appropriate given the observational

nature of the work. However, the statistical analysis of the meta-analysis data are woefully shallow and inadequate given your recognition that CUE and SOC are outcomes of multiple causative predictors. Put simply, you recognize correlation and mention it, but do not perform the necessary data analytical steps to build confidence in your analysis of the relationships presented. The univariate correlations in Figs. 2a and 2b should be replaced with plots of the partial (or conditional) coefficients from the mixed effects models, so that the relationships shown are corrected for the influence of other predictors (e.g., both depth and MAT for a). Further, you need to present results from analyses showing that your predictors are not correlated, and at least evaluate two-way interactions among the predictors. Currently, you have a single model shown in Extended Data Table 2 of unstandardized coefficients (without a SE), a cumulative R² (fixed + random), and no recognition that the predictors could be correlated and hence the possibility that the coefficients you present are spurious. Further, nor do you justify the use of both random slopes and random intercepts. Please revise this analysis to build credibility in your findings. When I analyzed your data given in Extended Data Table 1, I will note that your general findings are supported and your predictors appear uncorrelated in the various models I ran. For example, for SOC, using random intercepts and common slopes, the beta coefficients reveal a strong negative MAT effect on SOC, a weaker but still positive CUE effect on SOC, and then a strong negative MAT by CUE interaction. The **variance inflation factors** for the main effects model were all low and for the model with the pairwise interactions, fixed effect variance explained was 17.6% with a further ~60% of the variance explained by the “study id” random effect. As such, use a regular linear model (without random effects), the results were very different, revealing strong within-study associations in these data that should be mentioned. My take home is that the results from your meta-analysis will be bolstered by doing your due diligence in terms of the mixed-effects modeling. However, as currently presented, this part of your analysis falls far below publication standard.

R1.2 We appreciate referee #1’s constructive comments and suggestions on the mixed-effects modeling. Following your suggestion, we used mixed-effects models to explore CUE-SOC relationship in both the meta-analysis and data assimilation results by the microbial model and CLM5.

In the meta-analysis, we took CUE, mean annual air temperature (MAT) and depth of the measurement as the fixed effects to SOC content, as we expected these factors to have an effect. We then added the study sources (“Source” in Extended Data Table 1, n = 16) as the

random effect to acknowledge that different studies have used different methodologies to estimate CUE. We explored different structures of the mixed-effects model (i.e., random intercepts with common slopes or random intercepts with random slopes) to test the CUE-SOC relationship. Among all the fixed effect variables (i.e., CUE, MAT, and soil depth), only the correlation between CUE and MAT is significant at the significance level of 0.05 (Pearson correlation coefficient = -0.24, $P = 0.005$, $df = 130$). Therefore, the interaction between CUE and MAT was also considered in the mixed-effects model. Only results of model structures that converged in regressions were presented in Response Letter Table 2. We found that mixed-effects models with different structures all showed a positive relationship between CUE and SOC in the meta-analysis (Response Letter Fig. 6a and Response Letter Table 2). Furthermore, we found that the variance inflation factors for the main-effects predictors were all low. Thus, including the interaction term between CUE and MAT did not significantly improve the prediction power of the mixed-effects model. Accordingly, we did not include in the main text the interaction term between CUE and MAT in the mixed-effects model, but considered random intercepts with common slopes in regression.

In addition, we explored how CUE correlates with microbial biomass and non-microbial biomass carbon. We found that CUE is not only positively correlated with microbial biomass carbon, but also non-microbial biomass carbon (Response Letter Table 4). Such a pattern supports the argument that microbial partitioning of carbon toward microbial growth will enhance SOC accumulation via microbial by-products and necromass.

We applied a similar mixed-effects model to explore CUE-SOC relationship using the data assimilation results from both CLM5 and the microbial model. We set CUE as the fixed effects on SOC storage and treated the climate types that each soil profile belongs to as the random effects. The mixed-effects model considered random intercepts with common slopes in regression. We found similar positive CUE-SOC relationships by both the microbial model (Response Letter Fig. 6b) and CLM5 (Response Letter Fig. 6c-d). Results from the microbial model confirmed again the concurrence of high CUE and accumulation of both microbial and non-microbial organic carbon (Response Letter Table 5).

Moreover, as the referee suggested, it is critical to consider within-study or -ecoregion association of the data when explaining CUE-SOC relationship. The explained variance in SOC was only 37% by the fixed-effects but increased to 55% after considering the random effects in the meta-analysis (Response Letter Table 2). And the similar patterns existed in the regression results by CLM5 and the microbial model (Response Letter Table 3).

In the revised manuscript, we added the results of the mixed-effects modeling analysis in L97 - L104 and L423 - L439. Fig. 2 in the main text of the original manuscript has been changed to be the same with Response Letter Fig. 6 shown here. Response Letter Tables 2 - 5 are presented in Extended Data Tables 2 - 5 in the revised manuscript. All the statistics shown in Fig. 2 are from the results of the mixed-effects modeling.

Response Letter Fig. 6 | CUE-SOC relationship emerged from the meta-analysis (a), microbial model data assimilation with the subset of 1,000 vertical profiles (b), CLM5 data assimilation with the subset of 1,000 vertical profiles (c), and CLM5 data assimilation with all the 52,819 profiles (d). The subset of vertical profiles was used because the computational cost for the microbial model is too expensive (i.e., ~2,600 times more expensive than that for CLM5) to do data assimilation with all the 52,819 profiles. Panels c and d are presented for comparison of results between the subset and whole set of the vertical profiles. Black lines and statistics shown in the figure are the partial coefficients from linear mixed-effects model regression (see Response Letter 2 – 3 for details).

Response Letter Table 2 | Unstandardized coefficients of CUE-SOC relationship in the mixed-effects model with meta-analysis data. CUE, depth and mean annual temperature (MAT) were set as fixed effects to both logarithmic and original SOC content. The study source was set as the random effects. We explored different structures of the mixed-effects models (i.e., random intercepts with common slopes or random intercepts with random slopes) to test CUE-SOC relationship. Interaction between CUE and MAT was also considered. Only results of model structures that converged are presented. The total observation size $n_{obs} = 132$; the random effects size $n_{study} = 16$. The variance inflation factors for the main effects model were all low.

		Intercept	CUE	Depth	MAT	MAT*CUE
log10(SOC)~CUE + Depth + MAT + (1 Study Source) variance explained by mixed model: 55%, by fixed effect: 37%						
Fixed Effects	Estimates	1.16	0.76	-0.020	0.013	NA
	Std. Error	0.11	0.18	0.0034	0.0053	NA
	t value	10.58	4.14	-5.93	2.49	NA
	P	<0.0001	<0.0001	<0.0001	0.014	NA
Random Effects	Standard Deviation	0.16	NA	NA	NA	NA
log10(SOC)~CUE + Depth + MAT + MAT * CUE + (1 Study Source) variance explained by mixed model: 55%, by fixed effect: 38%						
Fixed Effects	Estimates	1.17	0.72	-0.020	0.012	0.0042
	Std. Error	0.17	0.43	0.0034	0.014	0.039
	t value	6.78	1.65	-5.90	0.90	0.11
	P	<0.0001	0.10	<0.0001	0.37	0.91
Random Effects	Standard Deviation	0.17	NA	NA	NA	NA
log10(SOC)~CUE + Depth + MAT + MAT * CUE + (CUE Study Source) variance explained by mixed model: 49%, by fixed effect: 20%						
Fixed Effects	Estimates	1.20	0.35	-0.020	0.0086	0.040
	Std. Error	0.15	0.49	0.0031	0.011	0.037
	t value	7.79	0.72	-6.46	0.75	1.10
	P	<0.0001	0.48	<0.0001	0.46	0.28
Random Effects	Standard Deviation	0.032	0.77	NA	NA	NA
SOC~CUE + Depth + MAT + (1 Study Source) variance explained by mixed model: 49%, by fixed effect: 20%						
Fixed Effects	Estimates	17.30	55.49	-0.65	0.13	NA
	Std. Error	7.62	12.60	0.23	0.37	NA
	t value	2.27	4.41	-2.84	0.36	NA
	P	0.026	<0.0001	0.0052	0.72	NA
Random Effects	Standard Deviation	12.74	NA	NA	NA	NA
SOC~CUE + Depth + MAT + (CUE Study Source) variance explained by mixed model: 49%, by fixed effect: 20%						

Fixed Effects	Estimates	9.40	57.50	-0.65	0.77	NA
	Std. Error	6.70	19.80	0.20	0.35	NA
	t value	1.40	2.91	-3.16	2.19	NA
	P	0.16	0.012	0.0020	0.030	NA
Random Effects	Standard Deviation	5.00	55.92	NA	NA	NA
SOC ~ CUE + Depth + MAT + MAT * CUE + (1 Study Source) variance explained by mixed model: 49%, by fixed effect: 20%						
Fixed Effects	Estimates	12.67	69.39	-0.65	0.57	-1.38
	Std. Error	11.80	29.56	0.23	0.92	2.65
	t value	1.07	2.35	-2.85	0.62	-0.52
	P	0.29	0.020	0.0051	0.54	0.60
Random Effects	Standard Deviation	12.55	NA	NA	NA	NA

Response Letter Table 3 | Unstandardized coefficients of CUE-SOC relationship in the mixed-effects model with data assimilation results. CUE was set as the fixed effects to logarithmic SOC content. Climate types that soil profiles belong to were set as the random effect. We applied a mixed-effects model that considered random intercepts with common slopes to test CUE-SOC relationship (i.e., $\log_{10}(\text{SOC}) \sim \text{CUE} + (1|\text{Climate Types})$). The random effects size $n_{\text{climate}} = 12$. The total observation size for each regression was reported in the table.

		Intercept	CUE
Microbial model with representative profiles, $n_{\text{obs}} = 983$ Variance explained by mixed model: 51%, by fixed effect: 3%			
Fixed Effects	Estimates	1.02	0.79
	Std. Error	0.076	0.071
	t value	13.33	11.01
	P	<0.0001	<0.0001
Random Effects	Standard Deviation	0.25	NA
CLM5 with representative profiles, $n_{\text{obs}} = 983$ Variance explained by mixed model: 61%, by fixed effect: 17%			
Fixed Effects	Estimates	-0.10	3.48
	Std. Error	0.11	0.21
	t value	-0.96	16.36
	P	0.34	<0.0001
Random Effects	Standard Deviation	0.24	NA
CLM5 with all profiles, $n_{\text{obs}} = 52280$ Variance explained by mixed model: 51%, by fixed effect: 27%			
Fixed Effects	Estimates	0.56	4.62
	Std. Error	0.068	0.029
	t value	-8.21	161.72
	P	<0.0001	<0.0001
Random Effects	Standard Deviation	0.23	NA

Response Letter Table 4 | Unstandardized coefficients of relationships of CUE with microbial and non-microbial biomass in the mixed-effects model with meta-analysis data. CUE, depth and mean annual temperature (MAT) were set as the fixed effects to microbial and non-microbial biomass (i.e., total SOC minus microbial biomass carbon) carbon content. The study source was set as the random effect. We set random intercepts with common slopes in regression (i.e., (non)microbial biomass~CUE + MAT + Depth + (1|Climate Types)). The total observation size $n_{sample} = 62$; the random effects size $n_{study} = 9$.

		Intercept	CUE	Depth	MAT
Microbial biomass(mgC kg⁻¹)~CUE + Depth + MAT + (1 Study Source) variance explained by mixed model: 63%					
Fixed Effects	Estimates	0.79	2.01	-0.011	-0.038
	Std. Error	0.32	0.58	0.0080	0.014
	t value	2.48	3.47	-1.37	-2.66
	P	0.018	0.0011	0.18	0.010
Random Effects	Standard Deviation	0.34	NA	NA	NA
Nonmicrobial biomass(gC kg⁻¹)~CUE + Depth + MAT + (1 Study Source) variance explained by mixed model: 67%					
Fixed Effects	Estimates	31.68	60.56	-0.45	-1.42
	Std. Error	10.16	18.21	0.25	0.45
	t value	3.12	3.33	-1.80	-3.14
	P	0.0034	0.0016	0.077	0.0027
Random Effects	Standard Deviation	11.39	NA	NA	NA

Response Letter Table 5 | Unstandardized coefficients of relationships of CUE with microbial and non-microbial biomass in the mixed-effects model with data assimilation results. CUE was set as the fixed effects to microbial and non-microbial biomass (i.e., total SOC minus microbial biomass carbon) carbon content. Climate types that soil profiles belong to were set as the random effect. We applied a mixed-effects model that considered random intercepts with common slopes in regression (i.e., (non)microbial biomass~CUE + (1|Climate Types)). The total observation size $n_{sample} = 983$, the random effects size $n_{climate} = 12$. The total observation size for each regression was reported in the table.

		Intercept	CUE
Microbial biomass(mgC kg⁻¹)~CUE + (1 Study Source) variance explained by mixed model: 11%			
Fixed Effects	Estimates	0.011	0.75

	Std. Error	0.056	0.14
	t value	0.21	5.55
	P	0.84	<0.0001
Random Effects	Standard Deviation	0.13	NA
Nonmicrobial biomass(gC kg⁻¹)~CUE + (1 Study Source) variance explained by mixed model: 22%			
	Estimates	11.51	54.86
	Std. Error	5.20	8.96
	t value	2.21	6.13
	P	0.041	<0.0001
Random Effects	Standard Deviation	15.41	NA

3. The remainder of my concerns are less consequential but I think important to address nonetheless. They are as follows, (a) Space-for-time. Your work uses spatial, observational data to build knowledge but in numerous instances you translate the significance for understanding change in time. Acknowledgement that you are making inferential jumps here by using space-for-time substitutions in terms of inferring process would be appreciated.

R1.3a We thank the referee for pointing it out. We revised the related sentences (e.g., L100 - L104, L217 - L220) to avoid confusion on space-for-time substitution.

(b) The main paper comes across as selectively choosing results that “agree” with the findings you present. For example, lines 202-205 talk about data that are consistent with a positive CUE-SOC relationship and you also cite evidence where CUE and MAT are negatively correlated. Please acknowledge and include some discussion to resolve other observational syntheses combined with modeling work that find the opposite (e.g., Ye et al. 2020 in *Global Biogeochemical Cycles*, <https://agupubs.onlinelibrary.wiley.com/doi/10.1029/2019GB006507>). It would be important for the reader to understand how your work therefore fits into the wider picture painted by the literature on regional to global observational data-model syntheses related to this topic.

R1.3b We thank the referee for making this great point. The negative relationship between CUE and MAT may not be general when the resource availability (e.g., nitrogen and

phosphorus) or exoenzymatic activities were considered^{13,14}. To avoid the potential confusion, we no longer present the MAT-CUE relationship in the revised manuscript.

(c) You refer repeatedly (e.g., lines 176-177) to the five mechanisms in CLM5 that influence SOC turnover and the fact that CUE emerges as the one to which spatial patterns are most sensitive and best predicted by. However, looking into those mechanisms, they are less directly coupled to SOC stocks than CUE because they control how much is going into litter pools or how fast SOC decomposes (which itself is dependent on the size of the SOC pool), but they are not the process/ parameter that actually dictates allocation of inputs to SOC. As such, please address in the main paper the extent to which this close coupling of CUE to SOC formation rates essentially predetermines your finding that it is the mechanism most closely matched to storage in CLM5.

R1.3c We greatly appreciate the referee for the very insightful point. From a perspective of modelling, there is no such a priori expectation that CUE should be a stronger predictor than others. Equation 2 of the manuscript showed that all the six mechanisms control the SOC storage at steady state. Conceptually, the role of CUE where it dictates the allocation of input carbon to SOC derives from its definition with no pre-assumption about its importance to SOC as a priori either. In this study, we used the PRODA approach with a process model (i.e., CLM5) and SOC data to reveal that CUE is the most important mechanism to global SOC storage. Nevertheless, we modified our text to reflect this point (e.g., L215 - L217).

(d) Coming back to right where I started, with regards to broader questions around how we do science and what we can learn (e.g., correlation vs. causation), I think it would be very helpful to explain in the main paper (i.e., in the paragraph starting line 110) the philosophy that underpins the PRODA tool. Although many new data science tools are exciting, they can also easily be misused, leading to spurious findings that are only revealed once the philosophy underpinning the approach is dug into. Given that most of the readership will be unfamiliar with the tools applied, and hence what they can do in terms of identifying mechanism (as opposed to, say, being optimized for outcome prediction; think of such things as IC-based model selection), I think it would be important to spend a handful of sentences explaining the philosophy of the approach.

R1.3d We thank the referee for the suggestions. PRODA has two components: data assimilation to constrain parameters at each of the sites and deep learning to find optimized parameterization over the globe. The underlying argument for this approach is that biogeochemical processes, such as decomposition and carbon partitioning between respiratory carbon release and microbial growth, vary from one site to another over space. The current generation of ESMs assumes most of the rates are constant over the globe. This is potentially one of the major sources of uncertainties in the predictions by ESMs as argued by Luo and Schurr (2020)¹⁵. Meanwhile, traditional machine learning techniques are specialised in identifying patterns from big databases. It is, however, impossible for the traditional machine learning techniques to identify mechanisms behind the data. In comparison, Bayesian data assimilation techniques (e.g., MCMC method) are capable of inferring causal understanding by integrating data with mechanistic models, yet they are in short of mining big data to get global patterns. PRODA is such a tool that combines data assimilation with machine learning, trying to obtain both global patterns from big data and the mechanistic understanding underlying such patterns.

We highlighted the philosophy underlying the PRODA in L156 - L165 of the revised manuscript. We hope the added sentences helped make it clearer about our method and its potential applications.

References

- 1 Allison, S. D., Wallenstein, M. D. & Bradford, M. A. Soil-carbon response to warming dependent on microbial physiology. *Nature Geoscience* **3**, 336-340 (2010).
- 2 Sinsabaugh, R. L., Shah, J. J. F., Findlay, S. G., Kuehn, K. A. & Moorhead, D. L. Scaling microbial biomass, metabolism and resource supply. *Biogeochemistry* **122**, 175-190 (2015).
- 3 Allison, S. D. Building predictive models for diverse microbial communities in soil. *Microbial biomass: A paradigm shift in terrestrial biogeochemistry*, 141-166 (2017).
- 4 Duan, Q., Gupta, V. K. & Sorooshian, S. Shuffled complex evolution approach for effective and efficient global minimization. *Journal of optimization theory and applications* **76**, 501-521 (1993).
- 5 Xu, X., Thornton, P. E. & Post, W. M. A global analysis of soil microbial biomass carbon, nitrogen and phosphorus in terrestrial ecosystems. *Global Ecology and Biogeography* **22**, 737-749 (2013).

- 6 Neff, J. C. & Asner, G. P. Dissolved organic carbon in terrestrial ecosystems: synthesis and a model. *Ecosystems* **4**, 29-48 (2001).
- 7 Guo, Z. *et al.* Soil dissolved organic carbon in terrestrial ecosystems: Global budget, spatial distribution and controls. *Global Ecology and Biogeography* **29**, 2159-2175 (2020).
- 8 Wilson, C. H. & Gerber, S. Theoretical insights from upscaling Michaelis–Menten microbial dynamics in biogeochemical models: a dimensionless approach. *Biogeosciences* **18**, 5669-5679 (2021).
- 9 Zhang, D., Hui, D., Luo, Y. & Zhou, G. Rates of litter decomposition in terrestrial ecosystems: global patterns and controlling factors. *Journal of Plant Ecology* **1**, 85-93 (2008).
- 10 Xu, X. *et al.* Soil properties control decomposition of soil organic carbon: Results from data-assimilation analysis. *Geoderma* **262**, 235-242 (2016).
- 11 Schädel, C. *et al.* Circumpolar assessment of permafrost C quality and its vulnerability over time using long-term incubation data. *Global change biology* **20**, 641-652 (2014).
- 12 Cai, A. *et al.* Long-term straw decomposition in agro-ecosystems described by a unified three-exponentiation equation with thermal time. *Science of the Total Environment* **636**, 699-708 (2018).
- 13 Ye, J. S., Bradford, M. A., Dacal, M., Maestre, F. T. & García-Palacios, P. Increasing microbial carbon use efficiency with warming predicts soil heterotrophic respiration globally. *Global Change Biology* **25**, 3354-3364 (2019).
- 14 Sinsabaugh, R. L. *et al.* Stoichiometry of microbial carbon use efficiency in soils. *Ecological Monographs* **86**, 172-189 (2016).
- 15 Luo, Y. & Schuur, E. A. Model parameterization to represent processes at unresolved scales and changing properties of evolving systems. *Global change biology* **26**, 1109-1117 (2020).

Referee #2 (Remarks to the Author):

See my comments in the attached.

This study proposes to use global and publicly available datasets complemented by a meta-analysis to draw conclusions on the potential drivers of global soil organic carbon (SOC) storage. The statistical modelling of SOC storage was made by a “PROcess-guided deep learning and DATA-driven modelling”, which, to summarize, aim to use a deep learning model to estimate the parameters of a process-based biochemical model.

Response: We deeply appreciate this referee for the great assessment of this study.

Overall I found this study of potential interest, but I also raise substantial concerns on specific aspects of the mapping, validation, and interpretation process. I am however positive that the authors could address these concerns by a revision and additional analyses, and that the revised manuscript could be a useful contribution.

Response: We thank this referee for the constructive suggestions. We have addressed all the concerns as described below.

Cross-validation

The cross-validation should be used to estimate the validation statistics (as is done), but for prediction and uncertainty quantification a model fitted with all the data should be used. In this paper, the authors used 10 models, resulting from fitting 10 models on the 10 cross-validation folds, and use the mean value from the 10 model predictions as their final prediction. They also use the standard deviation of the 10 predictions as an estimate of prediction uncertainty.

- - **Prediction** The final model for mapping should be calibrated with all the available observations. This is the “best guess model”, calibrated with all data available. Map accuracy is estimated previously, using the 10-fold random cross-validation.

R2.1a We are thankful to this referee for the great suggestion. We now use the best-guess model to show the results in Fig. 3 of the revised manuscript (also presented here Response Letter Fig. 7). Spatial patterns of SOC and its underlying mechanisms did not change much compared to the results presented in the original manuscript. The best guess model explains

57% of the spatial variation in SOC data, and estimates a global total SOC stock of 2147 Pg C.

We revised the manuscript (e.g., L166 - L167) to include the new results as shown in Fig. 3 (also presented here as Response Letter Fig. 7). We also clarified the use of the best-guess model to present our results in L699.

Redacted

- - **Prediction uncertainty** Prediction uncertainty taken as the standard deviation of 10 model predictions is not correct. There are several options to estimate the prediction uncertainty. The authors could use bootstrapping and obtain confidence intervals. Note that bootstrapping requires fitting at least 100 models to get a realistic estimate. Another option is to use quantile regression. Currently, the prediction uncertainty maps are a standard deviation value obtained from 10 predicted values. This is too little to obtain a reliable estimate of the standard deviation and, again, not a correct way to obtain a valid estimate of prediction uncertainty. Why do the authors not propagate the uncertainty of the parameters as obtained by the posterior parameter distribution in the Bayesian analysis? There is a great opportunity here with the

Bayesian analysis of the parameters and the distribution obtained by the posterior distribution. For the uncertainty, ideally, prediction intervals are reported. When obtaining prediction intervals is not possible, the authors could always obtain a confidence interval by bootstrapping.

R2.1b We thank the referee for the suggestion. Using a neural network to propagate the uncertainty of the parameters in the Bayesian data assimilation is a great idea that is worth being explored in the future. In the revised manuscript, we applied a bootstrapping of the neural network for 200 times to retrieve the confidence intervals of predictions and estimates. Results in Fig. 4 (Response Letter Fig. 8) and Extended Data Fig. 12 (Response Letter Fig. 9) are now all from the results of 200-time bootstrapping. The results from bootstrapping indicate our neural network is robust in predictions. The neural network model in bootstrapping explains a median of 56.3% of the variation of SOC data (one-sigma confidence interval: 53.6 - 57.3%) and estimates a median global total SOC stock of 2192 Pg (one-sigma confidence interval: 2100 - 2314 Pg).

In the revised manuscript, we introduced the bootstrapping method in L688 - L696 and L701 - L706. Fig. 4 of the main text and Extended Data Fig. 12 were substituted by the figures below (i.e., Response Letter Figs. 8 and 9, respectively).

Response Letter Fig. 8 | CUE as the main regulator of global SOC storage. Results were obtained from 200-time bootstrapping. Points/Lines show the median results. Error bars/shaded areas are the 1-sigma uncertainty (i.e., 68% confidence interval).

Redacted

Permutation importance

The permutation analysis suffers from the same problem of using 10 models. The authors report grouped permutation variable importance values from 10 models, with error bar representing the standard deviation obtained by cross-validation. This is not common, and I would say not correct. Authors should compute the permutation importance of a single model, the one calibrated using all data. This way they are sure that the permutation values indeed correspond to the model they are using for mapping. Since the permutation values are not additive, I am afraid that Extended Data Fig. 7 is not realistic. Also, error bar usually represents the permutation error. Since permutation involves randomness, it is important to repeating the permutation many times and also show the 5% and 95% quantile of importance values from the repetitions. The permutation error obtained this way is likely to appear much larger than that currently reported.

R2.2a We thank referee #2 for the suggestions. Following the suggestions, we applied 1000 times of permutation to the best guess model to assess the importance of environmental variables to different mechanisms. Now the results are shown in Response Letter Fig. 10. The uncertainty range, as suggested by the referee, is much larger than the previous version, but the conclusion on the importance of environmental variables to CUE did not change. Soil physical properties are still the most important ones in predicting the spatial variability of CUE.

In the revised manuscript, Extended Data Fig. 11 was updated as in Response Letter Fig. 10.

Response Letter Fig. 10 | Importance of environmental variables to different mechanisms. Results showed the median values from 1000-time permutations to the best-guess model. Error bars showed the 1-sigma uncertainty (68% confidence interval).

The MSE values obtained by permutation importance are not additive. This means that the permutation importance values for each individual covariate cannot be simply added to another one to calculate the permutation importance of the group. To compute the group permutation importance, the authors should make a permutation directly on the group of covariates. The results are likely to be different, because permutation importance is a method that is sensitive to dependence among covariates. Authors should make sure that they permuted the group of covariates simultaneously, not the individual covariates and then summed the individual permutation importance values. This is very unclear from the text at L. 663-668.

R2.2b Thanks to the referee for pointing this out. We did as Referee #2 suggested but failed to express the method sufficiently in our original manuscript. Briefly, we used the best-guess neural network that was calibrated with all available data to do the permutation test. For each category of environmental variables, we permuted their original information with uniformly distributed random values for 1,000 times. After each permutation, we used the best-guess neural network to predict parameter values respectively from original environmental information (i.e., without permutation) and from permuted values. We used the mean squared error (i.e., $MSE = \frac{\sum_{i=1}^N [\sum_{j \in M} (para_{i,NN}^j - para_{i,DA}^j)^2]}{N}$), where j refers to mechanisms, i to soil profiles, $para_{NN}$ to neural network predicted parameter value, and $para_{DA}$ to parameter values estimated from data assimilation) to account for the prediction deviation from profile-level (i) optimized parameter values. Parameters belonging to the same mechanisms ($j \in M$) were grouped together in the calculation. The permutation importance (PI) of environmental variables in category k ($cate_k$) to mechanism M was expressed as:

$$PI_{cate_k \text{ to } M} = \frac{\sum_{i=1}^N \left[\sum_{j \in M} (para_{i,permu\ NN}^j - para_{i,DA}^j)^2 \right]}{\sum_{i=1}^N \left[\sum_{j \in M} (para_{i,no\ permu\ NN}^j - para_{i,DA}^j)^2 \right]} \quad (8)$$

The PI values represent the increase of inaccuracy in the neural network prediction caused by the absent information of environmental variables in category k . Thus, a larger value of permutation importance indicated a greater importance of an environmental variable to the prediction of parameters.

After applying the 1000-time permutation, the main conclusion about variables' importance to CUE did not change. The physical features of the soil are still the most important to CUE. We revised the description in the L770 – L794 in the manuscript.

Statistical validation of maps

L. 579-590: There is a current hype about the idea that spatial autocorrelation should be accounted for when estimating validation statistics in a spatial context. This is a misconception and I urge the authors not to propagate this wrong idea in a scientific paper. It is good that they only tested this but did not include the results of spatial cross-validation in their analysis. In short, spatial autocorrelation does influence the estimation of map accuracy statistics, but spatial cross-validation is clearly not an answer to this problem. The only solution to obtain an unbiased estimate of map accuracy is to collect an additional probability sample from the population (that is, the world) and to use a design-based estimation of the

statistics. I acknowledge this is not feasible for this kind of global study, or very difficult to implement. But spatial cross-validation is not the solution. It provides overpessimistic validation statistics and has no underlying theory. Why did the authors not use instead a model-based estimation of map accuracy? Why didn't they use a heuristic method based on spatial weighting? There are many methods available for dealing with estimation of map accuracy in case of clustered data, but clearly spatial cross-validation is the worst. It is important that this paper does not propagate these recent misconceptions of statistical validation of maps. I recommend to simply remove this paragraph.

R2.3 Thanks to the referee for the suggestion. We removed the autocorrelation part from the manuscript as suggested.

Additional minor comments:

There are a few sentences that are difficult to read, such that at L. 76-77, for example. The manuscript would benefit from shorter sentences.

R2.4a Thanks for the suggestion. We revised the referred sentence and other sentences that may be difficult to read (e.g., L75- L84). Hopefully the new version of the manuscript is easier to read.

L. 87-90: This sentence suggests that deep learning IS a hypothesis. Perhaps adding “with” or “using” would clarify this point.

R2.4b Thanks for the suggestion. We revised the sentence in L87 - 90 as “*Here, we examined the relationship between CUE and the preservation of carbon as SOC using a combination of global-scale datasets, Earth system models (ESMs), deep learning (i.e., multilayer neural network), and meta-analysis, in light of interactions with climate, vegetation and edaphic properties*”. Hope the new version of the manuscript is clearer.

L. 91-93: Consider rephrasing, this sentence is quite unclear: “We collated 132 pairs of data sets from 16 experimental studies *...+ where SOC content were measured at 46 locations”: what is collected? The 132 pairs of data sets or the 46 measurement at locations? AHA this is now clear at L. 405-406.

R2.4c Thanks for the suggestion. We revised the sentence (L97 - L99) as “*we first collated 132 pairs of community-level CUE and SOC content data measured at 46 locations across continents from 16 experimental studies previously published in the peer-reviewed literature*” to make it clearer.

L. 415: How are the coarse fragments obtained? I think WoSIS has only few samples containing coarse fragment data.

R2.4d We thank the referee for the question and comment. We did not apply fragment information directly to the original data before data assimilation. Instead, we calculated total SOC stock by using the coarse fragment map from SoilGrids. We clarified this point in the new version (L445 - L448).

L. 418-419: How many observations were used to fit the pedotransfer function and to obtain the regression parameters for bulk density?

R2.4e In our study, 78,913 soil layers from 16,248 profiles that simultaneously recorded bulk density and SOC content were used in the pedo-transfer function to obtain the regressed parameters. The pedo-transfer function eventually explained 54.9% of variation of the bulk density. We also added this information in the revised manuscript (L451 - L454).

L. 447-460: This description is quite unclear. The authors used what is commonly referred to as a Nash-Sutcliffe modelling efficiency, that some would call the coefficient of determination (of the 1:1 line). I agree with their use of the term “coefficient of efficiency”. Several authors (e.g. Janssen & Heuberger, 1995) call it a modelling efficiency. But the description that the authors make of this statistic is a bit surprising. A modelling efficiency of 0.75 is very high. It means that 75% of the SOC depth distribution variance is explained by CLM5. The authors should also consider rephrasing this paragraph and make shorter sentences.

R2.4f Thanks to the referee for pointing this out. The modeling efficiency we used here is a site-level modeling efficiency. It could be very high because we were using 21 parameters in the CLM5 model to fit the vertical SOC profiles. We further clarified this point in L490 – L492 and cited the reference that the referee mentioned in the revised manuscript.

L. 557-560: Calling deep learning a neural network with four hidden layers and 256, 512, 512, and 256 neurons, respectively, is a bit misleading. I agree that it is not clear when a neural network becomes “deep”, but we usually agree that this is a complex model with many layers/neurons.

R2.4g Thanks for the suggestion. We changed the “deep learning model” to “multilayer neural network model” in the revised manuscript (e.g., L89 and L163).

L. 563-564: The average in the MSE equation is missing.

R2.4h Thanks for pointing out this typo. We corrected this typo in the revised manuscript in L784.

L. 649: How many permutations were made. The number of permutations should be sufficiently high to obtain reliable results.

R2.4i Thanks to the referee for the question and comment. Originally this number was 100. In the revised version we conducted 1,000 permutations to the best guess model. We reported this number in L781 of the revised manuscript.

Reference:

Janssen, P. H. M., & Heuberger, P. S. C. (1995). Calibration of process-oriented models. *Ecological Modelling*, 83(1-2), 55-66.

Referee #3 (Remarks to the Author):

This is an interesting study that aims to show the importance of carbon use efficiency for soil carbon storage. The authors leverage a literature synthesis of CUE measurements, 50K+ WoSIS soil profiles with SOC measurements, the CLM5 biogeochemical model, and their PRODA approach to estimate CUE for the soil profiles and globally. What stands out most is the computational framework, which will be an important tool for ongoing and future data-model integration. I also agree that better constraining CUE in models – especially given the interpretability of CUE across different types of measurements and model formulations – is crucial. However, there are several conceptual and practical considerations that compromise the present study and the novelty of its purported findings. I outline these main concerns below.

Response: We greatly appreciate referee #3 for the generally positive assessment of this study. We carefully addressed all the issues raised by referee #3 as described below.

1. I am not convinced that the main conclusions are particularly novel.

– The finding that CUE and SOC have a positive relationship has been previously demonstrated (Kallenbach et al., 2015; Malik et al., 2018; Buckeridge et al., 2020) and is largely expected. CUE by definition dictates how much carbon stays in the system versus how much leaves the system, especially here where the model-derived CUE is an emergent property of the whole system (rather than the fraction of C acquired that is specifically used for microbial growth and biosynthesis in the empirical measurements).

R3.1a We agree with the referee that “*CUE by definition dictates how much carbon stays in the system versus how much leaves the system*”, but the ‘system’ is—strictly speaking following the definition of CUE—microbial biomass. What we showed in this study, which fully supports previous studies as cited by the referee, is that CUE also promotes SOC accumulation, so the carbon storage in the whole soil system (please see detailed analyses in R1.2). In this sense, our results support the referee’s statement that CUE is “*an emergent property of the whole system (rather than the fraction of C acquired that is specifically used for microbial growth and biosynthesis in the empirical measurements)*”.

Moreover, there is an opposing hypothesis in the literature¹ based on CUE as a physiological trait of microorganisms, such that a high CUE could be translated toward a

high enzyme production so to enhance SOC decomposition and loss (Fig. 1b in the manuscript). Thus, it is not clear in the literature whether CUE is positively or negatively correlated with SOC influence SOC storage in the research community. Our use of both the microbial and CLM5 models in this revision allows the confrontation of the models with field data. Therefore, we think the framing of the question around CUE is not only defensible but also necessary.

We agree with the referee that our results strongly support CUE as being positively related to SOC accumulation, a finding that is consistent with empirical evidence by either regional data (e.g., Malik et al., 2018; Buckeridge et al., 2020) or microcosm experiments (e.g., Kallenbach et al., 2015). In this study, we resolved the controversy on the CUE-SOC relationship at global scale by analysing 132 experimental data sets from the literature and 52,819 vertical SOC profiles over the globe with two contrasting models (please see detailed analyses in R1.1).

More importantly, we identified CUE as the most important mechanism for determining global SOC storage among the six different mechanisms using the global SOC database. This finding potentially helps shift SOC research from a classic paradigm that focuses on the roles of plant carbon inputs and decomposition to a new paradigm that emphasizes microbial physiology in determining SOC accumulation. In the past, researchers have invested tremendous efforts to incorporate different mechanistic processes of soil carbon cycle in Earth system models (ESMs), with a hope of improving the representation of SOC storage. Yet huge gaps still exist between ESM simulated and observed SOC stocks. Our study suggests that microbial CUE is the most critical mechanism in determining global SOC storage. This finding could stimulate more research on microbial physiology and ecology in general and microbial CUE in particular so as to improve our predictive understanding of soil carbon cycle.

In the revised manuscript, we clarified the novel aspects of our study in L75 – L90 and L230 - L245.

– While the positive relationship between CUE and SOC may be expected, the slope and how it is affected by climate and vegetation type, etc., would be interesting to explore further. CUE is a dynamic property that is affected by many factors, including some of those treated as separate mechanisms herein. Understanding the processes shaping CUE is an important step in improving its representation in process-based models, but, other than Ext. Data Fig. 7,

this study does not yet provide clear insights on the processes leading to differences (and spread) in observed CUE.

R3.1b This referee highlights a very important issue here. In our analyses, the positive CUE-SOC relationship was supported at the global scale and in all different eco-regions with CLM5 (Response Letter Fig. 11). The absolute values of the regression slopes of the CUE-SOC relationship differ across gradients of soil textures, pH, climate types, and land use changes (Response Letter Fig. 11). We found that sandy soils present stronger effect size of CUE in relation to SOC than clay soils (Response Letter Fig. 11a). The slope of CUE-SOC relationship declines when the soil environment changes from acid to alkaline (Response Letter Fig. 11b). Meanwhile, SOC storage is more responsive to CUE in tropical than in boreal regions (Response Letter Fig. 11c). In terms of the influence of land use changes, agricultural and urban ecosystems generally show less sensitivity of SOC to CUE than other natural ecosystems (Response Letter Fig. 11d). These patterns emerged from our study imply varying impacts of different physical environments and human activities in shaping the relationship between CUE and SOC storage. Because of a lack of sufficient experimental data, currently we cannot build a rigorous benchmark from the field measurements to be compared with our results from data assimilation with CLM5. In the future, detailed understanding of the drivers of CUE-SOC relationship requires more empirical data and experiments.

We discussed the varying slopes of CUE-SOC relationship across different ecoregions in L131 – L135 of the revised manuscript. Response Letter Fig. 11 was presented as Extended Data Fig. 7 in the revised manuscript.

Response Letter Fig. 11 | Regression slopes of the CUE-SOC relationship across different ecoregions with CLM5. Results are the slopes of the linear regression of the CUE-SOC relationship in different ecoregions. Error bars are the standard errors from the linear regressions. Numbers above the bars are the dimensions of freedom (df) in linear regressions. Line types of the bar and error bar plot indicate the P values of the regression slopes.

– Furthermore, the CUE that is estimated from CLM5 is an emergent system value (including litter pools, it seems) which is quite different to those measured in soils (that also vary between different methods; Geyer et al. 2018). The slope of the CUE and SOC relationship looks significantly lower in the literature synthesis compared to the global estimates... why? What does this mean and what drives this slope?

R3.1c The referee asked a great question. This question indeed is very important and interesting. We acknowledge that the absolute values of the slopes of CUE-SOC relationship differed in meta-analysis and data assimilation results. As the referee pointed out, CUE measured by different methods also differed in different field experiments. The differences in the absolute values of the slope of CUE-SOC relationship likely derived from the methodological differences among different methods (i.e., model structures and measuring techniques) and other environmental factors (e.g., climate, edaphic, vegetation, and land use

history). Further studies are surely warranted on this issue. In the revised manuscript, we discussed this issue in L133 - L135.

If the authors used their same framework with SOC values from the CUE literature synthesis to estimate the corresponding CLM5-derived bulk CUEs, how would these compare to the measured ones?

R3.1d This referee made a great suggestion. In the revised manuscript, we used a similar mixed-effects model to explore CUE-SOC relationship (please see also our response to referee #1). We found consistent positive correlation between CUE and SOC. The detailed descriptions and statistics of the mixed-effects modelling can be found in R1.2

Fig. 2b/d with MAT is interesting (though also well-documented; Hagerty et al. 2014, Allison et al. 2010) and the slopes seem to somewhat agree between the literature synthesis and global estimates. How do these slopes compare to the temperature sensitivities reported in other studies?

R3.1e Again, this is a great question. As we explained in R1.3b, the relationship between MAT and CUE might not be a general pattern under different context^{13,14}. We no longer present this relationship in the manuscript.

2. The findings of CUE importance rely on the selected model (i.e., CLM5) and may not be robust.

The study is also framed around two alternative hypotheses that are not mutually exclusive and, ultimately, the current study does not appear equipped to truly address them.

— It would be helpful to include several global soil models, including process-based (i.e., microbial-explicit) models. The authors discuss microbial feedbacks extensively, but the approach they take is not appropriate to support their claims (more on this below).

R3.2a We thank Referee #3 for providing the great suggestion. We have included a microbial model in the revised manuscript. Please refer to our responses to referee #1 (R1.1) for the details of the microbial model and related results.

– The authors support their choice of CLM5 as a “process-oriented biogeochemical model... because it is depth resolved and expressed in matrix form” (L113). First, this is a little deceiving because CLM5 is not process-based, or at least does not include the processes outlined in the hypotheses. It does not include microbes, and hence the CUE is simply a fraction that stays in the system versus leaving the system when transferred between first-order pools. Second, the two reasons given for the selection of CLM5 alone as the backbone of their study are not convincing nor appropriate. There are several models that are depth-resolved and virtually any existing model can be written in matrix notation. Indeed, all first-order global models can be easily written in matrix notation and even process-based models can be linearized, as needed, near the steady-state (given that they invoke a steady-state assumption anyway).

R3.2b Thanks to the referee for pointing this out. We deleted the two justifications (i.e., depth-resolved and matrix representation) for using CLM5 and developed a microbial-explicit model for this study as suggested (please see R1.1 for more explanations). We hope these revisions are sufficient to address the referee’s concern.

– Regarding the framing of negative vs. positive CUE-SOC relationships, the authors propose two alternative hypotheses, but these two hypotheses are by no means mutually exclusive. Both schemes can, and do, occur in real life. The positive feedback of microbial biomass and enzyme production (as in Fig. 1b) could, in addition to catalyzing decomposition, promote mineral-organic associations through the increase in sorption of microbial necromass and byproducts. This should be discussed, and makes me wary of the current framing of these two simple pathways as conflicting hypotheses.

R3.2c We thank the referee for making another very interesting point. Theoretically, both the positive and negative CUE-SOC relationships could exist. But it is not clear which more likely dominates SOC dynamics in the real world. This is one of the motivations why we conducted this study to examine the two alternative relationships with data. We clarified this point in L91 - L96 and L105 - L125 of the revised manuscript.

– Furthermore, I am not sure that the alternative hypothesis of high CUE leading to loss of SOC truly makes sense. High CUE could drive more microbial biomass, and more

subsequent decomposition that passes through microbial biomass, but that could just result in a faster cycling system. It does not imply a decrease in storage. (It also doesn't imply young radiocarbon ages, as microbial recycling contributes to older carbon.) Yet, the authors state that microbial models that “simulate direct dependence of SOC loss on microbial biomass via enzyme activities... always generate a negative relationship between CUE and SOC” (L193). I do not believe that this is always the case and I would challenge the authors to demonstrate this. It would be interesting to try a microbial-explicit model in the same framework. The cited studies (e.g., Allison et al. 2010 and others) with simple microbial-explicit models show that microbial feedbacks can indeed lead to losses, but not that higher CUE (and all else equal) increases losses. I feel that they may be conflating trends in microbial biomass with a dependence on the value of CUE.

R3.2d We greatly appreciate the reasoning this referee made here. When using default parameter values in the microbial model by Allison et al. (2010), the CUE-SOC relationship is negative as illustrated in Response Letter Fig. 2. However, the CUE-SOC relationship can change from negative to positive when parameter values change (Response Letter Fig. 2).

When we assimilated the soil carbon data to constrain the microbial model, a positive CUE-SOC relationship emerged (please see R1.1 for more explanation). Although it is numerically possible to generate a negative CUE-SOC relationship, the soil data at the global scale offer observation-based evidence to support that CUE is positively correlated with SOC storage (Response Letter Fig. 4). We discussed this point in L111 - L122 of the revised manuscript.

3. Proportional changes in the selected ‘mechanisms’ are difficult to interpret in a meaningful way.

– When considering proportional changes in the selected mechanisms, what if the mechanisms or relative changes were normalized and based on the distribution of potential values? That is, an 8% change in one mechanism might constitute a small change within the potential range of values for that mechanism, whereas it might span the range of potential values for another mechanism. Instead, maybe values could be varied from their respective means to plus/minus one (or two) standard deviations. A supplemental figure depicting the distributions of all mechanisms (as with CUE in Ext. Data Fig. 5) would be helpful in that

regard; from the box plots in Fig. 3, it seems that some of the distributions are indeed wider than others.

R3.3a The referee raised a very important issue about parameter spaces. The distributions of potential values of these mechanisms define parameter spaces for the model. We need to explore the parameter spaces fully when we project future changes in SOC. However, this study evaluated relative importance of different mechanisms in determining SOC via parameter sensitivity analysis. The latter is different from the projection analysis using the full ranges of parameter spaces, which should be done in the future.

We also greatly appreciate the suggestion by the referee to generate a supplementary figure depicting the distributions of all mechanisms. We have seriously considered this suggestion. Currently we do not know the parameter space at each grid point. Spatial variation of parameter does not equal to the parameter space for one specific site. It is possible to generate such a figure in the future after we conduct some rigorous study.

We added one sentence in L217 - L220 (*“While the two sensitivity analyses evaluated the relative importance of the six mechanisms, projecting changes in SOC needs to use full distributions of potential values of these mechanisms in future research.”*) to clarify the point.

– The results also state that a 10% increase in SOC needs a 3% increase in CUE and a 7% decrease in the environmental modifier. The latter is a model construct and not very meaningful. How much does the overall ‘environmental modifier’ vary spatially in the best fit model? How much can it change under future conditions? It is not apparent what a 7% decrease in the environmental modifier really means with regards to temperature, moisture, etc. sensitivity, so it is unintuitive to interpret this.

R3.3b We agree with the referee that it is not intuitive to understand the meaning of a 7% decrease in the environmental modifier. As in most of the Earth system models, CLM5 defines the environmental modifier as response functions of SOC decomposition to changes in temperature and moisture. Parameters in these response functions usually have fixed values over the globe. The 7% decrease in the environmental modifier results from changes in these parameter values to the extent that environmental scalar decreases by 7%. We hope that this explanation helps. We also revised the method description in L762 – L767 to clarify this point.

Other comments:

- The “6 mechanisms” in this study are not really mechanisms per se. For example, “environmental modifier” is not a mechanism... I’d argue that neither is baseline decomposition, but rather it is a model construct of a suite of mechanisms.

R3.4a The referee is correct that all the “six mechanisms” we referred in the manuscript are ensembles of processes. For example, environmental modifier is an ensemble of all parameters related to regulations of soil organic carbon cycle by its physical environment. Similarly, the baseline decomposition is calculated from processes that are related to the intrinsic decomposition rate of different carbon pools. We calculated the integrated values of processes from the same category. These ensembles of processes are a useful way to describe the holistic system dynamics of SOC over the globe. We clarified this point in the revised manuscript L714 - L720.

- The following sentence in the abstract “Our findings support a hypothesis that high microbial CUE favors preservation of carbon as SOC instead of stimulation of soil respiration” seems trivial, by definition of CUE.

R3.4b We agree with the referee that high microbial CUE should favour preservation of carbon as SOC, in some cases by definition. However, previous studies, e.g., the microbial model by Allison et al. (2010), suggested a negative CUE-SOC relationship. This triggered discussions in the community on whether high CUE promotes SOC storage or loss (please see Response Letter Figure 2). One of the major contributions of this study is that we resolved the controversy on the CUE-SOC relationship. We found that high CUE promotes SOC storage at the global scale.

- The authors should consider mentioning more explicitly in the abstract that the ‘synthetic analysis’ of all of these profiles relies on a data-model integration (and specifically CLM5) to quantify the various “mechanisms”, as the majority of these mechanisms (e.g., carbon inputs, vertical transport, CUE) are not directly measured within the globally-distributed soil profiles. Simply saying “the retrieved CUE from the global SOC database” could be deceiving. The finding that CUE is “twice as important” may largely depend on the choice of model (i.e., CLM5), so this context is important.

R3.4c Thanks for the suggestion. We revised the sentence to be “*The retrieved CUE from the global SOC database using Community Land Model and deep learning is at least twice as important as ...*” in the abstract (L48 - L50).

- The depth findings are interesting, but I would like to see them supported empirically, not just using model-derived values. The mixed-effects model results suggest that depth is important for SOC, of course, but what about exploring depth effects on CUE. You give results from a mixed-effects model of SOC, but could also explore variation in CUE.

R3.4d Thanks to the referee for providing this valuable suggestion. We found that the depth has no significant correlation with measured CUE in our meta-analysis. This may be because most of the CUE in the meta-analysis was measured in the surface soil (0 – 30cm). Our results in Response Letter Fig. 4 demonstrated the difference in regulation of CUE in both surface and subsurface (deeper than 30cm) soils. In the future, more studies of microbial processes are needed in the subsurface soil to fully address this issue.

- Furthermore, with regards to depth in Ext. Data Fig. 4, how can we interpret these findings in the context of studies that have shown that similar amounts (or at times more) of SOC is microbially-derived at depth compared to the surface? Microbial recycling plays an important role at depth. This could be discussed further.

R3.4e The referee made a great point that microbial recycling plays an important role at depth. Our results as shown in Response letter Fig. 4 indicate that the explanatory power of microbial CUE decreases with depth for SOC. It is possible that microbial recycling is a different concept from (maybe a much broader one than) CUE. Also, the overall explanatory power by CLM5 decreases in deep layers. Nevertheless, it is a very interesting issue to explore in the future research. We included this point in the revised manuscript in L128 - L131.

- Can carbon input be included in Ext. Data Fig. 7? I understand it is not estimated in the same way, because it is CLM5 derived. But, the environmental variables listed do also affect carbon inputs.

R3.4f We thank the referee for pointing this out. Carbon input was not predicted by the neural network. Extended Data Fig. 7 in the original manuscript (Extended Data Fig. 11 in the revised manuscript) was generated from the results of the neural network. We are afraid that we may not be able to include carbon input in this figure.

- Why is there no/little uncertainty in input carbon (Ext. Data Fig. 10)? I suppose this again has to do with the fact that input carbon is derived from CLM5. How much of a difference does it make if MODIS NPP or similar is used? Of course NPP is only a proxy for what actually enters the soil, and inputs are very difficult to measure accurately (e.g., roots, exudation), especially at scale.

R3.4g The referee is correct that the input carbon derives directly from the simulation by CLM5. Originally, we did not show the uncertainty of input carbon while uncertainties for all the other variables were generated from neural network prediction. In the revised figure, we added the uncertainty of input carbon that derived from CLM5 simulation (Response Letter Fig. 9). Similarly, we tested the influence of such uncertainty on the final simulation of SOC storage (Response Letter Fig. 8a). Specifically, we perturbed the spatial variation of NPP by randomly adding or subtracting its two standard deviation value to or from its mean value in CLM5 simulations. The results indicated that variations in NPP influence the accuracy in representing SOC variation. But CUE is still the most important mechanism in representing SOC by CLM5.

We have revised the main text of the manuscript (L757 – L760) to clarify how we evaluate the influence of NPP. We added the new results to the Fig. 4a of the main text. The uncertainty of NPP from CLM5 simulations was also presented in the Extended Data Fig. 12.

- It is not apparent why CUE is weighted in the way it is. I understand that the goal is to weight CUE by the flux going through each pathway. In contrast, all of the other “mechanisms” are weighted simply by the pool carbon densities. Why isn’t the decomposition rate weighted by the fluxes as with the CUE? In the present way, the weighting of CUE incorporates the decomposition rates, environmental modifiers, and soil thickness... so it is an amalgamation of various “mechanisms” presented herein. It would be helpful if the authors elaborated on their choice of weighting and the ranges of each value. Even the weighting for carbon input allocation, inspired by the ‘beta’ exponential functions of Jackson et al. 1996, could be explained more clearly. What about the proportion of input

carbon allocated as surface litter? (Furthermore, I assume that soil depth D_z and thickness Δz are in units of meters, but this should be specified as with the other variables. Though eventually B should come out as unit-less.)

R3.4f Thanks to referee for asking this good question. CUE is defined by how much carbon stays in the system versus how much carbon leaves the system via microbial metabolism. The system-scale CUE needs to consider all the carbon fluxes in the soil system. Because SOC decomposition, vertical transport and environmental modifiers are all dependent on carbon pool sizes, we weighted the baseline decomposition rate, vertical movement rate and environmental modifiers of individual carbon pools by their carbon pool sizes. We discussed the reasons of weighting for different mechanisms in L721 - L745 in the revised manuscript.

- How large are the uncertainties from the profile-level data assimilation? (L602: “uncertainties of parameters and SOC simulation did not propagate from the profile-level data assimilation but only reflected uncertainties generated by the deep learning model”)

R3.4g Thanks to referee for asking this good question. Uncertainty of simulated SOC at individual soil profiles is reflected by the coefficient of efficiency in Extended Data Fig. 2d, which indicates the explained variation of SOC. For the uncertainty of parameters after data assimilation, Response Letter Fig. 12 shows the standard deviation of the posterior distributions of different parameters after standardizing their values to the interval of [-0.5, 0.5] by their prior ranges. In site-level data assimilation, the uncertainty of parameters is important. For the global scale simulation with the PRODA approach, we may need to find a representative point estimate of the posterior distributions to be predicted in the neural network. As introduced in the manuscript, the mean value of the parameters' posterior distribution was selected to be predicted by the neural network. Thus, in the prediction of neural network, the uncertainty from site-level data assimilation was not included.

Response Letter Fig. 12 | Variations of posterior distributions of parameters in CLM5 after data assimilation at each soil profile. Bars present the median values of standard deviation of different parameters' posterior distributions. Error bars are the 68% confidence intervals. The grey dashed line indicates the median value of the standard deviation of different parameters. The right panel shows the prior distribution under the uniform distribution assumption (black line) and the posterior distribution (blue line) when the distribution follows a normal distribution with mean of 0 and standard deviation as the grey dashed line represents.

- How would the results change if CLM5 were assimilated to global SOC (e.g., SoilGrids) instead of the profile values? This would introduce some uncertainty from the scale-up of SOC and it's underlying covariates, but you are also introducing uncertainty with the parameter estimation from similar covariates.

R3.4h We thank the referee for the question and comments. SoilGrids was derived from the WoSIS database and so did the PRODA-optimised SOC maps. It may introduce more uncertainty if we would use SoilGrids as the 'observation' in our data assimilation and PRODA prediction.

- As the authors say themselves, the findings are based on (and interpreted with) CLM5, which does not explicitly represent microbial processes. However, they say that "CLM5 explicitly represents CUE via partitioning... to respiration versus accrual to SOC" (L188). This is by definition of CUE. This sentence tries to support their choice of model, but really

CUE is a necessary construct in any soil model that dictates the allocation of carbon staying in the system versus leaving the system.

R3.4i We thank the referee for pointing this out and we agree that our previous description was not accurate. In the revised manuscript, we included a microbially explicit model and reached similar conclusions. We revised the sentence in L108 – L112 of the revised manuscript as “*The Community Land Model version 5 (CLM5) expresses CUE via partitioning metabolic substrate to microbial respiration versus accrual to soil organic carbon even though the model implicitly represents many microbial processes. In comparison, a microbial model explicitly simulates the direct dependence of SOC decomposition on microbial biomass via enzymatic activities...*”. We hope the revision is clearer in describing CLM5 and the microbial model.

- What fraction of the WoSIS and NCSCD profiles were from natural/undisturbed ecosystems? Is the steady-state assumption (L501-505) reasonable?

R3.4g Thanks to the referee for asking this interesting question. We agree with the referee that SOC is rarely at absolute steady state in real world as climate changes and human activities disturb land carbon cycle¹⁶. However, the disequilibrium term is very small in comparison to the magnitude of SOC storage¹⁷. Therefore, the possible disequilibrium may not change much on the results shown in the manuscript. In the database we used in this study, 62% (71,041 out of 113,926) of the soil profiles are from natural ecosystems (i.e., with land cover type that is neither cropland nor urban area). We clarified the reasons why we used the steady state assumption in L538 - L544 of the revised manuscript.

- L165: ‘Deep learning’ here is jargon-y and doesn’t give much information. Specify neural network in the text.

R3.4k Thank the referee for this suggestion. We specified the “deep learning” as “multilayer neural network” in the revised manuscript (e.g., L89 and L163).

- L152: The authors state that boreal regions have high quality (low C:N ratio) SOC substrates. They cite Reich et al. 2004, which focuses on N:P ratios, and states that “N is the major limiting nutrient in younger temperate and high-latitude soils,” implying that boreal

regions are N-limited. Indeed, boreal regions can have high C:N ratio litter and soil stocks, which is seemingly the opposite of what the authors state.

R3.4l We agree with the referee. We deleted this point in the revised manuscript. The decomposability of soil organic matter could be influenced by many factors. High availability of soil organic matter in the boreal regions may be the main reason to its high baseline decomposition values. The sentence has been changed to “*In boreal regions, SOC is more vulnerable to future loss as indicated by high baseline decomposition rates (Fig. 3f), likely resulting from high accessibility and/or mineralizability (e.g., lack of interactions with soil minerals) of SOC substrates in this region.*” (L182 - L185)

I won't go into minor details and typos (e.g., L153 lability not liability, L486 are not is, L649 specifically not specially), given the above major comments that need re-thinking.

R3.4m We apologize for these typos. We have corrected them in the revised manuscript and carefully checked the grammars and writing again before the re-submission. We hope the revised manuscript sufficiently addresses the referee's comments.

References

- 1 Allison, S. D., Wallenstein, M. D. & Bradford, M. A. Soil-carbon response to warming dependent on microbial physiology. *Nature Geoscience* **3**, 336-340 (2010).
- 13 Ye, J. S., Bradford, M. A., Dacal, M., Maestre, F. T. & García-Palacios, P. Increasing microbial carbon use efficiency with warming predicts soil heterotrophic respiration globally. *Global Change Biology* **25**, 3354-3364 (2019).
- 14 Sinsabaugh, R. L. *et al.* Stoichiometry of microbial carbon use efficiency in soils. *Ecological Monographs* **86**, 172-189 (2016).
- 16 Luo, Y. & Weng, E. Dynamic disequilibrium of the terrestrial carbon cycle under global change. *Trends in Ecology & Evolution* **26**, 96-104 (2011).
- 17 Lu, X., Wang, Y.-P., Luo, Y. & Jiang, L. Ecosystem carbon transit versus turnover times in response to climate warming and rising atmospheric CO₂ concentration. *Biogeosciences* **15**, 6559-6572 (2018).

Reviewer Reports on the First Revision:

Referee #2 (Remarks to the Author):

Authors have made a great job in revising the manuscript. I am really satisfied with the way they addressed my comments on the previous version of this manuscript. More information and analyses on the cross-validation, mapping, permutation importance and uncertainty quantification parts have been adequately included in this revision.

Referee #3 (Remarks to the Author):

The study by Tao et al. leverages a synthesis of CUE measurements, global soil profiles, and two biogeochemical models, to demonstrate through their PRODA (data assimilation & deep learning) approach the global importance of carbon use efficiency for soil carbon storage. I appreciate the considerable amount of work that the authors carried out for their revisions, and do believe that this is an interesting and impressive study that stimulates discussion. However, I still have several concerns regarding the conceptual framework and motivation, as well as the methods used and the interpretability of the results.

1. Framing of conceptual arguments and motivation.

I understand the desire to frame this study around the two conceptual hypotheses in Fig. 1. However, the processes behind these two hypotheses are not mutually exclusive. In reality, both occur and the amount allocated to enzymes that catalyze decomposition, in Fig. 1b, can vary depending on many factors, including nutrient availability and seasonal differences in moisture, temperature, and inputs that alter the microbial community. The positive feedback of microbial biomass and enzyme production (as in Fig. 1b) could, in addition to catalyzing decomposition, promote mineral-organic associations through an increase in sorption of microbial necromass and byproducts (Fig. 1a). These latter processes (mineral interactions, Fig. 1a) are not included in the microbial model, and yet, a positive CUE-SOC relationship still emerges (for Fig. 1b). What does this tell us about the two proposed arguments?

The positive CUE-SOC relationship is supported by both microbial and first-order models following data assimilation, so the discussion of the negative relationship seems to be largely conceptual and not supported by the data. Could the authors clarify if this is the case? Besides the Allison et al. 2010 modeling paper (which did not specifically dive into this topic, but rather the role of CUE on responses of soil to warming), can the authors provide additional citations (especially empirical) for the “CUE-SOC relationship controversy” as further motivation for their framing?

Indeed, the paper still comes across as selectively choosing and testing parts that “agree” with the findings presented, as also mentioned by another reviewer. The corresponding response (R1.3b) seemed to have missed this point, and more discussion of the literature that has suggested a negative SOC-CUE relationship is warranted. In other words, more introduction should be given to better motivate the gaps in the literature and the hypotheses and set-up of this paper.

Following on this, there was an important point made regarding how the 'mechanisms' feed into the model, and the corresponding response (R1.3c) again seemed to miss the point and just reiterated what the authors had already done. I would challenge the statement "from a perspective of modelling, there is no such a priori expectation that CUE should be a stronger predictor than others", and encourage the authors to think about this more deeply.

2. Definition and calculation of system-scale CUE.

I appreciate the second R3.4f response (note that there were two R3.4f and R3.4g responses), and understand the rationale behind the calculations here, but it seems that the present way of calculating CUE – weighting it by the baseline decomposition and environmental modifiers – makes it an amalgamation of various "mechanisms" presented herein. I feel that this is worth discussing in the context of it being more important than the other individual 'mechanisms' that are used in its calculation. Is it really fair to compare these 'mechanisms' on the same playing field, when one 'mechanism' is calculated using several of the 'mechanisms' it is being compared against?

The authors provide interesting results on SOC-CUE slopes across different soils and biomes for CLM5 in Response Letter Fig. 11, but these results could be discussed further in the main text (L135). The findings were mainly stated in the response letter, which is not helpful for readers. Even there, the authors simply state that, for example, "SOC storage is more responsive to CUE in tropical than in boreal regions" – but why? This is interesting in combination with the typically higher CUE in boreal than in the tropics (Fig. 3c). Though I'd also caution the authors in saying more 'responsive', since this is derived from a space-for-time substitution. The results in Response Letter Fig. 11 (Ext. Data. Fig. 7) were for CLM5, but what about the microbial model?

Regarding differences in CUE-SOC slopes across the models and measurements (e.g., in Fig. 2), the added text and R3.1c feel too brief and superficial. The authors give a sense of the covariates involved for CLM5 results, but not for the meta-analysis data or microbial model. As noted above, could similar results from the other data sources also be provided? It is also interesting that the slope from the microbial model is much better aligned with the meta-analysis data. Why? And how does this fit together with the authors' conclusions that the microbial model is well-approximated by the first-order model?

More discussion could also be provided on the microbial model CUE and parameter sensitivity analysis in the main text. How relevant is it? What does this mean physiologically for microbes and their enzyme production? How can this vary temporally? The response letter contains some discussion where Response Letter Fig. 2 is presented, but I could not find this in the main text (apologies if I missed it). How much did the value of beta vary in the assimilated global model? I assume that beta is 'allo_slope' in Extended Data Table 8; if so, please check for consistency in parameter names.

3. Robustness and predictability.

Given the sparse global distribution (Fig. S1), could the authors comment further on the robustness

of their results in data-poor regions and the predictability across studies? Would adding another study elsewhere change the results? What about (the lack of) data in the tropics and in deeper soils (e.g., below 30cm), where model results are then presented? Indeed, there was no/little data for these regions, but the results and patterns in declining explanatory power were used to imply process – for example, in Response Letter Fig. 4: “Declining explanatory power of CUE to the variation in SOC with soil depths indicates a shift from biotic-dominated to abiotic-dominated SOC accumulation.” I understand that data paucity is often a problem for such questions, but this is important when making global conclusions.

With regards to the robustness of regression slopes and intercepts, the question (to me at least) is less about subsampling the 1,000 soil profiles for a mixed effects model, and more about the (stratified) sampling from the 52,000 WoSIS profiles to the 1000 profiles in the data assimilation. If different 500-1000 soil profiles were subsampled (covering the same climate-weighted domains) from the WoSIS profiles, would the data assimilation results of CUE differ? This would be a better test of robustness, as opposed to subsampling the data assimilation output. More information on this subsampling would also be helpful, following on L628-631. How many points were selected from each climate zone and how much heterogeneity is there between points in a given climate zone? On this note, it would also be helpful to show that the (1) synthesis data, (2) 1000 subsampled points, and (3) 52,000 WoSIS profiles cover similar regions of a multi-dimensional covariate space.

What if other covariates were included in the mixed-effect models of SOC-CUE relationships? For example, soil moisture, clay content, NPP, etc. At least for the data assimilation results, where these covariates are reported.

What if the SCE method was used to assimilate SOC data into the CLM5 model as well? As a check on using a different assimilation method for the microbial model.

4. Proportional changes in ‘mechanisms’ are still difficult to interpret.

I think the authors may have misunderstood the R3.3a and R3.3b comments. I appreciate their responses and the fact that they do not know the parameter space at each grid cell. However, I feel that having some sense of what the relative change in a given parameter means is critical. I completely understand what the environmental modifier is from a modeling perspective, but again, it is a model construct and a 7% decrease is not very meaningful without context. I feel the results would be much more impactful if the authors could give this context. For example, in the case of the environmental modifier, how much does it vary spatially in the best fit model? What does this mean with regards to differences in temperature and moisture?

Again, I am not sure ‘mechanism’ is the correct word for environmental modifier, baseline decomposition, or most of the ‘mechanisms’ herein, even carbon use efficiency. Some of these are model constructs and others are, as the authors acknowledge in their response, emergent properties resulting from various underlying processes. I appreciate the added text on L714, but I still think ‘mechanisms’ might not be the correct terminology.

Other comments:

The authors state (L222) that the permutation analysis indicates that soil physical properties explain CUE's spatial variability more than climate or soil chemical properties, but the latter two appear quite close. Disregarding them seems misleading.

The Fig. 2 caption should mention how many profiles were in the meta-analysis. Currently only the WoSIS and profile subsets are listed.

The description of the empirical constraints as one pool being X-fold the size of another could be changed to percentages, which are more common (as in the studies cited). For example, if I understood correctly following on L615, SOC had to exceed MIC by at least 50-fold over the entire profile means that MIC can be at most 2% of SOC over the profile. Similarly, DOC could be at most 10% of SOC in the entire profile and 1% of SOC in the topsoil. The constraint on DOC in the entire profile seems quite high... maybe this was a typo by the authors? The cited Guo et al. paper estimates DOC at around 1% of SOC across depths. In any case, writing these constraints as percentages of SOC would be more intuitive to most readers.

L595 – should this read enzyme turnover rate? Also worth editing the Extended Data Fig. 4 caption to state enzyme production and turnover, as done in Response Letter Fig. 2. Tau is not well-defined or discussed in the manuscript and only really appears in Extended Data Fig. 4, unless I missed something. I would suggest putting a subscript 'enz,decay' on tau to prevent confusion with microbial turnover rates. I also wonder why only beta was selected for the sensitivity analysis and not the enzyme production rate k ?

In Extended Data Fig. 4 (i.e., Response Letter Fig. 2 – please use only one labeling scheme in future iterations if the figure is replicated exactly) shouldn't the case where beta = 1 (with tau = 1) depicted in dark red be the same as the case for tau = 1 (with beta = 1) in light blue? Why are these different? Apologies if I missed something, but either way, clarification in the caption would be helpful. Also, why are values for beta = 0.3 to 0.5 excluded with CUE = 0.2?

Why is it not possible to analytically solve for soil carbon pool sizes in the microbial models (L598), especially in the simple one used here? It may be more complicated due to nonlinearities, but can be explicitly solved, as done in past microbial modeling papers.

Using the word 'preservation' for soil carbon may imply longevity and could be misleading. I suggest using the word storage or stocks instead. Also, the words accumulation and loss suggest a dynamic change, whereas here you are using a space-for-time substitution, which often does not inform change under new conditions. Another reviewer brought up this issue of space-for-time, and I think the text could use more caveats on this point. The revised sentences (as in response R1.3a) do not mention potential problems with space-for-time (e.g., Abramoff et al. 2019) and currently state that "projecting changes in SOC needs to use full distributions of potential values of these mechanisms in future research" (L217). However, it's not only about the full distribution of potential values, but also how current patterns in these values can or cannot inform projections.

Editorial note:

Referee #1 was unable to review the revised version of the paper. Therefore, another referee (referee #4) was asked to assess the authors responses to referee # 1.

Referee #4 (Remarks to the Author):

I have been asked by the journal editor to review the authors' response to referee #1 comments, since she/he was not available to handle this revision. Hence, I have focused my revision only on such author response.

1. Assigning process to pattern:

I consider this the most important concern from referee#1, and I concur with her/his assessment that using only a first-order soil C model (CLM5) to address the outcome of the CUE vs SOC relationship was biased towards the positive trend, because of the absence of a feedback from microbial biomass towards SOC. The authors followed the suggested indications and used a four-pool microbial-explicit model as in Allison et al (2010) to check whether a positive CUE vs SOC relationship was also found with a microbial model. Importantly, authors run this model using the data assimilation framework for increased generality and assessed parameter sensitivity. The results of the microbial model also support the positive CUE vs SOC relationship, and I appreciate the comprehensive response letter on this regard. I do think the ms has greatly improved with this addition.

However, I do have a small concern about the previous analyses. If the outcome (+ or -) of the CUE-SOC relationship depends on the parameter values of enzyme production used, I wonder how this has affected the interpretation made by authors (i.e. non-linear scaling of enzyme production and microbial biomass hinders higher SOM decomposition at high CUE). I have not found any discussion about this, but I guess this is needed to guide the reader and avoid confusion, either in the caption of Ext Data Fig. 4 and/or in the main text.

2. Correlation vs. causation:

I think the authors have done a good job incorporating the study ID as a random factor in the meta-analysis as suggested by referee#1. Actually, different structures were tried and all results point to a similar CUE-SOC relationship.

As it was expected with such a hierarchical dataset in the meta-analysis (132 observations from only 16 experimental sites/papers), the variance explained in SOC by the mixed model is way larger (twice or even more in some model structures) than the fixed model. Although the number of sites (46) is actually not that bad, the meta-analytical database is quite poor in spatial spread and in number of studies. This clearly reflects the scarcity of field studies measuring CUE, but it is also a weakness of the meta-analysis and should be reflected in the text (or at least put into context with the need to be complemented with the other analytical approaches exploring the CUE-SOC relationship.

I found intriguing why results in Extended Data Table 2 show a positive MAT effect on SOC but a negative MAT effect on microbial biomass in Extended Data Table 2. What is driving this?

It seems many of your studies from the meta-analysis also include some management treatments that may confound these effects (eg. Fertilization, drought). Are you including these plots too in the meta-analysis or only the control ones? I did not find any info about this in the methods.

Overall, I think the concern of the previous reviewer regarding correlation vs causation has been satisfactorily dealt with in this revision.

3a. Space for time substitution:

This has been nicely incorporated into the main text.

3b. Discussions of literature findings that disagree with your results:

I don't think that removing the text on other plausible outcomes of the CUE vs MAT relationship (actually the negative one) is the solution to avoid confusion. Actually, I concur with referee #1 assessment that your findings must be discussed and confronted with previous results from the literature, specifically to those that also use model-data integration to represent the implications of microbial physiology across large spatial scales. Actually, the negative CUE vs MAT relationship found in Ye et al 2019 GCB was somehow robust to variation in C substrate availability, although it's true that other resources such as N and P were not assessed. With all this being said, I think this context should be brought back to the main text, and this should be useful to readers to better frame your results within the wider literature, even if this means bringing up apparently contradictory results. In my opinion, the different results here are profoundly influenced by the lack of CUE field assessments across diverse ecosystems, which is affecting our scientific understanding, but still we have to move forward, and your paper clearly helps to do so in a very compelling manner.

3c.

This has been nicely incorporated into the main text.

3d. PRODA approach:

The new paragraph makes a good introduction for the non-specialized readers about the goals of this approach, and benefits, from previous models. This helps to identify the novelty and to differentiate this study from previous ones.

Minor issue:

Revise the reference numeration in Extended Data Table 1, as this is not matching the reference list.

Author Rebuttals to First Revision:

Summary of responses to comments by referees

All the three referees are highly positive about our first-round revision. Referee #2 was satisfied with our responses to her/his comments on the previous version of this manuscript. Referee #3 was highly positive about our substantial revisions and suggested that our study is interesting, impressive, and going to stimulate discussion. Referee #4 was also positive about our comprehensive responses to comments raised by Referee #1, did think the manuscript was greatly improved with the last revisions, and suggested that our paper clearly helps move forward with future CUE field assessments across diverse ecosystems in a very compelling manner.

Referees #3 and #4 offered more comments on the revised manuscript. All these comments are constructive for us to improve our manuscript further. We have addressed all their concerns in the response letter and revised the manuscript accordingly. Here is a summary of the revision of our manuscript.

1. Motivation of this study: Following suggestions by referee #3, we further frame the motivations of this study around the two key questions (R3.1c): how important is CUE relative to other mechanisms in determining SOC storage over the globe? Would CUE be positively or negatively correlated to SOC, especially if CUE is important in determining SOC storage? In particular, we further detailed the controversy of the CUE-SOC relationship in both the empirical and modelling communities (R3.1b). Moreover, we clarified the role of data assimilation in disentangling the CUE-SOC relationship controversy (R3.1a). By fusing observations with models (i.e., prior knowledge where both positive and negative CUE-SOC relationships are possible), data assimilation updates our understanding to generate a posterior knowledge that a positive CUE-SOC relationship is more probable than a negative one. Consequently, the argument about the positive CUE-SOC relationship (Fig. 1a) is favoured in controlling the soil carbon cycle, although these two relationships (as shown in Fig. 1a and 1b) both have some likelihood to co-occur in reality.

2. System-level CUE and CUE-SOC regression slopes: Referee #3 raised concerns about the way we calculated system-level CUE. In the response letter (R3.2a) and revised manuscript, we further explained that the weighting scheme used in calculating system-level

CUE is necessary to combine the various carbon transfer efficiencies (i.e., a_{ij} in Equation 3) into a single metric that can be compared with measured CUE at the microbial community level. Moreover, we discussed the reasons for different CUE-SOC slopes from different sources and methods (R3.2c). Different isotope-labelling methods in field experiments and optimization algorithms in data assimilation (see also R3.3d) could lead to varying slopes. Although both the models used in this study follow a definition of CUE that is consistent with the one measured by isotope labeling methods, neither of the models explicitly simulates isotope (i.e., $^{13}\text{C}/^{14}\text{C}$ or ^{18}O) dynamics in microbial metabolisms. Therefore, it is not unexpected that the CUE-SOC slopes differ between the data assimilation and meta-analysis despite the fact that the microbial model may yield a CUE-SOC slope close to that of the meta-analysis.

3. Robustness of conclusions: Thanks to the suggestions by referees #3 and #4, we discussed the data coverage in the revised manuscript (R3.3a and R4.2b). We continued data assimilation with the microbial model in the past two months after we submitted the revised manuscript last time. Therefore, this round of revision added another 1,500 data assimilation results using the microbial model to the analyses (R3.3b). The newly added data assimilation results did not change any main conclusions discussed in this manuscript, confirming the robustness of our sampling method and conclusions drawn from the results. Following suggestions by referee #3, we also tested the CUE-SOC relationship using all the data assimilation results with CLM5 considering fixed effects from other covariates (e.g., clay content, NPP, etc.) in the mixed-effects modelling (R3.3c). Mixed-effects models with different structures all supported the positive relationship between CUE and SOC. Moreover, we applied the same SCE algorithm used in the microbial model data assimilation to CLM5 at the 2,500 representative sites (R3.3d). The results showed a high agreement on the retrievals of different mechanisms between the SCE and MCMC methods. Therefore, results from both data assimilation methods (i.e., MCMC and SCE) support the positive CUE-SOC relationship.

4. Relationship between temperature and CUE: Both referee #3 and referee #4 suggested more discussion about the relationships between mean annual temperature (MAT) and microbial CUE. In the response letter (R3.1c and R4.3b for details), we discussed that different definitions of CUE (i.e., physiologically defined CUE that can be measured by

isotope data versus stoichiometrically defined CUE that is measured by exoenzyme data) may cause differences in estimated MAT-CUE relationships. The physiologically defined CUE as measured by many previous studies, estimated by our meta-analysis, and simulated by the two models all decreases in warmer regions. The exoenzyme-derived CUE, which is confounded by microbial shifts in resource use in response to substrate stoichiometry in addition to the ratio of microbial growth over metabolized carbon, however, increases with temperature.

We hope that our responses and revision of the manuscript are satisfactory to the referees. And we look forward to further feedbacks and comments.

Below are our point-by-point responses (in blue) to referees' comments (in black).

Point-by-point responses to referees' comments:

Referee #2 (Remarks to the Author):

Authors have made a great job in revising the manuscript. I am really satisfied with the way they addressed my comments on the previous version of this manuscript. More information and analyses on the cross-validation, mapping, permutation importance and uncertainty quantification parts have been adequately included in this revision.

Response: We greatly appreciate the positive evaluation by this referee on our first-round revision.

Referee #3 (Remarks to the Author):

The study by Tao et al. leverages a synthesis of CUE measurements, global soil profiles, and two biogeochemical models, to demonstrate through their PRODA (data assimilation & deep learning) approach the global importance of carbon use efficiency for soil carbon storage. I appreciate the considerable amount of work that the authors carried out for their revisions, and do believe that this is an interesting and impressive study that stimulates discussion. However, I still have several concerns regarding the conceptual framework and motivation, as well as the methods used and the interpretability of the results.

Response: We greatly appreciate the generally positive evaluation on our manuscript by referee #3. We have addressed all the concerns as described below.

1. Framing of conceptual arguments and motivation.

I understand the desire to frame this study around the two conceptual hypotheses in Fig. 1. However, the processes behind these two hypotheses are not mutually exclusive. In reality, both occur and the amount allocated to enzymes that catalyze decomposition, in Fig. 1b, can vary depending on many factors, including nutrient availability and seasonal differences in moisture, temperature, and inputs that alter the microbial community. The positive feedback of microbial biomass and enzyme production (as in Fig. 1b) could, in addition to catalyzing decomposition, promote mineral-organic associations through an increase in sorption of microbial necromass and byproducts (Fig. 1a). These latter processes (mineral interactions, Fig. 1a) are not included in the microbial model, and yet, a positive CUE-SOC relationship still emerges (for Fig. 1b). What does this tell us about the two proposed arguments?

R3.1a: We thank referee #3 for asking this great question. A positive CUE-SOC relationship emerges from the two structurally different biogeochemical models after the two models are informed by data. It is because data contains information that can help constrain model parameters and predictions with a data assimilation framework (e.g., via MCMC and SCE methods in this study). The *prior* knowledge from the microbial model suggests that either a positive or a negative relationship between CUE and SOC could occur under different parameter combinations (Extended Data Fig. 3 of the revised manuscript). The new information from data of the 1,000 (we increased the number of profiles to 2,500 in this revision) vertical SOC profiles worldwide updates our understanding to generate a *posterior* knowledge that a positive CUE-SOC relationship is more probable than a negative one,

although both arguments (as shown in Fig. 1) underlying these two relationships have some likelihood to co-occur in reality.

The referee is correct that the microbial model includes processes of carbon allocation to enzymes that catalyze decomposition but does not explicitly include many related factors, such as nutrient availability, seasonal differences in moisture and temperature, and carbon inputs that alter the microbial community. The microbial model used in this study includes feedback processes between microbial biomass and enzyme production but does not include explicitly the processes related to mineral-organic associations through an increase in sorption of microbial necromass and byproducts. *“A model, no matter how complex it is, could not represent all the processes of one system at resolved scales. Interactions of processes at unresolved scales with those at resolved scales should be reflected in model parameters”* stated in Luo and Schuur (2020)¹. Although the microbial model does not include many processes, such as mineral-microbe interactions (i.e., the processes at unresolved scales), these processes at unresolved scales are represented in the estimated parameters of these processes at resolved scales after data assimilation. That is the reason why a positive CUE-SOC relationship emerges after data assimilation, even if some of the processes that may lead to a positive CUE-SOC relationship are not explicitly included in the microbial model.

The two structurally different biogeochemical models, after being constrained by data, both generate positive CUE-SOC relationships. It is because globally minimizing the mismatches between observed and modelled values trains models of different structures to predict similar dynamics of one system. This has been documented in the literature. For example, Wang et al. (2022)² used observations from one field warming experiment to train two models (a carbon-only model and a carbon-nitrogen coupled model) with data assimilation techniques. While estimated parameters differ between the two models, the predicted carbon pool sizes and their changes under warming are similar between the two models. MacBean et al. (2022)³ also showed that two different versions of a terrestrial biosphere model (i.e., ORCHIDEE) yielded similar predictions after being constrained by data assimilation. After data assimilation in this study, the Michaelis constant for SOC decomposition (i.e., $k_{m,decom}$) in the Michaelis-Menten equation of the microbial model is about 100 times that of its substrate concentration (i.e., SOC concentration) for most of the soil profiles (Extended Data Fig. 8 of the revised manuscript). Thus, the nonlinear kinetics for enzyme-based SOC decomposition in the microbial model can be approximated by the first-

order kinetics as in CLM5 with respect to SOC, leading to converging predictions by the two structurally different models.

We hope the above explanation helps understand the reasons why the two structurally different biogeochemical models yield similar positive CUE-SOC relationships after data assimilation. In the revised manuscript, we highlighted the possibility of co-existence of the two arguments about the CUE-SOC relationship (e.g., L92, L126 - L129, L230 - L232 and the legend of Fig. 1) and the power of data in identifying the most probable relationships by data-model fusion (i.e., data assimilation) (L112 – L116 and L630 – L640).

The positive CUE-SOC relationship is supported by both microbial and first-order models following data assimilation, so the discussion of the negative relationship seems to be largely conceptual and not supported by the data. Could the authors clarify if this is the case? Besides the Allison et al. 2010 modeling paper (which did not specifically dive into this topic, but rather the role of CUE on responses of soil to warming), can the authors provide additional citations (especially empirical) for the “CUE-SOC relationship controversy” as further motivation for their framing?

R3.1b: The referee is correct that our results in this manuscript suggest that both the microbial and first-order models after data assimilation support a positive CUE-SOC relationship. In the manuscript L230 - L232, we stated that *“Our results from the two structurally different models and meta-analysis of field experiments support the argument that a high microbial CUE favors storage of SOC more than respiratory loss”*

Although this study shows that the first argument (Fig.1a) has a high probability to play the dominant role in controlling SOC storage and generate the positive CUE-SOC relationship after the microbial model is constrained by the global SOC dataset, it has been debated whether the CUE-SOC relationship is positive or negative in both the empirical and modelling research communities. While data from field surveys^{4,5} and microcosm studies⁶ supported a positive CUE-SOC relationship, it has been discussed in theories via modelling studies⁷⁻⁹ and observed in empirical studies¹⁰ that a high CUE leads to increasing microbial biomass and stimulates extracellular enzyme activity (EEA). Separate data syntheses studies have suggested that increased EEA is related to enhanced soil respiration and SOC loss¹¹. For example, increased activity of glucosidase leads to microbial biomass loss^{12,13} and activity level of oxidase is negatively correlated with mineral SOC storage^{14,15}. Similarly, a high CUE associated with glucose addition is correlated with SOC priming along a gradient of soils

with varying C:N ratio and SOC content¹⁶. These separate empirical studies implied a possibility that CUE and SOC could be negatively correlated.

Similarly, many modelling studies have shown a negative relationship between CUE and SOC. Schimel and Weintraub (2003)¹⁷ proposed one of the first microbial models to highlight the role of exoenzyme activities in soil carbon cycle. The study by Allison et al.¹⁸ further anchored the importance of microbial physiology in regulating soil carbon cycle and triggered discussions about the role of CUE in determining SOC storage. In the past decades, dozens of microbial models with different structures have been built upon the basic structure proposed by Allison et al. (2010) (e.g., refs^{19,20}). Allison et al. (2010) stated that “As enzyme production is linked to biomass, the decline in CUE ultimately limited the enzyme catalyst for SOC decomposition”, suggesting a lower CUE may actually lowering the respiratory losses of SOC in the long term. This conclusion has also been drawn from other modelling studies with similar model structures^{19,21}. However, mathematical analysis on those microbial models further revealed that the role of CUE in regulating SOC dynamics relies on the value of key parameters²¹, highlighting the importance of fusing data with process models to gain emerging understanding of SOC storage.

Overall, many empirical and modelling studies have suggested either positive or negative CUE-SOC relationships. These debates further motivate our study to examine the relationship between CUE and SOC storage at the global scale.

In the revised manuscript, we further clarified our motivation of solving the CUE-SOC relationship controversy in L126 – L129 and included the above cited references in the main text (e.g., refs 30-31 35-37 of the main text).

Indeed, the paper still comes across as selectively choosing and testing parts that “agree” with the findings presented, as also mentioned by another reviewer. The corresponding response (R1.3b) seemed to have missed this point, and more discussion of the literature that has suggested a negative SOC-CUE relationship is warranted. In other words, more introduction should be given to better motivate the gaps in the literature and the hypotheses and set-up of this paper.

R3.1c: We thank this referee for the suggestion. We have framed our study based on two key questions: how important is CUE relative to other mechanisms in determining SOC storage over the globe? Is CUE positively or negatively correlated with SOC, especially if CUE is important in determining SOC storage? In the manuscript, we framed our study to address the

first question by reviewing a currently prevailing paradigm that emphasizes the importance of carbon input and decomposition in SOC accrual (L61 – L70), current knowledge gaps (L70 – L74), and the increasing evidence on the role of microbial physiology (as represented by CUE) in the soil carbon cycle (L75 – L77). Thanks to the suggestions by this referee, we have added additional citations to better frame this study for addressing the second question with a balance view on both the negative and positive relationships between CUE and SOC in the revision as we discussed in R3.1b.

While the relationship between temperature and microbial CUE (as discussed in R1.3b of the first-round response letter) is important to be explored, it is not a main question we focused on in this study. Thanks to the suggestions by referees #3 and #4, we included the MAT-CUE relationship in this revision, and we explained why it is negative in our study yet positive in Ye et al.'s study²². The detailed explanations can be found in R4.3b. Briefly, the CUE data used in our meta-analysis was defined as the ratio of carbon used in microbial growth relative to the total carbon used in metabolism. It was measured by isotopically (¹³C/¹⁴C or ¹⁸O) labelled substrates in empirical studies. The CUE defined in our models is conceptually consistent with the isotope-derived CUE. Ye et al. (2019) used CUE estimates from enzyme and substrate stoichiometry (i.e., C:N or C:P) relationships. In addition to the ratio of microbial growth over metabolized carbon as in the isotope-derived CUE, the stoichiometry-derived CUE is confounded by microbial shifts in resource use in response to substrate stoichiometry²³.

In the revised manuscript (L159 - L163) and Supplementary Discussion, we discussed the MAT-CUE relationship and the reasons why such relationship is different when CUE is defined by different methods.

Following on this, there was an important point made regarding how the 'mechanisms' feed into the model, and the corresponding response (R1.3c) again seemed to miss the point and just reiterated what the authors had already done. I would challenge the statement "from a perspective of modelling, there is no such a priori expectation that CUE should be a stronger predictor than others", and encourage the authors to think about this more deeply.

R3.1d: We thank the referee for the comment. We stated "*there is no such a priori expectation that CUE should be a stronger predictor than others*" for two reasons. First, the sensitivity analysis with the microbial model in our study indicated that CUE could have negative, null, and positive impacts on SOC storage (Extended Data Fig. 3 of the revised

manuscript). The model does not have an *a priori* expectation on either the sign (i.e., positive, null, or negative) of the CUE-SOC relationship or the importance (i.e., effects with a sign) of CUE to SOC storage. Second, researchers of the soil carbon cycle have not historically placed (at least not on purpose) higher weights to CUE than any other processes (e.g., carbon input or decomposition) in influencing SOC storage. As we discussed in the manuscript (L61 – L70), the conventional paradigm has emphasized the importance of plant carbon input and substrate decomposition in regulating SOC storage instead of microbial CUE. The importance of microbial CUE emerged from this study only after we fused global soil data into the models. Nevertheless, we acknowledged a possibility that the way of incorporating CUE into process models may make CUE more important than other mechanisms in influencing SOC in L209 – L211 of the revised manuscript.

2. Definition and calculation of system-scale CUE.

I appreciate the second R3.4f response (note that there were two R3.4f and R3.4g responses), and understand the rationale behind the calculations here, but it seems that the present way of calculating CUE – weighting it by the baseline decomposition and environmental modifiers – makes it an amalgamation of various “mechanisms” presented herein. I feel that this is worth discussing in the context of it being more important than the other individual ‘mechanisms’ that are used in its calculation. Is it really fair to compare these ‘mechanisms’ on the same playing field, when one ‘mechanism’ is calculated using several of the ‘mechanisms’ it is being compared against?

R3.2a: We thank this referee for raising this good comment, and we are sorry for the typo in numbering our previous response letter. As described in the manuscript, the system-level CUE is the sum of CUE along each of the carbon transformation pathways (i.e., a_{ij} in Equation 3) weighted by the carbon fluxes over all the pathways in the soil system. It is not an amalgamation of various mechanisms although the calculation of carbon fluxes as the weighting factor requires information of decomposition, environmental scalar and depth. Without the weighing scheme we introduced, it would not be possible to calculate system-level CUE values, which can be compared to observations. Specifically, a_{ij} (i.e., CUE along the carbon transfer pathway to recipient pool i from donor pool j) in equation 3 of the manuscript is weighted by the flux size from donor pool j (i.e., $\sum_z x_{j,z} k_j \xi_z \Delta z$), which measures the amount of carbon incorporated into pool i from pool j , over the total carbon flux

in the soil system (i.e., $\sum_j \sum_z x_{j,z} k_j \xi_z \Delta z$). Because carbon fluxes are generated from SOC decomposition, the carbon pool size of donor carbon pool $x_{j,z}$, its baseline decomposition rate (k_j) and environmental modifier (ξ_z) at each soil layer (z) were used in this calculation (as correctly noted by the referee).

We revised the manuscript L745 – L764 and L797 – L806 to describe how we calculated the system-level CUE with more details and how we conducted the second sensitivity analysis using the system-level CUE. We summarized our interpretation of bulk CUE in L760 – L764 as follows: *“Equation 3 combines all the microbial carbon use efficiencies, a_{ij} , along individual pathways from pool j to pool i into one single metric, CUE_{bulk} , that can be compared with measured CUE at the microbial community level that combines the carbon conversion efficiencies of multiple substrates into multiple microbial taxa”*

The authors provide interesting results on SOC-CUE slopes across different soils and biomes for CLM5 in Response Letter Fig. 11, but these results could be discussed further in the main text (L135). The findings were mainly stated in the response letter, which is not helpful for readers. Even there, the authors simply state that, for example, “SOC storage is more responsive to CUE in tropical than in boreal regions” – but why? This is interesting in combination with the typically higher CUE in boreal than in the tropics (Fig. 3c). Though I’d also caution the authors in saying more ‘responsive’, since this is derived from a space-for-time substitution. The results in Response Letter Fig. 11 (Ext. Data. Fig. 7) were for CLM5, but what about the microbial model?

R3.2b: We thank the referee for the comment. This is an extremely interesting issue that warrants full exploration. However, this manuscript has already covered many topics. Our manuscript is already longer than allowed to fully explain differences in SOC-CUE slopes and their causes except in Supplementary Information. Meanwhile, as we explained in the previous response letter, we did not include too much discussion about the slopes of CUE-SOC relationship across ecoregions because of the lack of sufficient experimental data (i.e., field-measured CUE). While we found consistent results from both experimental studies⁴ and our analysis (Supplementary Fig. 2b) that the CUE-SOC slope is higher in acid soils than neutral or alkaline soils, presently we cannot find a rigorous benchmark from the field measurements to evaluate the results from data assimilation. In the future, detailed

understanding of the differences and their drivers in the slopes of CUE-SOC relationship across ecoregions requires more empirical data and experiments. We hope to fully explore this issue in the future, especially if we can find a way to collaborate with the referee.

We added the results from the microbial model in Response Letter Fig. 1 in this revision. In most cases, the slopes of CUE-SOC relationships vary similarly across ecoregions between these two models. Yet results across climatic zones differed between the two models. Fully understanding the variation in slopes across ecoregions and between different models needs more experiments and data in the future.

In the revised manuscript, the Response Letter Fig. 1 and relevant discussion were presented as Supplementary Fig. 3 and in the Supplementary Discussion, respectively.

Response Letter Fig. 1 | Regression slopes of the CUE-SOC relationship across different ecoregions and soil types with the microbial model. Results are the slopes of the linear regression of the CUE-SOC relationship in different ecoregions and soil types. Error bars are the standard errors from the linear regressions. Numbers above the bars are the dimensions of freedom (*df*) in linear regressions. Line types of the bar and error bar plot indicate the *P* values of the regression slopes.

Regarding differences in CUE-SOC slopes across the models and measurements (e.g., in Fig. 2), the added text and R3.1c feel too brief and superficial. The authors give a sense of the covariates involved for CLM5 results, but not for the meta-analysis data or microbial model. As noted above, could similar results from the other data sources also be provided? It is also interesting that the slope from the microbial model is much better aligned with the meta-analysis data. Why? And how does this fit together with the authors' conclusions that the microbial model is well-approximated by the first-order model?

R3.2c: We thank the referee for the comment. Our study confirms the positive CUE-SOC relationship by all the methods and from different sources of data despite differences in the slopes among different methods. In Response Letter Fig. 1, we further showed that the positive CUE-SOC slopes retrieved from the microbial model vary similarly across ecoregions as those from CLM5.

The slopes of the positive CUE-SOC relationship vary in our meta-analysis and between the two models. Different methods used to measure or estimate CUE strongly influence the slopes of CUE-SOC relationship. For the results from the two models, we discussed in R3.3d below that the different CUE-SOC slopes mainly resulted from different methods applied to data assimilation (i.e., SCE versus MCMC). The SCE method enlarged the variance of retrieved CUE than that from the MCMC method and thus reduced the regression slope in the CUE-SOC relationship. We also observed different CUE-SOC slopes by different methods in field experiments. Our meta-analysis data included two kinds of isotopes (i.e., $^{13}\text{C}/^{14}\text{C}$ and ^{18}O) used to label substrate in measuring CUE. While the positive sign of the CUE-SOC relationship still held, results from the mixed-effects modelling suggested different CUE-SOC slopes with different isotope-derived CUE data (Response Letter Table 1). Labelling different compounds (i.e., organic carbon substrate for $^{13}\text{C}/^{14}\text{C}$ and water for ^{18}O) may eventually affect the absolute value of CUE-SOC slopes in empirical studies (see similar discussion in ref²⁴).

While the CUE-SOC slopes retrieved from the two models are similar to those measured by isotope labeling experiments in terms of the conceptual definition all based on carbon allocated to microbial growth over the total carbon used in metabolism, neither of the models explicitly simulates $^{13}\text{C}/^{14}\text{C}$ or ^{18}O dynamics in microbial metabolisms. Thus, it is not unexpected that the CUE-SOC slopes differ between the data assimilation and meta-analysis although the microbial model may yield a CUE-SOC slope close to that from the meta-analysis. In the future, process models need new model structures that explicitly represent

CUE as measured in field experiments so that we can directly fuse field-measured CUE into model simulation to better understand the underlying mechanisms of CUE and its consequent impacts on SOC storage.

In the revised manuscript, we included this point as a caveat of this study in L133 - L139 and Supplementary Discussion. Response Letter Table 1 is presented as Supplementary Table 5 of the revised manuscript.

Response Letter Table 1 | Unstandardized coefficients of CUE-SOC relationship in the mixed-effects model with different isotopic-derived CUE data in the meta-analysis. CUE, depth and mean annual temperature (MAT) were set as the fixed effects to SOC content. The study source was set as the random effect.

		Intercept	CUE	Depth	MAT
¹³ C/ ¹⁴ C derived relationship, $n_{obs} = 21$, $n_{study} = 6$ $SOC \sim CUE + Depth + MAT + (1 Study\ Source)$, explained variation = 79%					
Fixed Effects	Estimates	-22.67	16.28	6.61	0.04
	Std. Error	36.91	47.14	2.50	1.61
	t value	-0.64	0.35	2.61	0.026
	P	0.55	0.73	0.020	0.98
Random Effects	Standard Deviation	36.33	NA	NA	NA
¹⁸ O derived relationship, $n_{obs} = 111$, $n_{study} = 10$ $SOC \sim CUE + Depth + MAT + (1 Study\ Source)$, explained variation = 46%					
Fixed Effects	Estimates	14.99	61.06	-0.72	0.17
	Std. Error	7.33	12.56	0.21	0.35
	t value	2.04	4.86	-3.46	0.48
	P	0.046	<0.0001	0.0007	0.63
Random Effects	Standard Deviation	10.51	NA	NA	NA

More discussion could also be provided on the microbial model CUE and parameter sensitivity analysis in the main text. How relevant is it? What does this mean physiologically for microbes and their enzyme production? How can this vary temporally? The response letter contains some discussion where Response Letter Fig. 2 is presented, but I could not find this in the main text (apologies if I missed it). How much did the value of beta vary in the assimilated global model? I assume that beta is ‘allo_slope’ in Extended Data Table 8; if so, please check for consistency in parameter names.

R3.2d: We thank the referee for pointing out the typo. We also thank this referee for these important questions that are worthwhile to explore. We have added more discussion about the physiological meaning of the results from the parameter sensitivity analysis using the microbial model in the caption of Extended Data Fig. 3 (L1001 – L1005 of the revised manuscript) as follows: “*Lower values of the exponent β tend to cause positive relationships between CUE and SOC. This means that less-than-linear scaling of enzyme production and biomass does not allow decomposition to speed up at high CUE (i.e., the argument illustrated in Fig. 1b), whereas necromass contributions to SOC become more important and drive the positive CUE-SOC relationship (i.e., the argument illustrated in Fig. 1a)*”.

In the present study, we only focused on the results of the CUE-SOC relationship at steady state. The temporal changes of the β value can be estimated only if we have data collected over time, which may be explored in the future. Moreover, Response Letter Fig. 2 showed the distribution of β in the microbial model after data assimilation with the 2,500 representative SOC profiles across the world. The β value varies across orders of magnitudes but mostly remains around 1. When the β value is less than 1, a positive CUE-SOC relationship more likely emerges as shown in the Extended Data Fig. 3. However, the β value is not the only parameter that determines a positive or negative CUE-SOC relationship. Many other processes (e.g., microbial mortality rate and SOC decomposition rate) in the microbial model could influence the final CUE-SOC relationship. The sensitivity analysis with the microbial model was to show that either a positive or a negative CUE-SOC relationship could emerge with different combinations of parameter values.

We corrected the ‘allo_slope’ in Extended Data Table 8 (now the Supplementary Table 8 in the revised manuscript) to ‘beta’ to keep consistent with discussion.

We understand the referee’s desire for us to present all these interesting results in the main text. Unfortunately, we have limited space that allows us to present only key findings instead of all these interesting topics. We apologize for not being able to include more details on these questions in the main text.

Response Letter Fig. 2 | Distribution of the allometric slope in the microbial model after data assimilation with 2,500 representative SOC profiles across the world. Black dashed line indicates the value of 1. Note that the x axis has logarithmic scale.

3. Robustness and predictability.

Given the sparse global distribution (Fig. S1), could the authors comment further on the robustness of their results in data-poor regions and the predictability across studies? Would adding another study elsewhere change the results? What about (the lack of) data in the tropics and in deeper soils (e.g., below 30cm), where model results are then presented? Indeed, there was no/little data for these regions, but the results and patterns in declining explanatory power were used to imply process – for example, in Response Letter Fig. 4: “Declining explanatory power of CUE to the variation in SOC with soil depths indicates a shift from biotic-dominated to abiotic-dominated SOC accumulation.” I understand that data paucity is often a problem for such questions, but this is important when making global conclusions.

R3.3a: We thank the referee for the comment. It is true that the data we used in the meta-analysis did not cover all the ecoregions across the world. In the manuscript, we also mentioned that “*To explore whether the CUE-SOC relationship obtained from local experimental measurements is widespread across the globe, we retrieved CUE from globally*

distributed vertical SOC profiles” (L105 - L107). In the revised manuscript (L415 - L416), we further highlight the data scarcity in specific ecoregions: “*Our meta-analysis covered main ecoregions of the world yet there is still a lack of data in rainforest, desert, and tundra*”. The data coverage in our data assimilation is much wider than the meta-analysis and covered all the main ecoregions of the world (Response Letter Fig. 3). And the results from data assimilation with the two models all agree with those from the meta-analysis, showing a positive CUE-SOC relationship globally. In the future, more CUE data from field experiments are desirable to further explore the role of CUE at different soil depths in regulating global SOC storage. Moreover, the declining explanatory power of CUE to the variation in SOC with soil depths (Extended Data Fig. 4 of the revised manuscript) was drawn from the data assimilation results with either the representative 2,500 or all 52,819 vertical SOC profiles across the globe, where a minimum observation depth of 50 cm for all the soil profiles was guaranteed (L455 - L458 and Extended Data Fig. 1 of the revised manuscript).

In the revised manuscript, we discussed the scarcity of data in the meta-analysis (L415 - L416) and the coverage of representative SOC profiles (L626 - L627). Response Letter Fig. 3 is presented as Supplementary Fig. 7 of the revised manuscript.

Response Letter Fig. 3 | Coverage of data used in the meta-analysis with 132 data sets (a), microbial model data assimilation with the subset of 2,500 vertical profiles (b), and CLM5 data assimilation with all the 52,819 profiles (c).

With regards to the robustness of regression slopes and intercepts, the question (to me at least) is less about subsampling the 1,000 soil profiles for a mixed effects model, and more about the (stratified) sampling from the 52,000 WoSIS profiles to the 1000 profiles in the data assimilation. If different 500-1000 soil profiles were subsampled (covering the same

climate-weighted domains) from the WoSIS profiles, would the data assimilation results of CUE differ? This would be a better test of robustness, as opposed to subsampling the data assimilation output. More information on this subsampling would also be helpful, following on L628-631. How many points were selected from each climate zone and how much heterogeneity is there between points in a given climate zone? On this note, it would also be helpful to show that the (1) synthesis data, (2) 1000 subsampled points, and (3) 52,000 WoSIS profiles cover similar regions of a multi-dimensional covariate space.

R3.3b: We thank the referee for the comment. The coverage of the data used in different methods is shown in Response Letter Fig. 3. While it is true that the data used in the meta-analysis did not cover all the ecoregions, the SOC data used in data assimilation (either with the representative sites or all data sets) present good coverage of all main ecoregions of the world.

In Extended Data Fig. 5 (now Supplementary Fig. 1), we applied 1,000-time bootstrapping to address the robustness of the regression slope and intercept of CUE-SOC relationship. Specifically, subsamples with different sizes were further sampled from the 1,000 (2,500 in this revision) representative data assimilation results with replacement. The subsamples could either cover all the main ecoregions (as with the 2,500 representative profiles shown in Response Letter Fig. 3) or only a subset of them. Theoretically, after sufficient iterations of bootstrapping (e.g., 1,000 times in this study), the distribution of the regression slopes and intercepts (as shown in Supplementary Fig. 1) should indicate the range of all possible values that could occur. Our results in Supplementary Fig. 1 showed the robustness of the positive CUE-SOC relationship from the results of data assimilation with the microbial model.

To address the referee's concerns, we added another 1,500 data assimilation results using the microbial model in this revision. We have continued data assimilation with the microbial model in the past two months after we submitted the revised manuscript in case that more results are needed during this round of revision. All the newly added 1,500 SOC profiles were sampled following the same procedure with what we did for the previous 1,000 profiles. The newly added data assimilation results did not change any main conclusions discussed in this manuscript, confirming the robustness of our sampling method and conclusions drawn from the results. All the related figures (e.g., Fig. 1, Extended Data Figs. 4 and 8, Supplementary Figs. 1, 4, 9) and tables (e.g., Extended Data Table 2, Supplementary

Tables 3) in the revised manuscript have been updated based on the 2,500 data assimilation results.

What if other covariates were included in the mixed-effect models of SOC-CUE relationships? For example, soil moisture, clay content, NPP, etc. At least for the data assimilation results, where these covariates are reported.

R3.3c: We thank the referee for this good suggestion. We tried different model structures using data assimilation results with CLM5 and the whole SOC profile dataset. The fixed effects in mixed-effects models were set as either bulk density, cation exchange capacity, clay content, or NPP in addition to CUE. We found that the positive relationship between CUE and SOC was supported by all these mixed-effects models with different structures (Response Letter Table 2).

In the revised manuscript, the Response Letter Table 2 is presented as Supplementary Table 4.

Response Letter Table 2 | Unstandardized coefficients of CUE-SOC relationship (considering fixed effects from other covariates) in the mixed-effects model with CLM5 data assimilation results. CUE and one other environmental variable (i.e., bulk density, cation exchange capacity, clay content, or NPP) were set as the fixed effects to logarithmic SOC content. Climate types that soil profiles belong to were set as the random effect. We applied a mixed-effects model that considered random intercepts with common slopes to test CUE-SOC relationship (i.e., $\log_{10}(\text{SOC}) \sim \text{CUE} + \text{Selected Variable} + (1|\text{Climate Types})$). The random effects size $n_{\text{climate}} = 12$. The total observation size $n_{\text{obs}} = 52280$. The observation size is different from the total soil profile size (i.e., 52,819) because the environmental variable or climate type information is not available for some profiles.

		Intercept	CUE	Variable
log (SOC)~CUE + log (Bulk Density) + (1 Climate Types), explained variation = 60%				
Fixed Effects	Estimates	3.66	3.41	-0.53
	Std. Error	0.063	0.028	0.0049
	t value	58.32	120.67	-106.92

	P	<0.0001	<0.0001	<0.0001
Random Effects	Standard Deviation	0.17	NA	NA
$\log(SOC) \sim CUE + CEC + (1 Climate\ Types)$, explained variation = 57%				
Fixed Effects	Estimates	-0.48	3.87	0.0095
	Std. Error	0.056	0.028	0.00011
	t value	-8.52	137.76	86.72
	P	<0.0001	<0.0001	<0.0001
Random Effects	Standard Deviation	0.19	NA	NA
$\log(SOC) \sim CUE + Clay\ Content + (1 Climate\ Types)$, explained variation = 52%				
Fixed Effects	Estimates	-0.62	4.54	0.0041
	Std. Error	0.069	0.028	0.00011
	t value	-9.00	160.60	37.65
	P	<0.0001	<0.0001	<0.0001
Random Effects	Standard Deviation	0.24	NA	NA
$\log(SOC) \sim CUE + \log(NPP) + (1 Climate\ Types)$, explained variation = 58%				
Fixed Effects	Estimates	-2.42	5.97	0.22
	Std. Error	0.054	0.030	0.0023
	t value	-45.17	198.70	95.17
	P	<0.0001	<0.0001	<0.0001
Random Effects	Standard Deviation	0.17	NA	NA

What if the SCE method was used to assimilate SOC data into the CLM5 model as well? As a check on using a different assimilation method for the microbial model.

R3.3d: We appreciate this referee for this good suggestion. We applied the same SCE algorithm used in the microbial model data assimilation to CLM5 at the 2,500 representative sites. The results (Response Letter Fig. 4) showed high agreement on the retrievals of different mechanisms between the SCE and MCMC methods. Therefore, results from both data assimilation methods (i.e., MCMC and SCE) support the positive CUE-SOC relationship.

Moreover, we found that while SCE-retrieved CUE presented high correlation with those by the MCMC method (Pearson correlation = 0.7, $P < 0.0001$, Response Letter Fig. 4a), the SCE method also enlarged the variance of retrieved CUE. The enlarged variance of CUE further caused a lower CUE-SOC slope in regression. Using the same mix-effects model

structure (i.e., $\log_{10}(\text{SOC}) \sim \text{CUE} + (1|\text{Climate Types})$), we obtained $\log_{10}(\text{SOC}) = 0.77 + 1.07 * \text{CUE}$ from the results with SCE method ($n_{\text{sample}} = 2207$, $P(\text{CUE}) < 0.0001$, $P(\text{Intercept}) < 0.0001$, explained variation = 51%). Based on the above results, we conclude that the different CUE-SOC slopes between the microbial model and CLM5 in Fig. 2 of the main text partly resulted from the different methods of data assimilation.

The different variance of retrieved CUE by the SCE and MCMC methods resulted from the difference in optimization algorithms. The SCE method first searches regions in the parameter space to give similarly low-cost function values (i.e., the difference between observations and model simulations, lower values indicates better fit of model simulation to observations) and then uses the genetic algorithm to shuffle those regions to find a point that gives a global optimum (i.e., the lowest cost value). The MCMC method, on the other hand, is based on the Bayesian statistics and generates the probability distribution (i.e., *posterior* distributions) of estimated parameters which are defined by the *prior* parameter ranges. While the point estimate given by the SCE method is usually in the range of the *posterior* distributions of parameters with the MCMC method, the two estimates do not have a one-to-one relationship partly due to different shapes of the *posterior* distributions from the MCMC method, such as normal or skewed distributions. Because we used the mean value of the posterior distribution as the point estimate from the MCMC method, parameters that have skewed posterior distributions (e.g., CUE; the skewed posterior distributions of CUE-related parameters have been discussed in ref²⁵) will present increased variance among different sites by the SCE method.

We clarified the reasons of different CUE-SOC slopes from different methods in L133 - L139 of the revised manuscript and offered the detailed reasons in the Supplementary Discussion. Notwithstanding the difference in the absolute values of slopes of CUE-SOC relationships from different methods, the key conclusion of this study is that high CUE promotes global SOC storage, and this conclusion has been fully supported by our current models and methods.

Response Letter Fig. 4 | Retrievals of mechanisms using different data assimilation methods (i.e., MCMC and SCE methods) with CLM5 and the 2,500 representative soil profiles. Black lines are 1:1 lines and blue lines are the results of linear regression. Shaded areas are the 95% confidence of linear regression results.

4. Proportional changes in ‘mechanisms’ are still difficult to interpret.

I think the authors may have misunderstood the R3.3a and R3.3b comments. I appreciate their responses and the fact that they do not know the parameter space at each grid cell. However, I feel that having some sense of what the relative change in a given parameter means is critical. I completely understand what the environmental modifier is from a modeling perspective, but again, it is a model construct and a 7% decrease is not very meaningful without context. I feel the results would be much more impactful if the authors could give this context. For example, in the case of the environmental modifier, how much does it vary spatially in the best fit model? What does this mean with regards to differences in temperature and moisture?

R3.4a: We thank the referee for this good suggestion. The revised sentences (L199 – L209) now read as: *“The result (Fig. 4b) shows that a 10% increase in global SOC storage (equivalent to an additional 219 Pg SOC accumulation worldwide) requires a 3% increase in CUE (i.e., an increase of global median CUE from 0.39 to 0.40), a 7% decrease in environmental modifiers (i.e., equivalent to a decrease of global median temperature from*

12.2°C to 10.7°C, or a decrease of global median precipitation from 527mm to 445mm), an 8% decrease in baseline decomposition rate (i.e., decrease of the global median from 0.034 yr⁻¹ to 0.031 yr⁻¹), a 10% increase in carbon input (i.e., increase of global net primary productivity from 63 Pg C yr⁻¹ to 69 Pg C yr⁻¹), a 19% increase in vertical transport (i.e., increase of the global median from 0.046 yr⁻¹ to 0.055 yr⁻¹), or a 123% increase in input allocation (out of reasonable boundary in model simulation).” The changes in environmental modifier is associated with changes in global temperature and precipitation according to the statistical models between environmental modifier (ξ) and mean annual temperature (MAT) (i.e., $\log(\xi) \sim \alpha MAT + \beta$, regression $\alpha = 0.048$, $\beta = -2.43$, $R^2 = 0.52$) and precipitation (MAP) (i.e., $\xi \sim \alpha MAP + \beta$, regression $\alpha = 0.00014$, $\beta = 0.088$, $R^2 = 0.39$) across the globe as described in L806 - L812.

Again, I am not sure ‘mechanism’ is the correct word for environmental modifier, baseline decomposition, or most of the ‘mechanisms’ herein, even carbon use efficiency. Some of these are model constructs and others are, as the authors acknowledge in their response, emergent properties resulting from various underlying processes. I appreciate the added text on L714, but I still think ‘mechanisms’ might not be the correct terminology.

R3.4b: We totally understand the referee’s concern. However, ‘mechanism’ probably is the best term that we came up after serious consideration to refer these ensembles of processes (as we stated in L743 - L744). According to Merriam-Webster dictionary, mechanism is defined as “*the fundamental processes involved in or responsible for an action, reaction, or other natural phenomenon*”. Baseline decomposition is a process of SOC decomposition at a reference environmental condition. Carbon use efficiency is a process of microbial use of carbon for growth relative to the amount of metabolized carbon. Environmental modifier represents a suite of microbial processes in response to environmental changes. For example, microbes may become less active in metabolism and more dormant when soil becomes very dry. Microbes metabolize faster and respire more when temperature increases up to an optimum. Models represent these processes at aggregate levels so that we can examine system dynamics of land carbon cycle at the global scale. That said, we are eager to hear suggestions from this referee.

Other comments:

The authors state (L222) that the permutation analysis indicates that soil physical properties explain CUE's spatial variability more than climate or soil chemical properties, but the latter two appear quite close. Disregarding them seems misleading.

R3.5a: We thank the referee for pointing this out. We did not disregard them. The sentence also expresses soil physical properties are more explanatory than climate and soil chemical properties. To make it clearer we changed the sentence (L217 – L219) to “*A permutational analysis (Methods) indicates that soil physical properties are more explanatory to CUE's spatial variability than climatic, soil chemical, vegetation, and geographic variables.*”

The Fig. 2 caption should mention how many profiles were in the meta-analysis. Currently only the WoSIS and profile subsets are listed.

R3.5b: Following the suggestion, we added the number of data sets used in the meta-analysis to Fig. 2 caption in the revised manuscript. There is no vertical profile of field measurement of microbial CUE reported in the literature.

The description of the empirical constraints as one pool being X-fold the size of another could be changed to percentages, which are more common (as in the studies cited). For example, if I understood correctly following on L615, SOC had to exceed MIC by at least 50-fold over the entire profile means that MIC can be at most 2% of SOC over the profile. Similarly, DOC could be at most 10% of SOC in the entire profile and 1% of SOC in the topsoil. The constraint on DOC in the entire profile seems quite high... maybe this was a typo by the authors? The cited Guo et al. paper estimates DOC at around 1% of SOC across depths. In any case, writing these constraints as percentages of SOC would be more intuitive to most readers.

R3.5c: We thank the referee for the good suggestion. We revised the description in the manuscript (L609 – L616).

We set two constraints at different soil depths on DOC. The data reported in Guo et al.'s paper aggregated in the surface soil (i.e., 0 – 30cm)²⁶ and we set the constraint as “*DOC for the top 30cm should be no more than 1% of the SOC*”. Meanwhile, we found that for specific soil type (e.g., Spodosols) at the deeper horizons, the DOC could be as high as about

7% of SOC²⁷. We therefore applied the second constraint as “DOC of the entire soil depth should be no more than 10% of SOC”.

L595 – should this read enzyme turnover rate? Also worth editing the Extended Data Fig. 4 caption to state enzyme production and turnover, as done in Response Letter Fig. 2. Tau is not well-defined or discussed in the manuscript and only really appears in Extended Data Fig. 4, unless I missed something. I would suggest putting a subscript ‘enz,decay’ on tau to prevent confusion with microbial turnover rates. I also wonder why only beta was selected for the sensitivity analysis and not the enzyme production rate k?

R3.5d: We apologize for the typo. The “enzyme production rate” in L595 of the original manuscript (L587 in the revised manuscript) should be “turnover time for enzyme production ($\tau_{enz,prod}$)”. It is not the turnover time for enzyme decay. The two parameters used in the sensitivity analysis are both from the equation $PROD_{ENZ} = k_{enz,prod}MIC^\beta$ (L565 of the revised manuscript). The $\tau_{enz,prod}$ in the Extended Data Fig. 4 (now Extended Data Fig. 3 in this revision) equals to the inverse of $k_{enz,prod}$. We revised the description in L587 and added a subscript “enz, prod” on τ in Extended Data Fig. 3 as suggested.

In Extended Data Fig. 4 (i.e., Response Letter Fig. 2 – please use only one labeling scheme in future iterations if the figure is replicated exactly) shouldn’t the case where beta = 1 (with tau = 1) depicted in dark red be the same as the case for tau = 1 (with beta = 1) in light blue? Why are these different? Apologies if I missed something, but either way, clarification in the caption would be helpful. Also, why are values for beta = 0.3 to 0.5 excluded with CUE = 0.2?

R3.5e: We thank the referee for pointing this out. The referee is right that the case where beta = 1 (with tau = 1) is exact the same with the case tau = 1 (with beta = 1). We revised Extended Data Fig. 4 (now Extended Data Fig. 3 in this revision) as suggested.

Because the microbial model is a nonlinear system, some ill parameter combinations could lead the system to a condition where there is no steady state solution. In Extended Data Fig. 3, we only changed the two target parameters but kept all others as default values, which could cause those ill parameter value combinations. The missed points in Extended Data Fig. 3 are the case where we cannot find the steady state solution under such parameter

combinations. In the revised manuscript, we added explanations about the missing points in the caption of Extended Data Fig. 3 (L999 - L1000).

Why is it not possible to analytically solve for soil carbon pool sizes in the microbial models (L598), especially in the simple one used here? It may be more complicated due to nonlinearities, but can be explicitly solved, as done in past microbial modeling papers.

R3.5f: We thank the referee for the good question. In most cases, solutions for nonlinear differential equations cannot be analytically solved but can only be numerically analyzed. The previous microbial modelling studies either did the numerical forward integration²⁸ as we did in this study, or linearized the model before solving it²⁹, or applied the Newton-Raphson method (or its equivalents) to approximate the steady state solution^{30,31}. Newton-Raphson and linearization methods may be less accurate than the numerical forward integration. The microbial model we applied in this model is relatively complicated (four nonlinear differential equations for each soil layer and a total of 80 nonlinear differential equations for connections among 20 soil layers) for those approximation methods. Thus, we chose the forward integration method to reach the steady state solution.

Using the word ‘preservation’ for soil carbon may imply longevity and could be misleading. I suggest using the word storage or stocks instead. Also, the words accumulation and loss suggest a dynamic change, whereas here you are using a space-for-time substitution, which often does not inform change under new conditions. Another reviewer brought up this issue of space-for-time, and I think the text could use more caveats on this point. The revised sentences (as in response R1.3a) do not mention potential problems with space-for-time (e.g., Abramoff et al. 2019) and currently state that “projecting changes in SOC needs to use full distributions of potential values of these mechanisms in future research” (L217). However, it’s not only about the full distribution of potential values, but also how current patterns in these values can or cannot inform projections.

R3.5g: We thank the referee for the suggestion. We scrutinized related sentences to avoid interpreting the spatial patterns into temporal domains. For example, we changed the “preservation” to “storage” in L54 and L232 of the revised manuscript.

In the sentence “projecting changes in SOC needs to use full distributions of potential values of these mechanisms in future research”, we understand that examining the full

distributions of potential values of mechanisms includes testifying current pattern for space-for-time substitution. To make it clearer, we revised the sentence to “*While the two sensitivity analyses evaluated the relative importance of the six mechanisms, projecting future changes in SOC given this new knowledge will need to examine the full spaces of parameters related to these mechanisms and changing environments, and consider microbial physiological acclimation and genetic adaptations.*” (L211 – L215).

References

- 1 Luo, Y. & Schuur, E. A. Model parameterization to represent processes at unresolved scales and changing properties of evolving systems. *Global change biology* **26**, 1109-1117 (2020).
- 2 Wang, S., Luo, Y. & Niu, S. Reparameterization Required After Model Structure Changes From Carbon Only to Carbon-Nitrogen Coupling. *Journal of Advances in Modeling Earth Systems* **14**, e2021MS002798 (2022).
- 3 MacBean, N. *et al.* Quantifying and Reducing Uncertainty in Global Carbon Cycle Predictions: Lessons and Perspectives From 15 Years of Data Assimilation Studies with the ORCHIDEE Terrestrial Biosphere Model. *Global Biogeochemical Cycles* (accepted) (2022).
- 4 Malik, A. A. *et al.* Land use driven change in soil pH affects microbial carbon cycling processes. *Nature communications* **9**, 1-10 (2018).
- 5 Wang, C. *et al.* Large-scale importance of microbial carbon use efficiency and necromass to soil organic carbon. *Global Change Biology* (2021).
- 6 Kallenbach, C. M., Frey, S. D. & Grandy, A. S. Direct evidence for microbial-derived soil organic matter formation and its ecophysiological controls. *Nature communications* **7**, 1-10 (2016).
- 7 Sinsabaugh, R. Enzymic analysis of microbial pattern and process. *Biology and Fertility of Soils* **17**, 69-74 (1994).
- 8 Manzoni, S., Taylor, P., Richter, A., Porporato, A. & Ågren, G. I. Environmental and stoichiometric controls on microbial carbon-use efficiency in soils. *New Phytologist* **196**, 79-91 (2012).
- 9 Schimel, J. P. & Schaeffer, S. M. Microbial control over carbon cycling in soil. *Frontiers in microbiology* **3**, 348 (2012).

- 10 Sinsabaugh, R. L., Shah, J. J. F., Findlay, S. G., Kuehn, K. A. & Moorhead, D. L. Scaling microbial biomass, metabolism and resource supply. *Biogeochemistry* **122**, 175-190 (2015).
- 11 Chen, J. *et al.* Costimulation of soil glycosidase activity and soil respiration by nitrogen addition. *Global Change Biology* **23**, 1328-1337 (2017).
- 12 Jian, S. *et al.* Soil extracellular enzyme activities, soil carbon and nitrogen storage under nitrogen fertilization: A meta-analysis. *Soil Biology and Biochemistry* **101**, 32-43 (2016).
- 13 Jia, X. *et al.* Effects of nitrogen enrichment on soil microbial characteristics: From biomass to enzyme activities. *Geoderma* **366**, 114256 (2020).
- 14 Chen, J. *et al.* Soil carbon loss with warming: New evidence from carbon-degrading enzymes. *Global change biology* **26**, 1944-1952 (2020).
- 15 Chen, J. *et al.* A keystone microbial enzyme for nitrogen control of soil carbon storage. *Science Advances* **4**, eaaq1689 (2018).
- 16 Karhu, K. *et al.* Microbial carbon use efficiency and priming of soil organic matter mineralization by glucose additions in boreal forest soils with different C: N ratios. *Soil Biology and Biochemistry* **167**, 108615 (2022).
- 17 Schimel, J. P. & Weintraub, M. N. The implications of exoenzyme activity on microbial carbon and nitrogen limitation in soil: a theoretical model. *Soil Biology and Biochemistry* **35**, 549-563 (2003).
- 18 Allison, S. D., Wallenstein, M. D. & Bradford, M. A. Soil-carbon response to warming dependent on microbial physiology. *Nature Geoscience* **3**, 336-340 (2010).
- 19 Wieder, W. R. *et al.* Explicitly representing soil microbial processes in Earth system models. *Global Biogeochemical Cycles* **29**, 1782-1800 (2015).
- 20 Georgiou, K., Abramoff, R. Z., Harte, J., Riley, W. J. & Torn, M. S. Microbial community-level regulation explains soil carbon responses to long-term litter manipulations. *Nature Communications* **8**, 1-10 (2017).
- 21 German, D. P., Marcelo, K. R., Stone, M. M. & Allison, S. D. The Michaelis–Menten kinetics of soil extracellular enzymes in response to temperature: a cross-latitude study. *Global Change Biology* **18**, 1468-1479 (2012).
- 22 Ye, J. S., Bradford, M. A., Dacal, M., Maestre, F. T. & García-Palacios, P. Increasing microbial carbon use efficiency with warming predicts soil heterotrophic respiration globally. *Global Change Biology* **25**, 3354-3364 (2019).

- 23 Schimel, J., Weintraub, M. N. & Moorhead, D. Estimating microbial carbon use efficiency in soil: Isotope-based and enzyme-based methods measure fundamentally different aspects of microbial resource use. *Soil Biology and Biochemistry* **169**, 108677 (2022).
- 24 Hu, J., Huang, C., Zhou, S. & Kuzyakov, Y. Nitrogen addition to soil affects microbial carbon use efficiency: Meta-analysis of similarities and differences in ^{13}C and ^{18}O approaches. *Global Change Biology*.
- 25 Tao, F. *et al.* Deep Learning Optimizes Data-Driven Representation of Soil Organic Carbon in Earth System Model Over the Conterminous United States. *Frontiers in Big Data* **3**, doi:10.3389/fdata.2020.00017 (2020).
- 26 Guo, Z. *et al.* Soil dissolved organic carbon in terrestrial ecosystems: Global budget, spatial distribution and controls. *Global Ecology and Biogeography* **29**, 2159-2175 (2020).
- 27 Neff, J. C. & Asner, G. P. Dissolved organic carbon in terrestrial ecosystems: synthesis and a model. *Ecosystems* **4**, 29-48 (2001).
- 28 Wang, Y. p. *et al.* Microbial activity and root carbon inputs are more important than soil carbon diffusion in simulating soil carbon profiles. *Journal of Geophysical Research: Biogeosciences* **126**, e2020JG006205 (2021).
- 29 Shi, Z., Crowell, S., Luo, Y. & Moore, B. Model structures amplify uncertainty in predicted soil carbon responses to climate change. *Nature communications* **9**, 2171 (2018).
- 30 Abramoff, R. Z. *et al.* Improved global-scale predictions of soil carbon stocks with Millennial Version 2. *Soil Biology and Biochemistry* **164**, 108466 (2022).
- 31 Wieder, W., Grandy, A., Kallenbach, C., Taylor, P. & Bonan, G. Representing life in the Earth system with soil microbial functional traits in the MIMICS model. *Geoscientific Model Development* **8**, 1789-1808 (2015).

Referee #4 (Remarks to the Author):

I have been asked by the journal editor to review the authors' response to referee #1 comments, since she/he was not available to handle this revision. Hence, I have focused my revision only on such author response.

Response: We greatly appreciate referee #4 for agreeing to review the manuscript and providing constructive comments and suggestions. Below are our point-by-point responses to referee #4's comments.

1. Assigning process to pattern:

I consider this the most important concern from referee#1, and I concur with her/his assessment that using only a first-order soil C model (CLM5) to address the outcome of the CUE vs SOC relationship was biased towards the positive trend, because of the absence of a feedback from microbial biomass towards SOC. The authors followed the suggested indications and used a four-pool microbial-explicit model as in Allison et al (2010) to check whether a positive CUE vs SOC relationship was also found with a microbial model. Importantly, authors run this model using the data assimilation framework for increased generality and assessed parameter sensitivity. The results of the microbial model also support the positive CUE vs SOC relationship, and I appreciate the comprehensive response letter on this regard. I do think the ms has greatly improved with this addition.

R4.1a: We greatly appreciate the positive evaluation by the referee on our first-round revision of the manuscript.

However, I do have a small concern about the previous analyses. If the outcome (+ or -) of the CUE-SOC relationship depends on the parameter values of enzyme production used, I wonder how this has affected the interpretation made by authors (i.e. non-linear scaling of enzyme production and microbial biomass hinders higher SOM decomposition at high CUE). I have not found any discussion about this, but I guess this is needed to guide the reader and avoid confusion, either in the caption of Ext Data Fig. 4 and/or in the main text.

R4.1b: We thank the referee for this good comment. We added sentences in the caption of Extended Data Fig. 3 (L1001 - L1005) to discuss how the parameters involved in enzyme

production influence the final CUE-SOC relationship as “*lower values of the exponent β tend to cause positive relationships between CUE and SOC. This means that less-than-linear scaling of enzyme production and biomass does not allow decomposition to speed up at high CUE (i.e., the argument illustrated in Fig. 1b), whereas necromass contributions to SOC become more important and drive the positive CUE-SOC relationship (i.e., the argument illustrated in Fig. 1a)*”. We hope the added sentences are helpful for readers to better understand the results of the sensitivity analysis using the microbial model.

2. Correlation vs. causation:

I think the authors have done a good job incorporating the study ID as a random factor in the meta-analysis as suggested by referee#1. Actually, different structures were tried and all results point to a similar CUE-SOC relationship.

R4.2a: We thank the referee for the positive evaluation on our revised meta-analysis.

As it was expected with such a hierarchical dataset in the meta-analysis (132 observations from only 16 experimental sites/papers), the variance explained in SOC by the mixed model is way larger (twice or even more in some model structures) than the fixed model. Although the number of sites (46) is actually not that bad, the meta-analytical database is quite poor in spatial spread and in number of studies. This clearly reflects the scarcity of field studies measuring CUE, but it is also a weakness of the meta-analysis and should be reflected in the text (or at least put into context with the need to be complemented with the other analytical approaches exploring the CUE-SOC relationship).

R4.2b: We thank the referee for this good comment. It is true that the data we used in the meta-analysis did not cover all the ecoregions across the world. In the manuscript, we also mentioned that “*To explore whether the CUE-SOC relationship obtained from local experimental measurements is widespread across the globe, we retrieved CUE from globally distributed vertical SOC profiles...*” (L105 – L107). The data coverage from our data assimilation is much wider than the meta-analysis and covered all the main ecoregions of the world (Response Letter Fig. 3). And the results from data assimilation with the two models all agree with the meta-analysis, showing a positive CUE-SOC relationship globally. In the future, more CUE data from the field experiments are desired.

In the revised manuscript, we further discussed the scarcity of data in the meta-analysis (L415 – L416) and the coverage of representative SOC profiles (L626 - L627). Response Letter Fig. 3 is presented as Supplementary Fig. 7 of the revised manuscript.

I found intriguing why results in Extended Data Table 2 show a positive MAT effect on SOC but a negative MAT effect on microbial biomass in Extended Data Table 2. What is driving this?

R4.2c: We appreciate the referee for asking this good question. In Extended Data Table 2 (now the Extended Data Table 1 of the revised manuscript) the mixed-effects model (i.e., $SOC \sim CUE + Depth + MAT + (1|Study\ Source)$) presented a positive regression slope for MAT-SOC relationship (0.13, $P = 0.72$). Using the same structure with the microbial model data assimilation results, the mix-effects model (Supplementary Table 2) showed a negative MAT effect on SOC (-0.038, $P = 0.010$). The results may result from the fact that CUE and soil depth had strong effects on SOC and, at the same time, CUE showed significant correlation with MAT (Supplementary Fig 4). In the regression, the residual of SOC did not show significant relationship with MAT ($P = 0.72$) after controlling the effects of CUE and soil depth on SOC. The microbial biomass data was measured independently from the SOC data. The results from mix-effects model using the microbial biomass data indicates that after considering effects of CUE and soil depth, the residual microbial biomass can still be explained by MAT ($P = 0.010$). Overall, the regression slope of MAT in the mixed-effects model using the SOC data may be influenced by its correlation with CUE and the positive value of the slope was not significant ($P > 0.05$). In the future, more SOC data is required to further confirm the value of MAT effect in the mixed-effects model.

It seems many of your studies from the meta-analysis also include some management treatments that may confound these effects (eg. Fertilization, drought). Are you including these plots too in the meta-analysis or only the control ones? I did not find any info about this in the methods.

R4.2d: We thank the referee for pointing this out. In our study, when the data was from manipulation experiments, we only collected data at the control plots. We clarified this point in L412 - L413 of the revised manuscript.

Overall, I think the concern of the previous reviewer regarding correlation vs causation has been satisfactorily dealt with in this revision.

R4.2e: We thank the referee for the positive evaluation on the revised manuscript.

Other points:

3a. Space for time substitution: This has been nicely incorporated into the main text.

R4.3a: We thank the referee for the positive evaluation.

3b. Discussions of literature findings that disagree with your results: I don't think that removing the text on other plausible outcomes of the CUE vs MAT relationship (actually the negative one) is the solution to avoid confusion. Actually, I concur with referee #1 assessment that your findings must be discussed and confronted with previous results from the literature, specifically to those that also use model-data integration to represent the implications of microbial physiology across large spatial scales. Actually, the negative CUE vs MAT relationship found in Ye et al 2019 GCB was somehow robust to variation in C substrate availability, although it's true that other resources such as N and P were not assessed. With all this being said, I think this context should be brought back to the main text, and this should be useful to readers to better frame your results within the wider literature, even if this means bringing up apparently contradictory results. In my opinion, the different results here are profoundly influenced by the lack of CUE field assessments across diverse ecosystems, which is affecting our scientific understanding, but still we have to move forward, and your paper clearly helps to do so in a very compelling manner.

R4.3b: We thank the referee for the valuable comment. The different MAT-CUE relationships between our study and Ye et al.'s study mainly resulted from different methods of estimation and/or measurement of CUE. All the CUE values in our meta-analysis were based on measurement using isotopically ($^{13}\text{C}/^{14}\text{C}$ or ^{18}O , as reported in Supplementary Table 1) labelled substrates to estimate the proportion of an organic substrate that is converted into new forms relative to the amount consumed in metabolism. And previous studies using the similar isotope-labelling method all supported a negative MAT-CUE relationship^{5,18,32}.

Ye et al. (2019) used CUE estimated from extracellular enzyme data³³. Instead of defining CUE with the physiological partitioning of substrates, the enzyme-derived CUE was

defined from a perspective of elemental stoichiometry. The core assumption under this method states that “*CUE of an organism is a function of the difference between its elemental requirements for growth and the composition of environmental substrate*”³³. While there is still lack of studies testing the consistency of measured CUE from these two methods, the enzyme-derived CUE conceptually has additional assumptions in comparison with the isotope-derived CUE to assess how microbes shift resource use in response to substrate stoichiometry²³.

In our study, the estimated CUE by the two biogeochemical models is conceptually consistent with the physiologically-defined CUE measured by isotopic data. Thus, the negative MAT-CUE relationship retrieved from both of the models agrees with the pattern emerged from the meta-analysis (Response Letter Fig. 5).

In the revised manuscript, we discussed the MAT-CUE relationship in L157 – L163 and included detailed discussion in Supplementary Discussion. Response Letter Fig. 5 is presented as Supplementary Fig. 4 in the revised manuscript.

Redacted

3c. This has been nicely incorporated into the main text.

R4.3c: We thank the referee for the positive evaluation.

3d. PRODA approach: The new paragraph makes a good introduction for the non-specialized readers about the goals of this approach, and benefits, from previous models. This helps to identify the novelty and to differentiate this study from previous ones.

R4.3d: We thank the referee for the positive evaluation.

Minor issue: Revise the reference numeration in Extended Data Table 1, as this is not matching the reference list.

R4.3e: We apologize for the mistake. We have corrected the numeration in Extended Data Table 1 (now the Supplementary Table 1) in this revision.

References

- 5 Wang, C. *et al.* Large-scale importance of microbial carbon use efficiency and necromass to soil organic carbon. *Global Change Biology* (2021).
- 18 Allison, S. D., Wallenstein, M. D. & Bradford, M. A. Soil-carbon response to warming dependent on microbial physiology. *Nature Geoscience* **3**, 336-340 (2010).
- 23 Schimel, J., Weintraub, M. N. & Moorhead, D. Estimating microbial carbon use efficiency in soil: Isotope-based and enzyme-based methods measure fundamentally different aspects of microbial resource use. *Soil Biology and Biochemistry* **169**, 108677 (2022).

- 32 Conant, R. T. *et al.* Temperature and soil organic matter decomposition rates—
synthesis of current knowledge and a way forward. *Global change biology* **17**, 3392-
3404 (2011).
- 33 Sinsabaugh, R. L. *et al.* Stoichiometry of microbial carbon use efficiency in soils.
Ecological Monographs **86**, 172-189 (2016).

Reviewer Reports on the Second Revision:

Referee #3 (Remarks to the Author):

The study by Tao et al. uses carbon use efficiency (CUE) measurements, global soil profiles, and two soil models to demonstrate the importance of CUE for soil carbon storage. I still have several concerns regarding the approach and the interpretability of the results.

Definition and calculation of “mechanisms”:

I stand by my concern on the word ‘mechanism’, especially for the model constructs (e.g., environmental modifier, baseline decomposition) presented. Even by the authors’ dictionary definition, these constructs cannot be called ‘fundamental processes’ in the ecological and climatic sense that they intend. They are largely dependent on the models used and do not have true empirical analogs. They also aggregate a number of components in their definition. For example, even the authors say that “Environmental modifier represents a suite of microbial processes in response to environmental changes.” This does not mean that ‘environmental modifier’ is a process or a mechanism in the true sense of those words. I would call these values ‘model constructs’ or ‘model parameters/components’, as they depend directly on the models used. Even the calculation of bulk CUE is model-specific and predetermines the results to some degree.

In R3.2a, the authors reiterate the way they calculate system-level CUE. I do not disagree with the way bulk CUE was calculated for CLM5, nor that it is compared to the empirical results per se. However, I do maintain that comparing this system-level CUE (and its relative importance) to the various individual ‘mechanisms’ used to calculate it is unfair and misleading. Especially with the way the results are interpreted and framed in the abstract and manuscript.

What about the relative importance of these ‘mechanisms’ using a microbial model? For example, what would Figs. 3 and 4 look like using the model in Fig. 2b on a global scale, instead of CLM5? (More on this comment below.)

Robustness and predictability:

I still have reservations about the robustness of the results in data-poor regions. In their response, the authors state that the coverage in their data assimilation was much wider than the meta-analysis, and that their results (e.g., with regards to depth) are drawn from the data assimilation. Of course the global profiles are more representative, but the ‘retrieved’ CUE are model-derived and there are no observational data to validate the retrieval in these regions (tundra, tropics, and in deeper soil layers). A number of the main results hinge entirely on model-derived CUE values using CLM5.

Furthermore, where the meta-analysis and data assimilation results can be compared, they differ quantitatively, especially for the CLM5-derived results (e.g., CUE-SOC slopes vary by over a factor of 5). I am still not sure what the consequences of these differences are on the main results, which are largely based on CLM5. If this slope for the CLM5 results was an order of magnitude (or 5-fold)

lower, as in the meta-analysis, would the relative importance of the 'mechanisms' still hold? I appreciate the added discussion of slopes in the SI, but this doesn't address how the main findings would differ if the CUE-SOC slope in CLM5 was more similar to the observational values.

The assessment of how representative the meta-analysis sites and global profiles are could also include a multi-dimensional covariate space, in addition to the climate space. For example, how do SOC values or vegetation types compare across the meta-analysis sites and global profiles?

CUE-SOC relationships:

Expanding on comments above, the fact that the slopes (or retrieved CUE values) vary so much between the different data and model sources (or even data assimilation methods) is a little unsettling. The authors even state that "different methods to measure or estimate CUE strongly influence the slopes of the CUE-SOC relationship" – if these slopes vary so much, then how does this impact the downstream relative importance in the main findings of Fig. 3 and Fig. 4? That is, if the slope wasn't as large for the CLM5 results (e.g., ~ 0.8 instead of ~ 4), would this weaken the relative importance of CUE or change the other downstream global predictions?

What if the retrieved bulk CUE values for the global profiles were compared across the models? That is, what if bulk CUE retrieved using the microbial model (as in Fig. 2b) was compared to that retrieved using CLM5 (as in Fig. 2c) across all the representative global profiles. What does this tell us about the model-dependency of the bulk CUE definition?

Minor comments:

Fig. 2 caption – by "132 data sets" do you mean data points/measurements (i.e., a pair of CUE and SOC measurements)? I would change this to be more clear. And similarly in other figures (e.g., Supplementary Fig. 4).

The caption of Fig. 2 should include some information about the 'climate' color bar. Is this mainly temperature? If so, why not explicitly show temperature?

The caption of Fig. 3 should reflect that these results are based on CLM5.

Referee #4 (Remarks to the Author):

I am fully satisfied with the author responses to my concerns and those of referee #1. Overall, the rationale and the motivation of the study is much better outlined, and this is a great improvement from the previous version. I think the text is now much more consequent and less-biased, which is important for the robustness of the broad message from the study: microbial physiology is fundamental to predict the global spatial variation of SOC.

Minor concern:

I wonder if the width of arrows and/or boxes lines in Fig. 1 should be changed to guide the reader towards the outcome of each conceptual model. For instance in Fig. 1a, a positive CUE vs SOC will lead to higher SOC accumulation than in Fig. 1b. Conversely, in Fig. 1b, a negative CUE vs SOC will lead to lower SOC accumulation via higher CO₂ losses (at equal levels of plant C inputs as in Fig. 1a).

Author Rebuttals to Second Revision:

Summary of responses to comments by referees

We greatly appreciate the two referees for offering us constructive comments on the second-round revision of our manuscript. Referee #4 was satisfied with our responses to her/his comments on the previous version of this manuscript and gave us suggestions on minor revision of Fig. 1. Referees #3 offered more comments on the revised manuscript. All these comments stimulated us to further improve our manuscript. We have addressed all the concerns as described in this letter of responses and accordingly revised the manuscript. Here is a summary of the revision of our manuscript.

1. Definition and calculation of model components: Following suggestions by referee #3, we changed the word “mechanism” mostly to “model components” in the revised manuscript to refer the system-level value of different categories of processes in soil carbon cycle (R3.1a). Despite we used parameters that may belong to other components to calculate the system-level CUE, the way we calculate CUE does not introduce strong correlations between CUE and other model components (R3.1b). Thus, we can compare the relative importance of CUE to global SOC storage in comparison with other components. Moreover, we provide an analog to explain why the information of other components was used when calculating system-level CUE (R3.1b).

2. Robustness of the importance of CUE to global SOC storage: Referee #3 raised concerns about the robustness of the importance of CUE to global SOC storage when only CLM5 was used in analysis whereas the spatial CUE-SOC slopes differ so much between CLM5 and the microbial model. In this revision, we used the PRODA approach to generalise the microbial model data assimilation results from the 2,500 SOC profiles to the global scale and conducted the same sensitivity analyses as we did with CLM5 (R3.1c). The results showed that CUE is still the most important component to global SOC storage and its spatial variation. It largely results from the fact that information contained in big data constrains the microbial model to the extent that the Michaelis-Menten kinetics of decomposition in the microbial model can be approximated by the first-order kinetics as configured by CLM5. We also discussed robustness of the results retrieved from the data using the microbial model in the point-to-point responses (R3.1c) below.

3. Robustness of CUE definition: Referee #3 raised concerns about whether the model retrieved CUE after data assimilation can be validated by observational CUE measurements and comparable between different models. In the revised manuscript, we showed that predicted CUE values by the PRODA-optimised CLM5 are positively correlated with the field-measured CUE values in the meta-analysis (R3.2a). Meanwhile, CUE values retrieved from CLM5 and the microbial model are also significantly correlated (R3.3b).

We hope that our responses and revision of the manuscript are satisfactory to the referees. And we look forward to further feedbacks and comments.

Below are our point-by-point responses (in blue) to referees' comments (in black).

Point-by-point responses to referees' comments

Referee #3 (Remarks to the Author):

The study by Tao et al. uses carbon use efficiency (CUE) measurements, global soil profiles, and two soil models to demonstrate the importance of CUE for soil carbon storage. I still have several concerns regarding the approach and the interpretability of the results.

Response: We greatly appreciate referee #3 for the comments on our second revision. We have addressed all the concerns as described below.

Definition and calculation of “mechanisms”:

I stand by my concern on the word ‘mechanism’, especially for the model constructs (e.g., environmental modifier, baseline decomposition) presented. Even by the authors’ dictionary definition, these constructs cannot be called ‘fundamental processes’ in the ecological and climatic sense that they intend. They are largely dependent on the models used and do not have true empirical analogs. They also aggregate a number of components in their definition. For example, even the authors say that “Environmental modifier represents a suite of microbial processes in response to environmental changes.” This does not mean that ‘environmental modifier’ is a process or a mechanism in the true sense of those words. I would call these values ‘model constructs’ or ‘model parameters/components’, as they depend directly on the models used. Even the calculation of bulk CUE is model-specific and predetermines the results to some degree.

R3.1a: We thank the referee for the suggestions. We changed “mechanisms” mostly to “model components” in the revised manuscript to refer the system-level values of different categories of processes in soil carbon cycle.

In R3.2a, the authors reiterate the way they calculate system-level CUE. I do not disagree with the way bulk CUE was calculated for CLM5, nor that it is compared to the empirical results per se. However, I do maintain that comparing this system-level CUE (and its relative importance) to the various individual ‘mechanisms’ used to calculate it is unfair and misleading. Especially with the way the results are interpreted and framed in the abstract and

manuscript.

R3.1b: This referee's comment makes us to think more about this issue. The formula for calculating the system-level CUE could hinder a circular argument due to the possible correlations between CUE and other model components used in the calculation. Indeed, it is possible that CUE appears to cause larger variation in SOC compared to other components if the model components interact positively in the calculation of CUE. We examined whether modelled CUE correlates with the model components. The system-level CUE is weakly correlated with baseline decomposition (Pearson correlation coefficient = 0.07, $n = 45230$, $P < 0.0001$). The environmental modifiers mathematically cancel out in the CUE definition (Equation 3 of the revised manuscript).

An analog may be helpful for understanding whether or not it is fair to compare system-level CUE to various components used to calculate it. Let us consider a person who manages a factory. The factory processes crude oil and produces two oil products. The crude oil that comes to the factory is the reactant in the industrial process (equivalent to substrate input in soil carbon cycle) at cost c . The two oil products are produced at rates r_1 and r_2 , and consume crude oil with substrate use efficiency (i.e., efficiency of conversion from reactant, that is crude oil, to products) a_1 and a_2 . The two products have different prices p_1 and p_2 . Let us assume a simple case that production rates of the two products are functions of prices, $r_1 = b_1 p_1$ and $r_2 = b_2 p_2$. Also let us assume that all the products produced are sold. The total revenue Z of the factory equals sale Y minus cost X of the crude oil. That is

$$Z = Y - X = \sum_i r_i p_i - c \left(\sum_i \frac{r_i}{a_i} \right) = \sum_i b_i p_i^2 - c \left(\sum_i \frac{b_i p_i}{a_i} \right) \quad (1)$$

To maximize revenue of the factory, the manager has two options: (1) to improve crude oil use efficiency (i.e., a_i) and (2) to adjust selling prices of the two oil products (i.e., p_i). The factory manager may evaluate which of the two options is more effective to maximize the factory revenue. In this case, the manager needs to calculate the factory-wide crude oil use efficiency (A_F) and factory-wide price index (P_F). The factory-wide crude oil use efficiency (A_F) is

$$A_F = \sum_i a_i \frac{\frac{b_i p_i}{a_i}}{\sum_i \frac{b_i p_i}{a_i}} = \sum_i \frac{b_i p_i}{\sum_i \frac{b_i p_i}{a_i}} \quad (2)$$

In the above equation, $\frac{b_i p_i}{a_i}$ is the consumption rate of crude oil for product i . $\sum_i \frac{b_i p_i}{a_i}$ is the total consumption rate of crude oil for the whole factory. In equation 2, the calculation of the

factory-wide crude oil use efficiency (A_F) uses information of price p_i . The factory-wide price index (P_F) is

$$P_F = \sum_i p_i \frac{b_i p_i}{\sum_i b_i p_i} \quad (3)$$

The factory manager compares the sensitivity of factory revenue Z to A_F vs. P_F . This is equivalent to our study, where we used information of model components to calculate the system-level CUE (equivalent to A_F) before we did the sensitivity analysis of global carbon storage to CUE and other components.

What about the relative importance of these ‘mechanisms’ using a microbial model? For example, what would Figs. 3 and 4 look like using the model in Fig. 2b on a global scale, instead of CLM5? (More on this comment below.)

R3.1c: We appreciate the referee for the suggestion. Per the suggestion, we applied the PRODA approach to the microbial model using the 2,500 site-level data assimilation results and conducted the two sensitivity analyses to evaluate the relative importance of microbial CUE and other five components to global SOC storage as we did with CLM5. The PRODA-optimized microbial model presents similar spatial patterns of the model components (Response Letter Fig. 1) with those from CLM5 (Fig. 3 of the main text). The Pearson correlation coefficient between CLM5 and the microbial model retrieved CUE is 0.47 ($n = 45230$, $P < 0.0001$) across the globe (Response Letter Table 1). The agreement on the retrieved components from the global vertical SOC data with the two different models indicates that the spatial patterns of different components shown in Fig. 3 of the main text are robust.

Response Letter Fig. 1 | Maps of global SOC stock, residence time, and related model components retrieved with the microbial model. The maps were obtained from 2,500 representative soil profiles using the PRODA approach with the microbial model. Values shown are predicted by the best-guess model calibrated using all available. The grey shaded area in panel a indicates the grids where the steady-state was not reached after 20,000 years simulation. Boxplots represent the SOC properties and model components in different predefined climate zones (Supplementary Fig. 8). The lower, middle, and upper hinges show the first, median, and third quartiles of the distribution, respectively. Whiskers in the boxplot represent the 1.5 interquartile range from the hinges. Units for maps and boxplots are the same.

Response Letter Table 1 | Correlation between CLM5 and microbial model retrieved components. Results shown in the table are the global simulation results.

	R	P value
Microbial carbon use efficiency	0.47	<0.0001
Baseline decomposition (log10 transformed)	0.52	<0.0001
Environmental modifiers	0.76	<0.0001

Vertical transport (log10 transformed)	0.14	<0.0001
Carbon input allocation	0.36	<0.0001

The results of the two sensitivity analyses using the microbial model (Response Letter Fig. 2) are also consistent with those by CLM5 (Fig. 4 of the main text). Response Letter Fig. 2 shows that the relative importance of CUE to global SOC storage is still the highest among the six model components in the microbial model as in CLM5. Although the two data assimilation methods, i.e., MCMC and SCE, result in different slope values of the spatial CUE-SOC relationship (see more discussion below in R3.3a), our result shows that the elasticities of global SOC storage to proportional changes of CUE (i.e., the slopes shown in Fig. 4b of the main text and Response Letter Fig. 2b) are similar between the microbial model and CLM5 after data assimilation. The data assimilation likely helps constrain both the models to behave similarly. As we discussed in the previous revision, for example, the Michaelis-Menten constant for SOC decomposition in the microbial model is constrained to be much higher than the SOC substrate concentration after data assimilation. The nonlinear SOC decomposition in the microbial model can be approximated by the first-order kinetics as configured in CLM5.

Response Letter Fig. 2 | Relative importance of different model components using PRODA-optimized microbial model. Grey dashed lines in panel a indicate the explained spatial variation in SOC storage and the estimation of global SOC storage by the best guess model after 20,000 years simulation. Note that because in the microbial model, grids in the boreal regions did not reach steady state after 20,000 years simulation, the estimation on the global SOC stock is lower than that estimate by CLM5 model.

While we kept the methodological procedure of the PRODA approach and sensitivity analyses (e.g., the structure of neural network and the way of calculating system-level values of different model components) identical with what we did with CLM5, the following four issues need to be discussed.

First, the formula to calculate system-level CUE is identical for both the CLM5 and microbial model. But the parameters we included to calculate the system-level CUE in the sensitivity analyses are different in the microbial model from CLM5 as the former incorporates more processes, such as enzyme dynamics. Parameters that are used to describe the enzyme production (i.e., enzyme production rate $k_{enz,prod}$, and allometric slope β) and the following catalytical reactions in SOC decomposition and assimilation (i.e., Michaelis constants for decomposition $k_{m,decom}$ and assimilation $k_{m,assim}$) are directly linked with changes of microbial physiology and thus system-level CUE in this study. Thus, we used the system-level CUE of the microbial model to present the CUE-SOC slope in Fig. 2 of the main text to be consistent with CLM5. In addition to transfer coefficients among different carbon pools, we also merged the spatial variation of the above-mentioned parameters in the first sensitivity analysis and proportionally changed their values in the second sensitivity analysis to investigate the relative importance of system-level CUE to global SOC storage.

Second, we used different methods to obtain the steady-state values of soil carbon storage for CLM5 and the microbial model in the sensitivity analyses. CLM5 in the matrix form can be semi-analytically solved to obtain the steady-state values when parameter values change. Therefore, we can computationally afford to do sensitivity analyses as we did in this study. Notably, the sensitivity analysis has rarely been done with global carbon cycle models without the matrix form so far due to the computational cost, even with linear models like CLM5. Since the microbial model is nonlinear, we cannot semi-analytically solve the model to obtain the steady state values of soil carbon storage when parameter values change for the sensitivity analyses. We used a simulation approach to approximate the steady state of SOC storage by running the model for a maximum simulation time of 20,000 years. For the majority of the total 45,230 grids over the globe, the steady-state SOC storage was reached after 20,000-year simulation. In about 6,000 grids (mostly in boreal regions, as indicated by the grey shaded area in Response Letter Fig. 1a) where the steady-state SOC storage was not reached, we kept the carbon storage in the last year of simulation as the proxy of the steady-state SOC storage even with some degree of underestimation (as shown in Response Letter

Fig. 2). Even so, different model components were all relatively stable after 20,000 years' simulation in the microbial model. Therefore, the two sensitivity analyses are still valid.

Third, the microbial model that was optimized by the PRODA approach explains 28% spatial variation in the SOC storage, which is very low in comparison with 57% spatial variation of the SOC storage explained by the PRODA-optimized CLM5. Multiple reasons could lead to the low performance of the PRODA-optimized microbial model. For example, strong equifinality (See Luo et al. 2009¹ for description of equifinality) occurred in site-level data assimilation with the SCE method. It may not only influence the numerical value of the spatially positive CUE-SOC slope (see R3.3a for more details), but also make it difficult for the neural network to predict global patterns of parameters retrieved from site-level data assimilation. While the equifinality happens all the time in data assimilation, its degree varies with different methods. The MCMC method requires much more computation time to optimize a process-based model but appears to result in less equifinality than the SCE method. While PRODA-optimized CLM5 based on the SCE data assimilation only explains 42% of the spatial variation in SOC storage, its counterpart based on the MCMC method explains 57%. Moreover, estimated parameters in the nonlinear microbial model may be too irregular to reflect global patterns of processes underlying SOC storage and thus dampen the relationships between parameters and environmental covariates that can be retrieved by the neural network.

Fourth, microbial models themselves and methods to assimilate big data into microbial models are all at a stage of exploration. Like many other microbial models, the microbial model used in this study has more complex structure in describing soil carbon cycle but less power to explain the spatial variations in global SOC storage than CLM5 as described in the above paragraph. Despite dozens of microbial models have been proposed in the past decade or so, their performance at the global scale remains to be evaluated. Moreover, to the best of our knowledge, no global-wide data assimilation study has been conducted with microbial models due to the extremely high computation cost. The SCE method applied in this study is computationally much more efficient than the MCMC method so that it allows us to do global data assimilation at 2,500 representative SOC profiles with the microbial model. However, CLM5 parameter values estimated at individual sites using the SCE method tend to hit edge (i.e., the lower or upper limit of the prior range) more than the MCMC method (Response Letter Fig. 3). While the MCMC method has been widely used in data assimilation with land carbon cycle models, efficacy of the SCE method for that use has not been systematically evaluated. The combination of the complex, nonlinear

microbial model with a to-be-evaluated SCE method for data assimilation may result in estimated parameters that do not reflect global patterns that a neural network can retrieve to generalize its site-level data assimilation results to the global scale. Even so, we greatly appreciate this referee for inspiring us to experiment.

In the revised manuscript, we present the results of the microbial model together with CLM5 (main text L150 – L156, L188 – 189, L194 – L196, L220 – 224). We described the details of the PRDOA approach (L680 – 681, L768 – 779) and two sensitivity analyses (L861 – 872) using the microbial model. We highlight the limitation of the results using the microbial model (L156 – 160, L711 – 736). Response Figures 1-3 are presented as Supplementary 4, 6, and 12 in the revised manuscript, respectively. Response Letter Table 1 is shown as Supplementary Table 6 of the revised manuscript.

Response Letter Fig. 3 | Point estimates by the SCE method tend to hit edges of prior ranges of parameters. The data assimilation results at one SOC profile (ID: 160557, Longitude: 119.96°W, Latitude: 39.43°N) by both the MCMC and SCE methods. Violin plots indicate the posterior distributions of each parameter using the MCMC method. The lower, middle, and upper hinges of the boxplots show the first, median, and third quartiles of the posterior distribution, respectively. Whiskers in the boxplot represent the 1.5 interquartile range from the hinges. The red marks indicate the point estimates by the SCE method.

Robustness and predictability:

I still have reservations about the robustness of the results in data-poor regions. In their

response, the authors state that the coverage in their data assimilation was much wider than the meta-analysis, and that their results (e.g., with regards to depth) are drawn from the data assimilation. Of course the global profiles are more representative, but the ‘retrieved’ CUE are model-derived and there are no observational data to validate the retrieval in these regions (tundra, tropics, and in deeper soil layers). A number of the main results hinge entirely on model-derived CUE values using CLM5.

R3.2a: We thank the referee for the question on whether our PRODA retrieved CUE can be validated by the observational data. In this revision, we chose the grids on the global CUE map (as shown in Fig. 3 of the main text) that are nearest to the sites where the meta-analysis data are from to compare the PRODA-retrieved CUE with observations. Response Letter Fig. 4 indicates that the global CUE retrieved from vertical SOC profiles present significant positive correlation with observational data (Pearson correlation coefficient = 0.39, $n = 132$, $P < 0.0001$) although no information at these observation sites was used in the PRODA approach to inform the model. This significant positive correlation supports the robustness of the global CUE retrievals by the PRODA-optimized CLM5. Yet, there is no doubt that more data are needed to further validate the model-derived CUE in the future.

We mentioned the correlation between PRODA-predicted CUE and field measurements in L165 – L167 of the revised manuscript.

Response Letter Fig. 4 | Correlation between CUE values measured in field experiments and predicted by PRODA approach with CLM5. We located sites of data collection in the meta-analysis to the nearest grid in the CUE global map generated by PRODA-optimized CLM5.

Furthermore, where the meta-analysis and data assimilation results can be compared, they differ quantitatively, especially for the CLM5-derived results (e.g., CUE-SOC slopes vary by over a factor of 5). I am still not sure what the consequences of these differences are on the main results, which are largely based on CLM5. If this slope for the CLM5 results was an order of magnitude (or 5-fold) lower, as in the meta-analysis, would the relative importance of the ‘mechanisms’ still hold? I appreciate the added discussion of slopes in the SI, but this doesn’t address how the main findings would differ if the CUE-SOC slope in CLM5 was more similar to the observational values.

R3.2b: We thank the referee for the comments. In R3.1c, we have shown that the sensitivity analyses with the microbial model consistently support that the microbial CUE is the most important model component to global SOC storage and its spatial variation in comparison to

other five components in spite of differences in the spatial CUE-SOC slopes between the microbial model and CLM5.

The assessment of how representative the meta-analysis sites and global profiles are could also include a multi-dimensional covariate space, in addition to the climate space. For example, how do SOC values or vegetation types compare across the meta-analysis sites and global profiles?

R3.2c: We appreciate the referee for the suggestion. In the revised manuscript, we discussed the coverage of different sources of data in multi-dimensional covariate spaces. Response Letter Fig. 5 shows the distribution of SOC data in different climate, soil texture and land cover types. While the vertical SOC profiles cover all the covariate types, the representative subset of the SOC profile presents more even distribution of data in those regions. Data sites in the meta-analysis cover nearly all the land cover types, yet they only include part of the climate and soil texture types. In the future, more data about SOC storage and CUE should be collected in those data sparse regions.

In the revised manuscript, we substituted the original Supplementary Fig. 7 with Response Letter Fig. 5 and re-numbered it as Supplementary Fig. 9.

Response Letter Fig. 5 | Coverage of different sources of data in multi-dimensional covariate spaces. Panels show the percentage of data sites located at different climate (a), soil texture (b) and land cover (c) types in the meta-analysis with 132 data sets, microbial model data assimilation with the subset of 2,500 vertical profiles, and CLM5 data assimilation with all the 52,819 profiles. For different climate types: Af, Am and Aw are tropical rainforest, monsoon and savannah climates, respectively. BW and BS are arid desert and steppe climates, respectively. Cs, Cw and Cf are temperate climates with dry summer, dry winter, and without dry season, respectively. Ds, Dw and Df are cold climates with dry

summer, dry winter, and without dry season, respectively. E is polar climate. For different soil texture, Cl is clay, SiCl is silty clay, SaCl is sandy clay, ClLo is clay loam, SiClLo is silty clay loam, SaClLo is sandy clay loam, Lo is loam, SiLo is silty loam, SaLo is sandy loam, Si is silt, LoSa is loamy sand, Sa is sand.

CUE-SOC relationships:

Expanding on comments above, the fact that the slopes (or retrieved CUE values) vary so much between the different data and model sources (or even data assimilation methods) is a little unsettling. The authors even state that “different methods to measure or estimate CUE strongly influence the slopes of the CUE-SOC relationship” – if these slopes vary so much, then how does this impact the downstream relative importance in the main findings of Fig. 3 and Fig. 4? That is, if the slope wasn’t as large for the CLM5 results (e.g., ~ 0.8 instead of ~ 4), would this weaken the relative importance of CUE or change the other downstream global predictions?

R3.3a: We thank the referee for the comments. As we discussed in the R3.1c and R3.2b, the spatial CUE-SOC slope does not influence the importance of CUE to global SOC storage. The different data assimilation methods affect the spatial slope of CUE-SOC but have no influence in the importance of CUE on global SOC storage as quantified in the sensitivity analyses (Response Letter Fig. 2).

This referee’s comment also stimulated us to further investigate the reasons why CUE-SOC slopes differ between the different data and model sources. First, we observed that the range of estimated CUE of the microbial model is much larger with the SCE method (from 0.2 to 0.9) than that of the CLM5 with the MCMC method (from 0.3 to 0.6) while estimated SOC storage was similar between the two methods (Fig. 2b vs. 2c in the main text). This enlarged range of CUE leads to a much smaller CUE-SOC slope with the microbial model than that with CLM5. To understand the reason why the range of estimated CUE of the microbial model is much larger with the SCE method than that of CLM5 with the MCMC method, we used the SCE method for data assimilation with CLM5. The estimated CUE of CLM5 with the SCE method has a similar range with that of the microbial model, leading to similar CUE-SOC slopes between the two models (Response Letter Fig. 6).

Response Letter Fig. 6 | CUE-SOC relationship retrieved from the microbial model (MIC) and CLM5 with the two data assimilation methods (i.e., SCE and MCMC) at 2,500 representative SOC profiles. Black lines are the partial coefficients from mixed-effects model regression. CUE was set as the fixed effects to logarithmic SOC content. Climate types that soil profiles belong to were set as the random effect. We applied a mixed-effects model that considered random intercepts with common slopes to test CUE-SOC relationship (i.e., $\log_{10}(\text{SOC}) \sim \text{CUE} + (1|\text{Climate Types})$). The random effects size $n_{\text{climate}} = 12$.

Then, we explored why the SCE method can result in a much larger range of estimated CUE than the MCMC method even with the same CLM5 model. We picked two SOC profiles that have either a low or high CUE value after data assimilation using both the SCE and MCMC methods with CLM5. The MCMC method allows us to generate a posterior distribution of system-level CUE at each of these two profiles (i.e., the violin plots shown in Response Letter Fig. 7). The mean of the posterior distribution is a representative value of CUE that enters the calculation of the CUE-SOC slope. However, while the estimated system-level CUE from the SCE method goes to higher when the posterior mean CUE value is high and lower when the posterior mean CUE value is low (blue points in Response Letter Fig. 7). Thus, the range of estimated CUE with the SCE method is larger than that with the MCMC method, leading to a smaller CUE-SOC slope.

Response Letter Fig. 7 | CUE estimates with two data assimilation methods. Violin plots and boxplots show the posterior distributions of CUE after data assimilation with the MCMC method at two SOC profiles (ID: 57680 and 160557, Longitudes: 1.35°E and 119.96°W, Latitudes: 13.60°N and 39.43°N).. Points with different colours indicate the point estimates on CUE with the two data assimilation methods (i.e., MCMC and SCE). The lower, middle, and upper hinges of the boxplots show the first, median, and third quartiles of the distribution, respectively. Whiskers in the boxplot represent the 1.5 interquartile range from the hinges

Second, we observed that CUE measurements in the meta-analysis present a similar range (from 0.04 to 0.8) with that retrieved from the microbial model (from 0.2 to 0.9) (Fig. 2a and 2b of the main text) yet the slope of the CUE-SOC relationship in the meta-analysis is smaller than that in the microbial model. The different slope values in the meta-analysis and the microbial model may be attributed to the smaller range of SOC covered in the meta-analysis (from 1.5 gC kg⁻¹ to 140 gC kg⁻¹) compared to the microbial model results (from 0.5 gC kg⁻¹ to 607 gC kg⁻¹). Lack of CUE estimates from soils at the lower and upper ends of the SOC range may cause a smaller SOC-CUE slope in the meta-analysis.

In the revised manuscript, we added the above discussion in the Supplementary Discussion. Response Letter Figures 6 and 7 are shown as Supplementary Figs. 13 and 14, respectively.

What if the retrieved bulk CUE values for the global profiles were compared across the models? That is, what if bulk CUE retrieved using the microbial model (as in Fig. 2b) was compared to that retrieved using CLM5 (as in Fig. 2c) across all the representative global profiles. What does this tell us about the model-dependency of the bulk CUE definition?

R3.3b: We thank the referee for the suggestions. The retrieved CUE values from CLM5 and microbial model data assimilation at the 2,500 representative profiles are significantly and positively correlated (Pearson correlation coefficient = 0.45, $n = 2,500$, $P < 0.0001$, Response Letter Fig. 8). Meanwhile, as we discussed in R3.1c, the PRODA-predicted global map of CUE based on the microbial model data assimilation also shows significant positive correlation with the results based on CLM5 (Response Letter Table 1). The agreement on the CUE retrievals from the two different models indicates the robustness of our results on CUE shown in this study. The spatial patterns as well as the importance of CUE to global SOC storage discussed in this study are consistent across the different model structures after data assimilation.

In the revised manuscript, Supplementary Table 6 shows the correlation between CLM5 and microbial model derived CUE across the globe.

Redacted

Minor comments:

Fig. 2 caption – by “132 data sets” do you mean data points/measurements (i.e., a pair of CUE and SOC measurements)? I would change this to be more clear. And similarly in other figures (e.g., Supplementary Fig. 4).

R3.4a: We thank the referee for the suggestion. We changed “data sets” to “measurements” in the revised manuscript.

The caption of Fig. 2 should include some information about the ‘climate’ color bar. Is this mainly temperature? If so, why not explicitly show temperature?

R3.4b: The referee is correct that the “climate” colour bar is mainly about temperature. Following the suggestion, we changed the colour bars of Fig. 2b – 2c and Extended Data Fig. 5a – 5c to represent mean annual temperature in the revised manuscript.

The caption of Fig. 3 should reflect that these results are based on CLM5.

R3.4c: We thank the referee for the suggestion. We added a note in caption of Fig. 3 to clarify that the results are based on CLM5.

Reference:

- 1 Luo, Y. *et al.* Parameter identifiability, constraint, and equifinality in data assimilation with ecosystem models. *Ecological Applications* **19**, 571-574 (2009).

Referee #4 (Remarks to the Author):

I am fully satisfied with the author responses to my concerns and those of referee #1. Overall, the rationale and the motivation of the study is much better outlined, and this is a great improvement from the previous version. I think the text is now much more consequent and less-biased, which is important for the robustness of the broad message from the study: microbial physiology is fundamental to predict the global spatial variation of SOC.

Response: We greatly appreciate the positive evaluation of referee #4 on our revision and manuscript.

Minor concern:

I wonder if the width of arrows and/or boxes lines in Fig. 1 should be changed to guide the reader towards the outcome of each conceptual model. For instance in Fig. 1a, a positive CUE vs SOC will lead to higher SOC accumulation than in Fig. 1b. Conversely, in Fig. 1b, a negative CUE vs SOC will lead to lower SOC accumulation via higher CO₂ losses (at equal levels of plant C inputs as in Fig. 1a).

R4.1a: We thank the referee for the suggestion. We revised Fig. 1 of the main text (as shown in Response Letter Fig. 9) to better illustrate how different regulation pathways of CUE will influence the final SOC storage.

Response Letter Fig. 9 | Revised figure for Fig. 1 of the main text.

Editorial Note:

An adjudicating referee (referee #5) was asked to comment on the debate between referee #3 and the authors.

Reviewer Reports on the Third Revision:

Referee #5 (Remarks to the Author):

Review of manuscript: "Microbial carbon use efficiency promotes global soil carbon storage"

Overall assessment:

The manuscript's central thesis is that microbial physiological carbon use efficiency is the dominant control on soil carbon accrual worldwide. The authors find support for this argument in the results of data assimilation and sensitivity analysis applied to a first order linear and a microbially explicit soil carbon model.

This manuscript is technically impressive, generally easy to follow, and ambitious in its scope. However, in my view its central argument is based on a misapplication of the concept of "microbial carbon use efficiency". The transfer coefficients in first-order soil carbon models are not equal to "microbial carbon use efficiency", although these two things are clearly related. I am going to elaborate on this point because I do not think it should be readily dismissed.

Transfer coefficients may indeed implicitly incorporate microbial carbon use efficiency — and yet they also implicitly incorporate other factors. For instance, as originally conceptualized in the 1980s, these parameters represented the efficiency with which organic matter was "humified" or converted to more "recalcitrant" compounds; or alternatively the efficiency with which minerals protect or stabilize SOC (Parton et al. 1987; Parton et al. 1994). For instance, the transfer coefficient in DAYCENT — a prototypical first order SOC model that forms the basis for the SOC representation in CLM5 — was originally conceptualized as soil-texture dependent, reflecting the efficiency of physical stabilization by the mineral matrix (Parton et al. 1987). Transfer into the passive SOC pool was described in a purely empirical way to approximate SOC residence times, with no reference to microbial physiology and no clear physiological interpretation (Parton et al. 1987). In this sense the coefficients describe an emergent process that combines microbial physiology and the other abiotic (non-physiological) processes.

My point here is not that we should remain bound by old (and perhaps outmoded) theoretical interpretations (like the recalcitrance concept, which I am not defending); rather, it is to emphasize that the transfer coefficients in first order SOC models are nebulous, operationally defined parameters, and that our interpretation of them is somewhat subjective. In this case, there is no strong basis for interpreting these coefficients as "microbial CUE" alone to the exclusion of other factors.

From a more technical standpoint, first order soil carbon models do not directly simulate microbial metabolism and growth; hence there is no clear way to invert a first order model to estimate microbial CUE as defined in the manuscript (Lines 47-49). The "bulk CUE" index discussed in this manuscript might be a rough proxy for microbial CUE and ought to be correlated with it at some level (as the authors clearly show), but demonstrating that a model component is correlated with a measurement is not convincing evidence that the model component and the measurement are

synonymous. Consequently, the analysis focused on CLM5—which makes up the backbone of the paper—does not actually isolate the role of microbial physiology in carbon sequestration, and the manuscript overreaches by assigning too much importance to microbial physiology.

Evaluation of responses to reviewer 3's comments:

I have been asked to specifically evaluate the author's response to the reviewer 3's concerns. Here is my point-by-point assessment, using the authors' codes designating each point of contention:

R3.1a: I concur with reviewer 3 that "model component" is a more appropriate term than "mechanism". This has been addressed adequately, except in Figure 4, where the term mechanism still needs to be replaced.

R3.1b: Here I think that the authors may have missed the point. It does seem true that in a purely mathematical sense, bulk CUE depends to some extent on the factors that it is being compared to, and the authors have done a good job of addressing this issue. However, the fact remains that bulk CUE extracted from CLM5 does not isolate the role of microbial physiology in carbon cycling. Perhaps it is the interpretation rather than the method of calculation that is the real issue (see overall assessment above).

R3.1c: The authors have addressed this concern adequately at some level by expanding the role of the microbial model. I would like to point out, however, that a correlation between bulk CUE extracted from CLM5 and CUE derived from the microbial model is not evidence that CLM5 can isolate microbial physiology; rather it is evidence that the transfer coefficients from CLM5 are a suitable proxy for physiological CUE. Broadly: correlations alone cannot be used to argue that two quantities are equivalent.

In addition, this response to Reviewer 3's comment is illuminating, in that it explains clearly that "system level" CUE was extracted from the microbial model, which was not entirely clear in the main text of the manuscript. I appreciate that retrieving "system level" CUE was important for intercomparability with CLM5, but on the other hand this decision undermines the utility of the microbial model. The microbial model contains an actual CUE parameter which governs partitioning between microbial growth and respiration in a clearly interpretable way. This parameter really might be directly relatable to the CUE measurements collected in the meta-analysis (which cannot be said for CLM5 transfer coefficients). Are the optimized values of this parameter equal to "system level" CUE? If not, what does system level CUE represent?

These questions relate to the broader debate about interpreting CUE measurements. From a more empirical standpoint, CUE can have multiple meanings and interpretations depending on the scale of analysis or the method used (Geyer et al. 2016; Geyer et al. 2019). Does system level CUE isolate the ratio of carbon used for growth versus metabolism? Or is it an index that is also sensitive to other factors like microbial turnover (see Hagerty et al. 2014; Hagerty et al. 2018)? The manuscript would benefit from more explicit consideration of these questions, and some direct discussion of how the "cue_mic" parameter relates to system level CUE.

R3.2a: The authors have addressed this comment by comparing PRODA retrieved CUE estimates to the observations from the meta-analysis. This is good, but it also raises new issues. First, Response Letter Figure 4 clearly shows that CLM5-derived CUE and CUE from field experiments are correlated, but the relationship appears to be strongly biased (i.e., the slope is very far from 1:1). CLM5/PRODA retrieved CUE mostly varies between 0.35 and 0.42, whereas observed CUE values vary between 0.1 and 0.7. This is related to the slope mismatch issue that Reviewer 3 raised earlier and which the authors address at length. I think at some level this mismatch is to be expected because “bulk” or “system level” CUE estimate from CLM5 is only roughly analogous to the CUE being measured in experiments—so we might expect a positive relationship, but not a 1:1 relationship. This supports the interpretation that CLM5-derived CUE is a proxy for true microbial CUE and not identical to it.

I would also like to point out a concern with the meta-analysis dataset. My concern is that the analysis combines ^{13}C -based and ^{18}O -based CUE estimates. It has been shown clearly that these two methods are actually measuring fundamentally different quantities—one substrate specific CUE, the other non-specific (Geyer et al 2019). We really ought to stop treating these measurements as synonymous—no level of statistical analysis can get around the fact that they are based on different assumptions and typically yield very different values (e.g., ~ 0.6 for ^{13}C glucose CUE, versus around 0.3 for ^{18}O CUE). A lot of the variation in compilations of CUE measurements might actually be due to the fact that different ways of measuring or calculating CUE will automatically yield different answers.

R3.2b I concur with Reviewer 3 that the large difference in the CUE-SOC slope in the observations and the CLM5-derived CUE estimates is a major concern (this relates to my response to R3.2a above). It is encouraging that the microbial model produces a closer match, but this does nothing to increase my confidence that CLM5 can be used to represent the experimental data accurately, and the manuscript remains highly reliant on the CLM5 analysis.

R3.2c Here I think the authors have done a good job of addressing Reviewer 3’s comment. However, I have a major concern that is related. I am concerned about the fact that during model optimization, 72,350 soil profiles were originally considered but ultimately only 52,819 were analyzed (27% excluded). In the Methods, it is mentioned that profiles were excluded if they did not show a clear monotonic decline in SOC with depth. It is implied that SOC depth profiles that show a bimodal SOC distribution are the result of measurement errors or some sort of aberrant “geologic process”:

“While these atypical vertical SOC distributions could be caused by geological processes even if not by measurement errors, they may not offer information to help understand processes underlying SOC storage.”

I do not think this is a defensible argument. In particular, an entire soil order – the Spodosol order in the USDA taxonomy – is partly defined by the presence of a sandy, relatively SOC-poor E horizon that overlies a finer textured B horizon rich in reactive Al and Fe oxyhydroxides and SOC. The process of soil development can naturally generate a bimodal SOC distribution in this case: this is most certainly a “process underlying SOC storage”. I am worried that the down-selection procedure has systematically biased this analysis to exclude soils with complex SOC-depth profiles that emerge due

to pedogenesis. If this is true, the analysis might be downplaying the role of minerals in protecting microbial products and hence predetermining its findings at some level.

R3.3a: This response shows that the data assimilation algorithm has a strong influence on the range of system level CUE values extracted from the model. When CLM5 is optimized via the SCE procedure, it yields a CUE SOC relationship several times less steep than if it is optimized by the MCMC procedure. This is an interesting methods comparison and I laud its thoroughness, but ultimately this result indicates that CUE extracted from the data-assimilation / PRODA approach is sensitive to the algorithm used. This makes it very hard to evaluate the degree of agreement between the models, or between the models and observations: to what extent are these results an artifact of the method applied? The fact that SCE and MCMC derived results are correlated is small comfort given the difference in output ranges when different algorithms are applied.

R3.3b: Once again, this response is based on correlations, not a rigorous analysis of the absolute goodness of fit between different model outcomes or the observations. Correlation statistics are not strong evidence that the analysis is robust to the model type or data assimilation approach, particularly given that the absolute values of the CUE estimates occupy very different ranges.

Summary of critiques:

- (1) Microbial CUE has been defined too broadly, and is not a good descriptor of the model components being optimized in the CLM5 case, which do not isolate microbial physiology or represent microbial growth and metabolism.
- (2) The actual CUE parameter in the microbial model is not compared to “system level” CUE, and the exact relationship between these two quantities is not explored.
- (3) CLM5 derived CUE estimates cover a much narrower range than the observations, which undermines the idea that CLM5 derived bulk CUE is equivalent to measured microbial CUE.
- (4) Observed CUE data in the meta analysis combine methods that quantify fundamentally different aspects of CUE.
- (5) Down-selection of soil profiles may be systematically biased to ignore important biogeochemical processes (e.g., podzolization).
- (6) CUE estimates are dependent on the data assimilation algorithm used; correlations are evidence of a relationship, but not equivalence.

Detailed comments:

Lines 113-116: I’m not sure this statement is true, or at least it seems like a rather slanted interpretation of CLM5. Also, as far as I can tell, the reference cited does not actually support this claim, or delve into the details of CLM5 soil biogeochemistry much at all.

Line 523: Here the transfer coefficients in CLM5 are called “microbial CUE”. This is a misrepresentation in my view: these coefficients are analogous to microbial CUE and implicitly include it, but are not equivalent to it, because CLM5 does not represent microbes explicitly.

Lines 229-237: Could the relationship with bulk density simply be because these soils have more organic C?

Line 447: Why not random slopes?

Lines 637-638: What does it mean to optimize so many parameters (23) on a profile-by-profile basis? Aren't there many more unknowns than observations in this case? It seems like there would be a significant danger of overfitting the model. The total number of optimized parameters globally must number in the tens of thousands, unless I am misunderstanding something.

Lines 667-669: Does CLM5 really explicitly simulate mineral regulation of microbial CUE? Or does it represent the efficiency of transfers between operationally defined SOC pools, which are sensitive to both microbial physiology and both the direct and indirect (physiologically mediated) effects of mineralogy?

Lines 688-689: So this suite of environmental variables might predict CUE? In this case, aren't there "environmental modifiers" at play here, in that the environment modifies CUE?

Lines 702-710: This is a fairly weak cross validation approach: only 10% of the data, sampled with replacement (so presumably the same observations can be used in testing and training). In addition it would be much better to test on spatially coherent regions rather than by random sampling: spatial autocorrelation between training and testing data will yield artificially high performance on testing (Roberts et al. 2017: <https://doi.org/10.1111/ecog.02881>). This should be standard practice when performing cross validation on spatial data—I would go ahead with it, even if many papers are unfortunately still published with spatially random testing data.

Figure 2: In panel a, the R squared value is actually equal to 0.55? This is surprising—unless "explained variation" refers to some other statistic? Also, the p values can't be zero—please report them in terms of a maximum values, e.g. <0.001.

Figure 4: Proportional change in mechanisms: are these mechanisms? Or model components/indices? I think the latter.

SI lines 96-98: Based on SI figure 2, it seems that the CUE-SOC relationship is strongly modulated by soil texture and pH. In this case, is "microbial CUE" the control, or is this modeling exercise identifying some sort of emergent interplay between microbes and the soil physico-chemical environment?

SI table 2: what is "nonmicrobial biomass"?

SI table 8: It wasn't clear until I reached this table, but it seems that the actual CUE term in the microbial model is being largely ignored here. Instead, a more aggregated "system level" CUE index is being calculated. How does the actual CUE parameter (mic_cue) behave? This seems critical.

SI table 10: Water holding capacity is not a chemical property, and soil bulk density is separate from soil texture.

SI figure 7: What do these environmental relationships mean, and why do they emerge? What controls CUE? This is not a minor question, and should get more attention.

+++++

Additional comments by referee #5

I think this manuscript would require more than a few caveats—its central argument would need to be fundamentally revised. For instance, I do not think the title of the paper is a defensible statement, since it largely depends on the assumption that CLM5-derived CUE is equivalent to “microbial CUE”. Like the title, many of the arguments made in the paper appear to be significant overstatements given that the modelling approaches used in the analysis do not isolate microbial CUE from other factors.

It is difficult for me to imagine what the manuscript would look like if the arguments were scaled back appropriately. Consequently I am afraid I can’t recommend accepting this manuscript. I realize that a great deal of effort has been put into producing and reviewing this research, so this is a difficult verdict to communicate. My apologies that I cannot offer a more positive assessment!

I think the approach that depends on CLM5 is fundamentally not appropriate for isolating microbial CUE—it can only provide a proxy. The “CUE_mic” parameter in the microbial model could arguably be used to isolate CUE via model inversion (this would be distinct from the approach used in the manuscript: see section R3.1c of my review for details). Refocusing the paper on the microbial model in this way would require a significant re-write, but this might be one path forward. I still have significant reservations however, including the two about data sources (points 4 and 5). I stand by my overall assessment of the manuscript.

Author Rebuttals to Third Revision:

Summary of point-by-point responses to comments by referee #5

We very much thank referee #5 for offering us insightful and constructive comments on the manuscript, which are extremely helpful for us to revise the manuscript. Here is a summary of the revision of our manuscript.

1. Model-retrieved microbial CUE: Following the suggestions by referee #5, we used a microbial model only and removed CLM5 from the revised manuscript. The microbial model explicitly represents the organic carbon partitioning processes in microbial metabolism along three litter and one mineral soil organic carbon pathways. In the revised manuscript, we adopted an analytical solution for the microbial model from a published paper (Georgiou, et al., 2017. *Nature Communications* **8**, 1-10) to conduct all the analyses as we did previously with 57,267 vertical SOC profiles across the globe. We found that our original conclusions remain the same as before on the CUE-SOC relationship and the relative importance of CUE in comparison with six other components. Please see R5.1a for detailed explanation.

2. System-level CUE: The model-retrieved system-level CUE was calculated according to the definition (i.e., $CUE = \frac{\text{biomass production}}{\text{substrate uptake}}$) by integrating all the CUE values of the microbial assimilation pathways for both litter organic carbon and dissolved organic carbon (DOC) in the mineral soil. The CUE for the mineral soil part is conceptually closer to the values measured in those experiments compiled in our meta-analysis. We also found that the system-level CUE is strongly correlated with the CUE for the soil part (Pearson correlation coefficient = 0.98, df = 56,270, P < 0.001). Please see R5.2d for more details.

3. Meta-analysis: We examined measured CUE values using carbon (¹³C or ¹⁴C) and oxygen (¹⁸O) isotopes in our meta-analysis. The mean (variance) values are 0.33 (0.049) using the carbon isotopes and 0.33(0.022) using the oxygen isotope. Our analysis using the mixed-effects model shows that the positive CUE-SOC relationship holds across studies using different isotopic methods. Please see R5.2g for more details.

4. Profile selection criteria: We have relaxed our profile selection criteria. With the relaxed criteria and the newly adopted analytical solution, we used 57,267 soil profiles of SOC in our analysis instead of 2,500 profiles in the previous analysis. We found no significant

discrimination against profiles belonging to any soil orders or ecosystems. The profiles used in this study cover all soil orders (including Spodosol) and are inclusive to those with irregular vertical SOC shapes. While the majority of the 57,267 profiles (66.2%) showed monotonically decreasing SOC stocks with soil depths, 4.4% of them recorded the highest SOC stock at the middle of the soil depths and 29.4% of them showed zigzagged SOC stock with increasing soil depths. Please see R5.2i for more details.

We hope that our responses and revision of the manuscript are satisfactory to the referee. And we look forward to further feedback and comments.

Below are our point-by-point responses (in blue) to referees' comments (in black).

Referee #5 (Remarks to the Author):

Review of manuscript: “Microbial carbon use efficiency promotes global soil carbon storage”

Overall assessment:

The manuscript’s central thesis is that microbial physiological carbon use efficiency is the dominant control on soil carbon accrual worldwide. The authors find support for this argument in the results of data assimilation and sensitivity analysis applied to a first order linear and a microbially explicit soil carbon model.

This manuscript is technically impressive, generally easy to follow, and ambitious in its scope. However, in my view its central argument is based on a misapplication of the concept of “microbial carbon use efficiency”. The transfer coefficients in first-order soil carbon models are not equal to “microbial carbon use efficiency”, although these two things are clearly related. I am going to elaborate on this point because I do not think it should be readily dismissed.

Response: We thank the referee for offering us constructive comments. In the revised manuscript, we no longer used CLM5 but only used the microbial model that explicitly expresses microbial carbon partitioning processes as the backbone to retrieve microbial CUE via data assimilation and deep learning. We adopted an analytical solution for the microbial model from a published paper (Georgiou, et al., 2017. *Nature Communications* **8**, 1-10) so that we can integrate all available SOC profiles with the process-based model by the PRODA approach. The main conclusions on the positive CUE-SOC relationship as well as the relative importance of CUE in comparison with other model components remain the same as before. R5.1a below describes the details of what we have done using the microbial model.

Transfer coefficients may indeed implicitly incorporate microbial carbon use efficiency — and yet they also implicitly incorporate other factors. For instance, as originally conceptualized in the 1980s, these parameters represented the efficiency with which organic matter was “humified” or converted to more “recalcitrant” compounds; or alternatively the efficiency with which minerals protect or stabilize SOC (Parton et al. 1987; Parton et al. 1994). For instance, the transfer coefficient in DAYCENT – a prototypical first order SOC

model that forms the basis for the SOC representation in CLM5 – was originally conceptualized as soil-texture dependent, reflecting the efficiency of physical stabilization by the mineral matrix (Parton et al. 1987). Transfer into the passive SOC pool was described in a purely empirical way to approximate SOC residence times, with no reference to microbial physiology and no clear physiological interpretation (Parton et al. 1987). In this sense the coefficients describe an emergent process that combines microbial physiology and the other abiotic (non-physiological) processes.

R5.1a We thank the referee for the insightful comments. We agree with the referee that transfer coefficients used in CLM5 do not explicitly represent the concept of microbial carbon use efficiency (i.e., the fraction of microbial metabolic substrate carbon for microbial biomass accumulation). To address the referee’s concerns, we removed all the text and results related to CLM5 from the revised manuscript and used a microbial model that explicitly represents the microbial CUE as the backbone of this study to discuss the relative importance of CUE to global SOC storage in comparison with other components.

Specifically, we kept the basic structure of carbon cycling through the four mineral soil pools (i.e., enzymes, DOC, mineral-associated SOC, and microbial biomass) in the microbial model used in the previous version of the manuscript. The microbial CUE is estimated as the efficiency of microbial assimilation of dissolved organic carbon (DOC) in the mineral soil to microbial biomass (i.e., η_{DOC} in Response Letter Table 1). To isolate microbial from non-microbial processes in litter carbon cycling, we slightly modified the carbon transfers from litter organic carbon pools to mineral soil carbon pools (Response Letter Fig. 1). When carbon transfers from three litter pools (i.e., metabolic, cellulose, and lignin litter pools) to the soil microbial pool, CO₂ is released and microbial CUE can be estimated for assimilating organic carbon from the three litter pools into microbial biomass (i.e., η_{ML} and η_{CL-LL} in Response Letter Table 1). The direct carbon transfers from the metabolic and cellulose litter pools to the DOC pool or from the lignin litter pool to the mineral-associated SOC (mSOC) pool do not involve microbial processes nor release CO₂ (Response Letter Fig. 1). By doing so, we were able to isolate microbial CUE from other processes in our analysis with the microbial model.

Our results using the microbial model showed a positive correlation between CUE and SOC storage (Response Letter Fig. 2). The spatial patterns of CUE and other components are similar with those in the previous version of the manuscript (Response Letter Fig. 3). CUE is still the most important component in determining the global SOC storage and its

spatial distributions in comparison with the other six components investigated in this study (Response Letter Fig. 4). We also updated all other related figures and tables (please see figures and tables in the Extended Data and Supplementary Information). The detailed description of the microbial model is in Section 2 of the manuscript (page 24 - 29).

Response Letter Table 1 | Parameters in the vertically-resolved microbial model that were optimized in the profile-level data assimilation.

No.	Name	Related components	Description	Conventional values	Unit	Prior range
1	η_{DOC}		Microbial CUE for DOC assimilation	0.6	unitless	[0.01 0.7]
2	η_{ML}		Microbial CUE for metabolic litter assimilation	0.5	unitless	[0.4, 0.9]
3	η_{CL-LL}	Microbial carbon use efficiency	Microbial CUE for cellulose/lignin litter assimilation	0.5	unitless	[0, 0.4]
4	$K_{m,assim}$		Concentration of DOC for half max DOC assimilation reaction	4×10^2	gCm^{-3}	[300 3000]
5	τ_{assim}		Inverse of $v_{max,assim}$ in DOC assimilation	0.011	year	[0.03 0.001]
6	τ_{decom}		Inverse of $v_{max,decom}$ in SOC decomposition	1.1×10^{-4} , 4.6×10^{-5} , 2×10^{-7}	year	[0 3×10^{-4}]
7	$K_{m,decom}$		Concentration of SOC for half max SOC decomposition reaction	6×10^5	gCm^{-3}	[10^5 10^6]
8	$\tau_{ENZ,prod}$		Turnover time for enzyme production	22	year	[15 30]
9	τ_{ML}	Decomposition	Turnover time of metabolic litter	0.0541	year	[0 0.1]
10	τ_{CWD}		Turnover time of coarse woody debris	3.33	year	[1 6]
11	τ_{CL-LL}		Turnover time of cellulose and lignin litter	0.2041	year	[0.1 0.3]
12	$\tau_{ENZ,decay}$		Turnover time for enzyme decay	0.11	year	[0.001 1]
13	τ_{MIC}		Turnover time for microbial mortality	0.57	year	[0 2]
14	$a_{SOC,MIC}$		Fraction of microbial necromass that is stabilized as SOC	0.5	year	[0 1]
15	$a_{CL,CWD}$		Fraction of decomposed CWD that goes to cellulose litter	0.75	unitless	[0.5, 1]
16	$a_{DOC,ML}$	Carbon transfer fraction	Fraction of total decomposed metabolic litter that goes to DOC	0.05	unitless	[0 0.1]
17	$a_{DOC,CL}$		Fraction of total decomposed cellulose litter that goes to DOC	0.15	unitless	[0.05 0.3]
18	$a_{SOC,LL}$		Fraction of total decomposed cellulose litter that goes to SOC	0.8	unitless	[0.6 0.95]
19	$w\text{-scaling}$	Environmental modification	Scaling factor to soil water scalar	1	unitless	[0 5]
20	$q10$		Temperature sensitivity	1.5	unitless	[1.2 3]
21	$cryo$	Vertical transport	Cryoturbation rate	0.0005	m^2yr^{-1}	[3×10^{-5} 16×10^{-4}]
22	$diffus$		Bioturbation rate	0.0001	m^2yr^{-1}	[3×10^{-5} 5×10^{-4}]
23	b	Carbon input allocation	Parameter controlling vertical distribution of carbon input to litter pools	PFT dependent	unitless	[0.5 1]

Response Letter Fig. 1 | Structure of the microbial-process explicit model used in this study.

Response Letter Fig. 2 | The CUE-SOC relationship that emerged from the meta-analysis of 132 measurements (a) and data assimilation using the microbial model with 57,267 globally distributed vertical SOC profiles (b). Black lines and statistics shown are

the partial coefficients from mixed-effects model regressions (see Extended Data Tables 1-2 for details). This figure serves as Fig. 2 in the main text.

Response Letter Fig. 3 | Maps of global SOC stock and the underlying components. This figure serves as Fig. 3 in the main text.

Response Letter Fig. 4 | Microbial CUE as the primary regulator of global SOC storage. This figure serves as Fig. 4 in the main text.

My point here is not that we should remain bound by old (and perhaps outmoded) theoretical interpretations (like the recalcitrance concept, which I am not defending); rather, it is to emphasize that the transfer coefficients in first order SOC models are nebulous, operationally defined parameters, and that our interpretation of them is somewhat subjective. In this case, there is no strong basis for interpreting these coefficients as “microbial CUE” alone to the exclusion of other factors.

R5.1b We greatly appreciate referee #5 for pointing out this critical difference between the transfer coefficients and microbial CUE. Now, we used the microbial model to isolate microbial CUE from other factors (please see our responses in R5.1a).

From a more technical standpoint, first order soil carbon models do not directly simulate microbial metabolism and growth; hence there is no clear way to invert a first order model to estimate microbial CUE as defined in the manuscript (Lines 47-49). The “bulk CUE” index discussed in this manuscript might be a rough proxy for microbial CUE and ought to be correlated with it at some level (as the authors clearly show), but demonstrating that a model component is correlated with a measurement is not convincing evidence that the model component and the measurement are synonymous. Consequently, the analysis focused on CLM5—which makes up the backbone of the paper—does not actually isolate the role of microbial physiology in carbon sequestration, and the manuscript overreaches by assigning too much importance to microbial physiology.

R5.1c We thank the referee for the comments. In the revised manuscript, we isolated the microbial CUE from other factors by using the microbial model that explicitly represents the microbial carbon partitioning processes in soils. Please see R5.1a for more details.

Evaluation of responses to reviewer 3’s comments:

I have been asked to specifically evaluate the author’s response to the reviewer 3’s concerns. Here is my point by point assessment, using the authors’ codes designating each point of contention:

Response: We thank the referee for doing so.

R3.1a: I concur with reviewer 3 that “model component” is a more appropriate term than

“mechanism”. This has been addressed adequately, except in Figure 4, where the term mechanism still needs to be replaced.

R5.2a We thank the referee for pointing out this typo. We corrected the “mechanisms” to “components” in Fig. 4.

R3.1b: Here I think that the authors may have missed the point. It does seem true that in a purely mathematical sense, bulk CUE depends to some extent on the factors that it is being compared to, and the authors have done a good job of addressing this issue. However, the fact remains that bulk CUE extracted from CLM5 does not isolate the role of microbial physiology in carbon cycling. Perhaps it is the interpretation rather than the method of calculation that is the real issue (see overall assessment above).

R5.2b We thank the referee for the comments. The revised manuscript used Equation 1 (page 20 of the revised manuscript) in the Method section to represent CUE as ratio of microbial biomass production over substrate uptake. We calculated system-level CUE (Equation 10 on page 34 of the revised manuscript) by lumping all the terms of microbial biomass production (Equation 9 on page 33 of the revised manuscript) and substrate uptake (Equation 8 on page 33 of the revised manuscript) from the three litter pools and one DOC pool. We hope our expression in the revised manuscript can avoid any confusion. Meanwhile, we used the microbial model as the backbone in the revised manuscript to isolate microbial CUE from other factors.

R3.1c: The authors have addressed this concern adequately at some level by expanding the role of the microbial model. I would like to point out, however, that a correlation between bulk CUE extracted from CLM5 and CUE derived from the microbial model is not evidence that CLM5 can isolate microbial physiology; rather it is evidence that the transfer coefficients from CLM5 are a suitable proxy for physiological CUE. Broadly: correlations alone cannot be used to argue that two quantities are equivalent.

R5.2c We thank the referee for the comments. We have re-conducted all the analyses using the microbial model that explicitly represents microbial carbon partitioning processes to isolate microbial CUE from other factors. Please see R5.1a for more details.

In addition, this response to Reviewer 3’s comment is illuminating, in that it explains clearly that “system level” CUE was extracted from the microbial model, which was not entirely clear in the main text of the manuscript. I appreciate that retrieving “system level” CUE was important for intercomparability with CLM5, but on the other hand this decision undermines the utility of the microbial model. The microbial model contains an actual CUE parameter which governs partitioning between microbial growth and respiration in a clearly interpretable way. This parameter really might be directly relatable to the CUE measurements collecting in the meta-analysis (which cannot be said for CLM5 transfer coefficients). Are the optimized values of this parameter equal to “system level” CUE? If not, what does system level CUE represent?

R5.2d We thank the referee for the comments. Now we completely revised the description in the manuscript on calculation of system-level CUE with the microbial model. Specifically, we calculated the system-level CUE strictly following the definition of microbial CUE (i.e., $CUE = \frac{\text{biomass production}}{\text{substrate uptake}}$) (see Equation 1 on page 20 of the manuscript).

The system-level CUE (i.e., CUE_{system}) integrates four CUE values along the microbial assimilation pathways, three from the litter organic carbon pools and one from the DOC of the mineral soil part (see R5.1a for detailed description). The calculation is described in the main text (L713 – L741) as below:

“We calculated the system-level CUE of the microbial model according to its definition using Equation 1. Specifically, assimilating both litter organic carbon and DOC of the mineral soils contributes to the production of microbial biomass (Extended Data Fig. 3). In these processes, the total substrate carbon utilised in microbial metabolism is:

Substrate uptake

$$= \sum_i \sum_z [x_{L_i,z} k_{L_i,z} (1 - a_{nonMIC,L_i}) \xi_z \Delta z] + \sum_z \left(v_{max,assim} \xi_z x_{MIC,z} \frac{x_{DOC,z}}{K_{m,assim} \xi_z + x_{DOC,z}} \Delta z \right) \quad (8)$$

where Δz is the thickness of the z th soil layer and i is from 1 to 3, representing metabolic, cellulose and lignin litter pools. The a_{nonMIC,L_i} from the A matrix (Equation 6) is $a_{DOC,ML}$ for

metabolic litter, $a_{DOC,CL}$ for cellulose litter, and $a_{mSOC,LL}$ for lignin litter. Correspondingly, the total microbial biomass production is:

Biomass Production

$$= \sum_i \sum_z [\eta_{L_i} x_{L_i,z} k_{L_i,z} (1 - a_{nonMIC,L_i}) \xi_z \Delta z] + \sum_z \left(\eta_{DOC} v_{max,assim} \xi_z x_{DOC,z} \frac{x_{DOC,z}}{K_{m,assim} \xi_z + x_{DOC,z}} \Delta z \right) \quad (9)$$

Substituting Equations 8-9 into Equation 1 gives the system-level CUE:

CUE_{system}

$$= \frac{\sum_i \sum_z [x_{L_i,z} k_{L_i,z} (1 - a_{nonMIC,L_i}) \xi_z \Delta z] + \sum_z \left(v_{max,assim} \xi_z x_{MIC,z} \frac{x_{DOC,z}}{K_{m,assim} \xi_z + x_{DOC,z}} \Delta z \right)}{\sum_i \sum_z [\eta_{L_i} x_{L_i,z} k_{L_i,z} (1 - a_{nonMIC,L_i}) \xi_z \Delta z] + \sum_z \left(\eta_{DOC} v_{max,assim} \xi_z x_{DOC,z} \frac{x_{DOC,z}}{K_{m,assim} \xi_z + x_{DOC,z}} \Delta z \right)} \quad (10)$$

The CUE_{system} combines all the microbial carbon use efficiencies for both litter organic carbon and DOC into one single metric. The variation of CUE_{system} is mainly controlled by CUE of the mineral soils (i.e., η_{DOC} in Supplementary Table 6). We found a strong correlation between the system-level CUE and η_{DOC} retrieved from the 57,267 SOC profiles via data assimilation (Pearson correlation coefficient = 0.98, $df = 56,270$, $P < 0.001$, Supplementary Fig. 1).”

The correlation between the system-level CUE and microbial CUE (η_{DOC}) retrieved from the 57,267 SOC profiles via data assimilation is also presented here as Response Letter Fig. 5.

Because litter is often removed from soil samples before incubations in the experiments for studying microbial CUE, we used the retrieved microbial CUE of the mineral soil part (i.e., η_{DOC} in Response Letter Table 1) to show the CUE-SOC relationship (Fig. 2 of the manuscript), which is comparable with the relationship from our meta-analysis. When we evaluated the relative importance of the seven components in determining SOC storage, however, we used system-level values of CUE and other components.

Response Letter Fig. 5 | Dependence of system-level CUE on the CUE of the mineral soil part.

These questions relate to the broader debate about interpreting CUE measurements. From a more empirical standpoint, CUE can have multiple meanings and interpretations depending on the scale of analysis or the method used (Geyer et al. 2016; Geyer et al. 2019). Does system level CUE isolate the ratio of carbon used for growth versus metabolism? Or is it an index that is also sensitive to other factors like microbial turnover (see Hagerty et al. 2014; Hagerty et al. 2018)? The manuscript would benefit from more explicit consideration of these questions, and some direct discussion of how the “cue_mic” parameter relates to system level CUE.

R5.2e We thank the referee for the suggestions. As explained in R5.2d, the system-level CUE does isolate the ratio of carbon used for growth versus metabolism. When microbial turnover occurs, microbial biomass carbon goes through a cycle of transfer from microbial carbon pool to DOC, mSOC, and enzyme pools back to the microbial pool with a fraction of carbon released to CO₂. Thus, we can isolate microbial CUE from other factors such as microbial

turnover. In R5.2d, we discussed the strong dependence of the system-level CUE on the CUE of the mineral soil part (i.e., η_{DOC}).

R3.2a: The authors have addressed this comment by comparing PRODA retrieved CUE estimates to the observations from the meta-analysis. This is good, but it also raises new issues. First, Response Letter Figure 4 clearly shows that CLM5-derived CUE and CUE from field experiments are correlated, but the relationship appears to be strongly biased (i.e., the slope is very far from 1:1). CLM5/PRODA retrieved CUE mostly varies between 0.35 and 0.42, whereas observed CUE values vary between 0.1 and 0.7. This is related to the slope mismatch issue that Reviewer 3 raised earlier and which the authors address at length. I think at some level this mismatch is to be expected because “bulk” or “system level” CUE estimate from CLM5 is only roughly analogous to the CUE being measured in experiments—so we might expect a positive relationship, but not a 1:1 relationship. This supports the interpretation that CLM5-derived CUE is a proxy for true microbial CUE and not identical to it.

R5.2f We thank the referee for the comments. We used the results by the microbial model in the revised manuscript to show the CUE-SOC relationship. The microbial model-retrieved CUE for the mineral soils (i.e., η_{DOC}) presented similar variations with those from the meta-analysis and thus we observed similar slopes in the CUE-SOC relationships (Response Letter Fig. 2). The issue related to CLM5-derived CUE no longer exists as we completely deleted CLM5 results from this version of the manuscript.

I would also like to point out a concern with the meta-analysis dataset. My concern is that the analysis combines ^{13}C -based and ^{18}O -based CUE estimates. It has been shown clearly that these two methods are actually measuring fundamentally different quantities—one substrate specific CUE, the other non-specific (Geyer et al 2019). We really ought to stop treating these measurements as synonymous—no level of statistical analysis can get around the fact that they are based on different assumptions and typically yield very different values (e.g., ~ 0.6 for ^{13}C glucose CUE, versus around 0.3 for ^{18}O CUE). A lot of the variation in compilations of CUE measurements might actually be due to the fact that different ways of measuring or calculating CUE will automatically yield different answers.

R5.2g We greatly appreciate this referee for pointing out this issue. We checked our data sets used in the meta-analysis. The mean (variance) of measured CUE is 0.33 (0.049) with the $^{13}\text{C}/^{14}\text{C}$ method and 0.33 (0.022) with the ^{18}O method (Response Letter Fig. 6).

Response Letter Fig. 6 | Distribution of microbial CUE values measured by $^{13}\text{C}/^{14}\text{C}$ ($n = 21$) and ^{18}O ($n = 111$) methods in the meta-analysis.

We also used the mixed-effects model to examine if the isotope methods would affect CUE-SOC relationship. We used the study sources (“Source” in Supplementary Table 1) as the random effects for different isotope methods to estimate CUE. The positive CUE-SOC relationship shown in Fig. 2a is the result after considering the impacts of different CUE values measured by different methods (detailed statistics is shown in Extended Data Tables 1 - 2). We also conducted the mixed-effects model analyses by separately applying data measured by either carbon isotopes (i.e., $^{13}\text{C}/^{14}\text{C}$) or oxygen isotope (i.e., ^{18}O) (Supplementary Table 7). While results from the mixed-effects model suggest that the random factor (i.e., study source) contributes to the variation in both the slopes and intercepts of the CUE-SOC relationship (i.e., the variation of random effects as shown in Extended Data Table 1), the positive relationship between CUE and SOC is supported by all the above analyses.

R3.2b I concur with Reviewer 3 that the large difference in the CUE-SOC slope in the

observations and the CLM5-derived CUE estimates is a major concern (this relates to my response to R3.2a above). It is encouraging that the microbial model produces a closer match, but this does nothing to increase my confidence that CLM5 can be used to represent the experimental data accurately, and the manuscript remains highly reliant on the CLM5 analysis.

R5.2h We understand the concerns by the referee. We have used the results of the microbial model only in the revised manuscript. The CUE-SOC relationship emerged from the microbial model data assimilation showed a similar slope with that from the meta-analysis (Response Letter Fig. 2).

R3.2c Here I think the authors have done a good job of addressing Reviewer 3's comment. However, I have a major concern that is related. I am concerned about the fact that during model optimization, 72,350 soil profiles were originally considered but ultimately only 52,819 were analyzed (27% excluded). In the Methods, it is mentioned that profiles were excluded if they did not show a clear monotonic decline in SOC with depth. It is implied that SOC depth profiles that show a bimodal SOC distribution are the result of measurement errors or some sort of aberrant "geologic process": "While these atypical vertical SOC distributions could be caused by geological processes even if not by measurement errors, they may not offer information to help understand processes underlying SOC storage." I do not think this is a defensible argument. In particular, an entire soil order – the Spodosol order in the USDA taxonomy – is partly defined by the presence of a sandy, relatively SOC-poor E horizon that overlies a finer textured B horizon rich in reactive Al and Fe oxyhydroxides and SOC. The process of soil development can naturally generate a bimodal SOC distribution in this case: this is most certainly a "process underlying SOC storage". I am worried that the down-selection procedure has systematically biased this analysis to exclude soils with complex SOC-depth profiles that emerge due to pedogenesis. If this is true, the analysis might be downplaying the role of minerals in protecting microbial products and hence predetermining its findings at some level.

R5.2i We thank the referee for the comments on our profile selection criteria. We relaxed the selection criteria. Now we used 57,267 profiles in this study, instead of the original 2,500 profiles in the previous version, for the microbial model. We used all the 72,377 SOC profiles in our site-level data assimilation. We adopted two criteria to select the data

assimilation results. The G-R statistics quantifies the convergence of the estimated parameters from three independent Markov chains in data assimilation. A larger G-R value indicates inconsistent results on the estimated parameters from independent data assimilation runs (i.e., equifinality). Results with strong equifinality issue will hinder the neural network model to generalise site-level data assimilation results to the global scale (i.e., too much noise in training data) and potentially introduce more overfitting problems. Thus, we set a threshold (i.e., $G-R < 1.05$) to control the convergence of data assimilation results and thus control the overfitting problem in the later neural network training (see R5.4e for more details). After this procedure, a total of 59,476 profiles was left. Moreover, we used the modelling efficiency (i.e., E , see Equation 2 of the manuscript) as one additional metric in selecting profiles. A small value of E indicates that the model cannot capture the variability in the data, suggesting that such SOC vertical profiles may not offer enough information on the processes underlying SOC storage investigated in this study. In the revised manuscript, we relaxed the threshold of E from $E > 0.75$ to $E > 0.0$ to include as many profiles as possible. Eventually, we used 57,267 profiles in this study.

Our profile selection criteria did not cause significant discrimination to profiles in specific soil orders or ecosystems. The profiles eventually used in this study covers all ecoregions and soil orders (including Spodosols, Response Letter Fig. 7). Meanwhile, the soil profiles included in this study are inclusive to those with irregular vertical shapes (Response Letter Fig. 8). While the majority of the 57,267 profiles (66.2%) show monotonically decreasing SOC stocks with soil depths, 4.4% of them record the highest SOC stock at the middle of the soil depths and 29.4% of them show zigzagged SOC stock with increasing soil depths. We apologize that the descriptions in the previous version of the manuscript were not accurate.

In the revised manuscript, we introduced our profile selection criteria with more details (L468 – L510) and added the results about criteria inspection (Supplementary Figs. 5 - 6).

Response Letter Fig. 7 | Coverage of different sources of data in multi-dimensional covariate spaces. Panels show the percentage of data sites located at different climates (a), soil textures (b) soil orders (c), and land cover types (d) in the profiles used in the PRODA approach of this study (i.e., 57,267 profiles) and the meta-analysis with 132 data sets. For different climate types: Af, Am and Aw are tropical rainforest, monsoon and savannah climates, respectively. BW and BS are arid desert and steppe climates, respectively. Cs, Cw and Cf are temperate climates with dry summer, dry winter, and without dry season, respectively. Ds, Dw and Df are cold climates with dry summer, dry winter, and without dry season, respectively. E is polar climate. For different soil texture, Cl is clay, SiCl is silty clay, SaCl is sandy clay, ClLo is clay loam, SiClLo is silty clay loam, SaClLo is sandy clay loam, Lo is loam, SiLo is silty loam, SaLo is sandy loam, Si is silt, LoSa is loamy sand, Sa is sand.

Response Letter Fig. 8 | Different vertical shapes of SOC profiles used in this study. Shown in the figure are 1,000 profiles randomly selected from the 57,276 profiles. SOC values are normalised by the value at the first soil layer of each profile.

R3.3a: This response shows that the data assimilation algorithm has a strong influence on the range of system level CUE values extracted from the model. When CLM5 is optimized via the SCE procedure, it yields a CUE SOC relationship several times less steep than if it is optimized by the MCMC procedure. This is an interesting methods comparison and I laud its thoroughness, but ultimately this result indicates that CUE extracted from the data-assimilation / PRODA approach is sensitive to the algorithm used. This makes it very hard to evaluate the degree of agreement between the models, or between the models and observations: to what extent are these results an artifact of the method applied? The fact that SCE and MCMC derived results are correlated is small comfort given the difference in output ranges when different algorithms are applied.

R5.2j We thank the referee for the comments. In the revised manuscript, we adopted an analytical solution reported by Georgiou et al. (2017)¹ for the steady state values of the mineral soil organic carbon pools in the microbial model (Equation 7 on page 28 of the

revised manuscript). The analytical solution enables the MCMC method for data assimilation. We used the MCMC method instead of the SCE method in the microbial model data assimilation. (The SCE method presents more equifinality problem than the MCMC method as we discussed in the previous version of the manuscript.) Now, the CUE-SOC relationship retrieved from the 57, 267 vertical profiles of SOC is similar to that from the meta-analysis. In addition, estimations of parameters from the MCMC method at site level are more easily to be generalized to the global scale by the neural network.

R3.3b: Once again, this response is based on correlations, not a rigorous analysis of the absolute goodness of fit between different model outcomes or the observations. Correlation statistics are not strong evidence that the analysis is robust to the model type or data assimilation approach, particularly given that the absolute values of the CUE estimates occupy very different ranges.

R5.2k We thank the referee for the comments. In the revised manuscript, we used the microbial model alone to do data assimilation and other analyses. The variance of CUE retrieved from the microbial model after data assimilation is similar to that observed in the meta-analysis (Response Letter Fig. 2). We extracted the CUE values predicted by the PRODA approach in those pixels where the CUE was measured for the meta-analysis and then evaluated the agreement between the model-retrieved CUE values (i.e., CUE_{proda}) and field measurements (i.e., CUE_{meta}) by the mixed-effects model. We considered the methodological differences among different studies as the random effects in the mix-effects model (i.e., $CUE_{meta} \sim CUE_{proda} + (1|Study\ Source)$). The regression analysis indicates that CUE estimated with the PRODA-optimised microbial model agrees well with the field observations ($R^2 = 0.54$, Response Letter Table 2). The regression slope is not significantly different from 1 at a significance level of 0.05 (Response Letter Table 2).

In the revised manuscript, the Response Letter Table 2 is presented as Supplementary Table 5.

Response Letter Table 2 | Relationship between PRODA-retrieved CUE values in these pixels where CUE was measured in the meta-analysis and the measured values. CUE predicted by the PRODA approach (CUE_{proda}) was set as the fixed effects to measurements in the meta-analysis (CUE_{meta}). The study source was set as the random effect. We assumed

random intercepts in the regression. The total observation size $n_{sample} = 132$; the random effects size $n_{study} = 16$. The difference of the regression slope between CUE_{meta} and CUE_{proda} from 1 was tested by offsetting CUE_{proda} in the same mix-effects model structure.

		Intercept	CUE_{proda}
$CUE_{meta} \sim CUE_{proda} + (1 Study\ Source)$, $R^2 = 0.54$			
Fixed Effects	Estimates	0.14	0.66
	Std. Error	0.068	0.22
	t value	1.99	3.01
	P	0.050	0.0032
Random Effects	Standard Deviation	0.11	NA
$CUE_{meta} \sim CUE_{proda} + 1 * CUE_{proda} + (1 Study\ Source)$			
Fixed Effects	Estimates	0.14	-0.34
	Std. Error	0.068	0.22
	t value	1.99	-1.56
	P	0.050	0.12
Random Effects	Standard Deviation	0.11	NA

Summary of critiques:

(1) Microbial CUE has been defined too broadly, and is not a good descriptor of the model components being optimized in the CLM5 case, which do not isolate microbial physiology or represent microbial growth and metabolism.

R5.3a We thank the referee for all the constructive critiques. In this revision, we deleted CLM5 results from this manuscript and only used the microbial model which explicitly represents the microbial carbon partitioning processes to isolate CUE from other factors. The main conclusions of this study remain the same. Please see our responses in R5.1a for details.

(2) The actual CUE parameter in the microbial model is not compared to “system level” CUE, and the exact relationship between these two quantities is not explored.

R5.3b We apologize for not clearly explaining the system-level CUE in the previous version of the manuscript. In this revision, we calculated the system-level CUE strictly following the

definition of microbial CUE (i.e., $CUE = \frac{\text{biomass production}}{\text{substrate uptake}}$). The system-level CUE integrates all the microbial CUE values along the microbial assimilation pathways for both litter organic carbon and DOC of the mineral soil part (see R5.2d). We found a strong dependence of system-level CUE on the CUE for the mineral soil part (i.e., η_{DOC}).

(3) CLM5 derived CUE estimates cover a much narrower range than the observations, which undermines the idea that CLM5 derived bulk CUE is equivalent to measured microbial CUE.

R5.3c We removed results of CLM5 from the revised manuscript and used the results only from the microbial model as the backbone of this study (R5.1a). CUE retrieved from the microbial model presents a similar range with that in the meta-analysis.

(4) Observed CUE data in the meta analysis combine methods that quantify fundamentally different aspects of CUE.

R5.3d We thank the referee for pointing this out. While it is true that the two methods quantify fundamentally different aspects of CUE, we checked our data sets from the meta-analysis. The mean (variance) values of the measured CUE values are 0.33 (0.049) with the $^{13}C/^{14}C$ method and 0.33 (0.022) with the ^{18}O method in our database. We also used the mix-effects model to address the methodological difference among different studies in quantifying the relationship between CUE and SOC storage and the positive CUE-SOC relationship is supported by all the analyses. Please see our responses in R5.2g for details.

(5) Down-selection of soil profiles may be systematically biased to ignore important biogeochemical processes (e.g., podzolization).

R5.3d We relaxed our profile selection criteria. We now used 57,267 profiles, instead of the original 2,500, in our analysis with the microbial model. Our analyses also suggested that our profile selection criteria do not cause significant discrimination to profiles in specific soil orders or ecoregions. Please see our responses in R5.2f for details.

(6) CUE estimates are dependent on the data assimilation algorithm used; correlations are

evidence of a relationship, but not equivalence.

R5.3d We thank the referee for pointing this issue out. We only used the MCMC method to do the data assimilation in the revised manuscript. The MCMC method became possible because we adopted an analytic solution of the microbial model. Thus, the issue caused by the data assimilation algorithm is resolved. Moreover, the CUE retrieved from the microbial model is conceptually the same with CUE measurements in the meta-analysis and agrees well with the observations in the meta-analysis. Please see R5.1a for more details.

Detailed comments:

Lines 113-116: I'm not sure this statement is true, or at least it seems like a rather slanted interpretation of CLM5. Also, as far as I can tell, the reference cited does not actually support this claim, or delve into the details of CLM5 soil biogeochemistry much at all.

R5.4a We addressed this issue by totally removing results of CLM5 from the revised manuscript.

Line 523: Here the transfer coefficients in CLM5 are called “microbial CUE”. This is a misrepresentation in my view: these coefficients are analogous to microbial CUE and implicitly include it, but are not equivalent to it, because CLM5 does not represent microbes explicitly.

R5.4b We agree with the referee. We addressed this issue by totally removing results of CLM5 from the revised manuscript.

Lines 229-237: Could the relationship with bulk density simply be because these soils have more organic C?

R5.4c We agree with the referee that soils with rich organic matter may also contribute to a lower bulk density. We have discussed the possible mutual dependence between well-structured soil and abundance of soil organic carbon in the manuscript L227 – L228.

Line 447: Why not random slopes?

R5.4d In the mixed-effects model, we considered both random slopes and intercepts in the regressions but we only showed the results that was converged in regressions (L433 – L440). Because we used the logarithmic values of SOC in regressions, the difference in slopes among different study sources may be suppressed and the mixed-effects model with random slopes cannot be converged in regressions. We also used the original values of SOC, in addition to the logarithmic values, in the mixed-effects model regressions and presented the converged results in Extended Data Tables 1 - 2.

Lines 637-638: What does it mean to optimize so many parameters (23) on a profile-by-profile basis? Aren't there many more unknowns than observations in this case? It seems like there would be a significant danger of overfitting the model. The total number of optimized parameters globally must number in the tens of thousands, unless I am misunderstanding something.

R5.4e We optimised 23 parameters in data assimilation at each profile. Thus, the optimised parameter values vary across profiles. The referee is correct that the neural network eventually predicts 57,267 sets of parameters over the globe. We controlled the overfitting issue by several means. First, we found that the vertical SOC shape is a strong constraint to parameters in addition to the number of measured SOC content values. Thus, we only kept soil profiles with more than two layers of observations in this study. Meanwhile, the depth of observation at each profile has to be deeper than 50cm to ensure sufficient information about vertical SOC shapes to constraint the model (L469 – L472). Second, we used the G-R statistics to exclude non-convergent data assimilation results that may lead to equifinality (L478 – L486). Third, the neural network used in this study is also a tool to generalise results from different profiles at the site-level data assimilation to the global scale. Only features (i.e., the relationship between parameters and environmental variables) that are commonly shown across profiles will be learnt by the neural network to eventually generate the global maps of different parameters. Fourth, the bootstrapping applied in this study further quantified the uncertainty of the neural network prediction results (L679 – L689). These means of controlling overfitting ensure effective retrieval of enough information from the 57,267 vertical profiles of SOC to constrain the seven components of the microbial model.

Lines 667-669: Does CLM5 really explicitly simulate mineral regulation of microbial CUE? Or does it represent the efficiency of transfers between operationally defined SOC pools, which are sensitive to both microbial physiology and both the direct and indirect (physiologically mediated) effects of mineralogy?

R5.4f We thank the referee for asking this issue. This issue is no longer relevant as we have totally deleted results of CLM5 from the revised manuscript.

Lines 688-689: So this suite of environmental variables might predict CUE? In this case, aren't there "environmental modifiers" at play here, in that the environment modifies CUE?

R5.4f Environmental modifiers refer to the modification of temperature and soil moisture to SOC decomposition process (L558 – L561). Here all the environmental variables are used to predict the spatial patterns of CUE and other parameters.

Lines 702-710: This is a fairly weak cross validation approach: only 10% of the data, sampled with replacement (so presumably the same observations can be used in testing and training). In addition it would be much better to test on spatially coherent regions rather than by random sampling: spatial autocorrelation between training and testing data will yield artificially high performance on testing (Roberts et al. 2017:<https://doi.org/10.1111/ecog.02881>). This should be standard practice when performing cross validation on spatial data—I would go ahead with it, even if many papers are unfortunately still published with spatially random testing data.

R5.4f We used the bootstrapping method to quantify the uncertainty of the PRODA-optimised model simulations in this study. Bootstrapping is considered as a more rigorous method to quantify uncertainty than the cross-validation method while its costs much more time to implement. Random sampling with replacement in the training set is the core feature of bootstrapping. Moreover, we only used observations that were not included in training to test the final performance of the neural network after training. Taking 10% of the data as validation in training is a common procedure in training a neural network, especially when the data is abundant.

We had considered the spatial autocorrelation in the initial version of the manuscript, but removed the related results from the revised manuscript following the suggestion by referee #2, who commented “*there is a current hype about the idea that spatial autocorrelation should be accounted for when estimating validation statistics in a spatial context. This is a misconception and I urge the authors not to propagate this wrong idea in a scientific paper... Spatial cross-validation ... provides overpessimistic validation statistics and has no underlying theory*”. Nevertheless, in the initial version of the manuscript, considering spatial autocorrelation did not influence the model performance much in a ten-fold cross-validation.

Figure 2: In panel a, the R squared value is actually equal to 0.55? This is surprising—unless “explained variation” refers to some other statistic? Also, the p values can’t be zero—please report them in terms of a maximum values, e.g. <0.001.

R5.4f The “explained variation” mentioned in this manuscript refers to the R^2 (i.e., coefficient of determination) in regressions. Because we used the mixed-effects model that considered the random effects of the identified factors in the regression (i.e., varying intercepts and slopes with different source IDs in the meta-analysis and with different climate types in data assimilation results), the performance of the mixed-effects model is usually better than a linear regression model. The fixed-effect results of the mixed-effects model are close to the results of a linear regression model, where the intercept and slope are fixed across source IDs in the meta-analysis and climate types in the data assimilation results. We also reported results of the fixed-effect in the manuscript (Extended Data Table 1).

We thank the referee for the suggestion. We have changed “P = 0” to “P < 0.001” in the related figures.

Figure 4: Proportional change in mechanisms: are these mechanisms? Or model components/indices? I think the latter.

R5.4f We thank the referee for pointing out this typo. We revised it to “components”.

SI lines 96-98: Based on SI figure 2, it seems that the CUE-SOC relationship is strongly modulated by soil texture and pH. In this case, is “microbial CUE” the control, or is this modeling exercise identifying some sort of emergent interplay between microbes and the soil

physico-chemical environment?

R5.4f We thank the referee for asking this issue. This issue is no longer relevant as we have totally deleted results of CLM5 from the revised manuscript.

SI table 2: what is “nonmicrobial biomass”?

R5.4f We defined the “non-microbial biomass carbon” as the remaining SOC after excluding microbial biomass in L103 – L104 of the revised manuscript.

SI table 8: It wasn’t clear until I reached this table, but it seems that the actual CUE term in the microbial model is being largely ignored here. Instead, a more aggregated “system level” CUE index is being calculated. How does the actual CUE parameter (*mic_cue*) behave? This seems critical.

R5.4f We thank the referee for asking this question. In this revision, we used the microbial model to retrieve the actual CUE parameter (i.e., η_{DOC}) by the PRODA approach with 57,267 vertical SOC profiles. The quantitative relationship between the actual CUE and SOC is similar with its counterpart from the meta-analysis. Meanwhile, the system-level CUE was calculated in the revised manuscript to integrate CUE values of different microbial assimilation pathways for both litter organic carbon and mineral soil organic carbon pools. The system-level CUE showed strong dependence on the CUE of the mineral soil part (i.e., η_{DOC} , the actual CUE suggested by the referee). Please see R5.1a and R5.2d for more detailed explanations.

SI table 10: Water holding capacity is not a chemical property, and soil bulk density is separate from soil texture.

R5.4f We thank the referee for the comment. We renamed the class “soil texture” to “soil structure” to make it more inclusive. We classified the water holding capacity and bulk density as properties of soil structure.

SI figure 7: What do these environmental relationships mean, and why do they emerge? What controls CUE? This is not a minor question, and should get more attention.

R5.4f We agree with the referee that the relationship between CUE and environmental variables are important to be explored. We discussed the possible explanations of these relationships in L221 - L228. We did not expand the discussion because there is a lack of observations that hinders us to make more conclusive interpretations. Meanwhile, this topic is out of the main scope of this study, that is to identify the positive or negative sign of CUE-SOC relationship and to quantify the relative importance of CUE in comparison with other components at the global scale. This topic may be explored more in future studies.

Additional comments by referee #5

I think this manuscript would require more than a few caveats—its central argument would need to be fundamentally revised. For instance, I do not think the title of the paper is a defensible statement, since it largely depends on the assumption that CLM5-derived CUE is equivalent to “microbial CUE”. Like the title, many of the arguments made in the paper appear to be significant overstatements given that the modelling approaches used in the analysis do not isolate microbial CUE from other factors.

It is difficult for me to imagine what the manuscript would look like if the arguments were scaled back appropriately. Consequently, I am afraid I can't recommend accepting this manuscript. I realize that a great deal of effort has been put into producing and reviewing this research, so this is a difficult verdict to communicate. My apologies that I cannot offer a more positive assessment!

I think the approach that depends on CLM5 is fundamentally not appropriate for isolating microbial CUE—it can only provide a proxy. The “CUE_mic” parameter in the microbial model could arguably be used to isolate CUE via model inversion (this would be distinct from the approach used in the manuscript: see section R3.1c of my review for details). Refocusing the paper on the microbial model in this way would require a significant re-write, but this might be one path forward. I still have significant reservations however, including the two about data sources (points 4 and 5). I stand by my overall assessment of the manuscript.

R5.5a We appreciate the referee for offering us constructive comments on our manuscript, which helped us think deeper on the concept of microbial carbon use efficiency. Following the suggestions and comments by the referee, we deleted CLM5 and only used the microbial

model that explicitly represents the microbial carbon partitioning processes in the PRODA approach to isolate microbial CUE from other factors in the revised manuscript. We re-conducted all the analyses to quantify the CUE-SOC relationship and the relative importance of CUE in comparison with six other components. Our main conclusion on the critical role of CUE in global SOC storage using the microbial model remains the same with previous analyses. Moreover, we further explained how we considered the methodological differences among different studies in quantifying the CUE-SOC relationship in the meta-analysis by the mixed-effects model. We also carefully inspected our profile selection criteria to make sure there is no significant discrimination against profiles belonging to specific soil orders or ecosystems.

We hope that our responses and revisions are satisfactory to the referee.

References

- 1 Georgiou, K., Abramoff, R. Z., Harte, J., Riley, W. J. & Torn, M. S. Microbial community-level regulation explains soil carbon responses to long-term litter manipulations. *Nature Communications* **8**, 1-10 (2017).

Reviewer Reports on the Fourth Revision:

Referee #5 (Remarks to the Author):

The authors have made a significant effort to overhaul this analysis, dropping the first-order CLM model entirely and focusing on a microbial model. This has eliminated some of the basic conceptual flaws that I discussed in my previous review.

The authors have also addressed my concerns about the meta-analysis to a large extent. I am still not convinced that 18O and 13- or 14C based CUE are measuring the same thing. Even if the mean values are similar, the methods are targeting different components of substrate uptake. But at least the analysis now seems to control for this methodological difference, which is sufficient.

I also appreciate that the soil profiles used for training no longer exclude non-monotonic carbon depth patterns.

I still have qualms about the overall model-dependence of the conclusions presented here, but at this point I think it would be better if these play out in future scientific debate rather than via the review process.

Two minor suggestions worth addressing:

This version of the paper relies on a steady-state solution to a microbial model. I think the steady state assumption should be clearly stated in the main body of the paper, perhaps in the paragraph introducing the modeling approach, lines 107-125. As it stands now the steady state assumption is only clear in the methods. In reality, soil carbon stocks are unlikely to be at steady state—particularly in agricultural soils. Hence the need to make this assumption clear.

I would also revise the discussion of soil physical / structural properties (lines 221-229). As it stands, this paragraph emphasized bulk density above other structural properties. Soil texture is also part of this group of variables (supplementary Figure 3). The partial dependence plots in supplementary Figure 3 do not give me a high level of confidence that the neural network is identifying actual mechanistic or function relationships between CUE and soil physical parameters—the responses are inexplicably “wiggly” and hard to interpret (typical for machine learning). Perhaps what they show is that CUE is related to soil type or structure in some more generalized sense, without suggesting a readily interpretable functional relationship.

Author Rebuttals to Fourth Revision:

Referees' comments:

Referee #5 (Remarks to the Author):

The authors have made a significant effort to overhaul this analysis, dropping the first-order CLM model entirely and focusing on a microbial model. This has eliminated some of the basic conceptual flaws that I discussed in my previous review.

Response: We thank the referee for the positive evaluation on our revision.

The authors have also addressed my concerns about the meta-analysis to a large extent. I am still not convinced that ^{18}O and ^{13}C - or ^{14}C based CUE are measuring the same thing. Even if the mean values are similar, the methods are targeting different components of substrate uptake. But at least the analysis now seems to control for this methodological difference, which is sufficient.

Response: We thank the referee for finding our responses sufficient to address her/his concerns.

I also appreciate that the soil profiles used for training no longer exclude non-monotonic carbon depth patterns.

Response: We thank the referee for the positive evaluation on our profile selection criteria.

I still have qualms about the overall model-dependence of the conclusions presented here, but at this point I think it would be better if these play out in future scientific debate rather than via the review process.

Response: We thank the referee for the comments. We agree with the referee that more discussion is needed on the importance of microbial carbon use efficiency (CUE) using different process-based models in the future. We added one sentence in the main text of the manuscript L239 – L240 to highlight the point: *“Moreover, future studies need to carefully examine how sensitive the evaluation of the relative importance of CUE to global SOC storage is to different model structures”*.

Two minor suggestions worth addressing:

This version of the paper relies on a steady-state solution to a microbial model. I think the steady state assumption should be clearly stated in the main body of the paper, perhaps in the paragraph introducing the modeling approach, lines 107-125. As it stands now the steady state assumption is only clear in the methods. In reality, soil carbon stocks are unlikely to be at steady state—particularly in agricultural soils. Hence the need to make this assumption clear.

Response: We thank the referee for the suggestion. We added one sentence in the main text L121 – L123 to state that the steady-state assumption was applied in conducting data assimilation: “A steady-state assumption for the soil carbon cycle (i.e., SOC storage does not change with time) at each observational profile is employed to facilitate computation (see Methods)”. We also provide a short paragraph in the methods L686 – L693 to explain the reasons of applying the steady state assumption: “It should note that the data assimilation was conducted under one assumption that SOC profiles are at steady state (i.e., $(dX(t))/dt=0$). This assumption makes data assimilation computationally more feasible than that under non-steady states (see the non-steady-state data assimilation^{67,68}). While soil carbon stocks in some ecosystems (e.g., agricultural soils) may not be at the steady state because of the concurrent climate change and human activities, previous research demonstrated that such disequilibrium component of the transient carbon cycle dynamics, especially in SOC pools, is minor in comparison with the amount of SOC storage that was developed over thousands of years⁶⁹.”.

I would also revise the discussion of soil physical / structural properties (lines 221-229). As it stands, this paragraph emphasized bulk density above other structural properties. Soil texture is also part of this group of variables (supplementary Figure 3). The partial dependence plots in supplementary Figure 3 do not give me a high level of confidence that the neural network is identifying actual mechanistic or function relationships between CUE and soil physical parameters—the responses are inexplicably “wiggly” and hard to interpret (typical for machine learning). Perhaps what they show is that CUE is related to soil type or structure in some more generalized sense, without suggesting a readily interpretable functional relationship.

Response: We thank the referee for the comment and suggestion. The referee is right that the main point of this paragraph is to highlight that the spatial patterns of CUE is more closely related with soil structural variables than other environmental features, rather than quantifying specific

functional relationships. We, therefore, removed Supplementary Fig. 3 from the manuscript and revised the related discussion from L245 – L252: *“Previous studies have discussed the importance of soil structural variables such as bulk density, texture and porosity in affecting microbial activities. A well-structured soil with medium physical heterogeneity may help foster niche complementarity for diverse soil microbial communities and eventually benefit high CUE⁴⁶⁻⁴⁸. In turn, accumulation of SOC due to high CUE, could also benefit the development of fertile soils. More quantitative understanding is needed on the mechanistic relationships between CUE and soil structural variables so as to facilitate the effective management of soil carbon storage in the future”*